# A Spectral Algorithm for List-Decodable Covariance Estimation in Relative Frobenius Norm

**Ilias Diakonikolas**
University of Wisconsin–Madison
ilias@cs.wisc.edu

**Daniel M. Kane**
University of California, San Diego
dakane@cs.ucsd.edu

**Jasper C.H. Lee**
University of Wisconsin–Madison
jasper.lee@wisc.edu

**Ankit Pensia**
IBM Research
ankitp@ibm.com

**Thanasis Pittas**
University of Wisconsin–Madison
pittas@wisc.edu

## Abstract

We study the problem of list-decodable Gaussian covariance estimation. Given a multiset $T$ of $n$ points in $\mathbb{R}^d$ such that an unknown $\alpha < 1/2$ fraction of points in $T$ are i.i.d. samples from an unknown Gaussian $\mathcal{N}(\mu, \Sigma)$, the goal is to output a list of $O(1/\alpha)$ hypotheses at least one of which is close to $\Sigma$ in relative Frobenius norm. Our main result is a $\mathrm{poly}(d, 1/\alpha)$ sample and time algorithm for this task that guarantees relative Frobenius norm error of $\mathrm{poly}(1/\alpha)$. Importantly, our algorithm relies purely on spectral techniques. As a corollary, we obtain an efficient spectral algorithm for robust partial clustering of Gaussian mixture models (GMMs) — a key ingredient in the recent work of [BDJKKV22] on robustly learning arbitrary GMMs. Combined with the other components of [BDJKKV22], our new method yields the first Sum-of-Squares-free algorithm for robustly learning GMMs, resolving an open problem proposed by Vempala [Vem22] and Kothari [Kot21]. At the technical level, we develop a novel multi-filtering method for list-decodable covariance estimation that may be useful in other settings.

## 1 Introduction

Robust statistics studies the efficient (parameter) learnability of an underlying distribution given samples some fraction of which might be corrupted, perhaps arbitrarily. While the statistical theory of these problems has been well-established for some time [Hub64; HR09], only recently has the algorithmic theory of such estimation problems begun to be understood [DKKLMS16; LRV16; DK23].

A classical problem in robust estimation is that of multivariate robust mean estimation — that is, estimating the mean of an unknown distribution in the presence of a small constant fraction of outliers. One of the original results in the field is that given samples from a Gaussian $\mathcal{N}(\mu, I)$ with an $\epsilon$-fraction of outliers (for some $\epsilon < 1/2$), the unknown mean $\mu$ can be efficiently estimated to an error of $O(\epsilon\sqrt{\log 1/\epsilon})$ in the $\ell_2$-norm [DKKLMS16]. Note that the use of the $\ell_2$-norm here is quite natural, as the total variation distance between two identity-covariance Gaussians is roughly proportional to the $\ell_2$-distance between their means; thus, this estimator learns the underlying distribution to total variation distance error $O(\epsilon\sqrt{\log 1/\epsilon})$. This bound cannot be substantially improved, as learning to error $o(\epsilon)$ in total variation distance is information-theoretically impossible.

While the above algorithmic result works when $\epsilon$ is small (less than $1/2$), if more than half of the samples are corrupted, it becomes impossible to learn with only a single returned hypothesis—the corruption might simply simulate other Gaussians, and no algorithm can identify which Gaussian is the original one. This issue can be circumvented using the *list-decodable* mean estimation

paradigm [CSV17], where the algorithm is allowed to output a small *list of hypotheses* with the guarantee that at least one of them is relatively close to the target. List-decodable learning is closely related to *semi-verified learning* [CSV17; MV18], where a learner can choose to audit a small amount of data. The framework has been shown to capture a multitude of applications, including *crowdsourcing* [SVC16; MV18], *semi-random community detection* in stochastic block models [CSV17] and *clustering* (as we also show in this work). For Gaussian mean estimation in particular, if an $\alpha$-fraction of the samples are clean (i.e., uncorrupted), for some $\alpha < 1/2$, there exist polynomial-time algorithms that return a list of $O(1/\alpha)$ hypotheses such that (with high probability) at least one of them is within $\tilde{O}(\sqrt{1/\alpha})$ of the mean in $\ell_2$-distance [CSV17]. Note that while this $\ell_2$-norm distance bound does not imply a good bound on the total variation distance between the true distribution and the learned hypothesis, it does bound their distance away from one, ensuring some non-trivial overlap.

Another important, and arguably more complex, problem in robust statistics (the focus of this work) is that of robustly estimating the covariance of a multivariate Gaussian. It was shown in [DKKLMS16] that given $\epsilon$-corrupted samples from $\mathcal{N}(0,\Sigma)$ (for $\epsilon < 1/2$) there is a polynomial-time algorithm for estimating $\Sigma$ to error $O(\epsilon \log 1/\epsilon)$ in the relative Frobenius norm, i.e., outputting a hypothesis covariance $\widetilde{\Sigma}$ satisfying $\|\widetilde{\Sigma}^{-1/2}\Sigma\widetilde{\Sigma}^{-1/2} - I\|_{\mathrm{F}} \leq \epsilon$. This is again the relevant metric, since, if $\|\widetilde{\Sigma}^{-1/2}\Sigma\widetilde{\Sigma}^{-1/2} - I\|_{\mathrm{F}} \leq 2/3$, then the total variation distance between $\mathcal{N}(0,\Sigma)$ and $\mathcal{N}(0,\widetilde{\Sigma})$ is proportional to the relative Frobenius norm $\|\widetilde{\Sigma}^{-1/2}\Sigma\widetilde{\Sigma}^{-1/2} - I\|_{\mathrm{F}}$ [DMR18].

A natural goal, with a number of applications, is to extend the above algorithmic result to the list-decodable setting. That is, one would like a polynomial-time algorithm that given corrupted samples from $\mathcal{N}(0,\Sigma)$ (with an $\alpha$-fraction of clean samples, for some $\alpha < 1/2$), returns a list of $O(1/\alpha)$ many hypotheses with the guarantee that at least one of them has some non-trivial overlap with the true distribution in total variation distance. We start by noting that the sample complexity of list-decodable covariance estimation is $\mathrm{poly}(d/\alpha)$, albeit via an exponential time algorithm. The only known algorithm for list-decodable covariance estimation (with total variation error guarantees) is due to [IK22]. This algorithm essentially relies on the Sum-of-Squares method and has (sample and computational) complexity $d^{\mathrm{poly}(1/\alpha)}$. Intriguingly, there is compelling evidence that this complexity bound cannot be improved. Specifically, [DKS17; DKPPS21] showed that any Statistical Query (SQ) algorithm for this task requires complexity $d^{\mathrm{poly}(1/\alpha)}$. Combined with the reduction of [BBHLT20], this implies a similar lower bound for low-degree polynomial tests. These lower bounds suggest an intrinsic information-computation gap for the problem.

The aforementioned lower bound results [DKS17; DKPPS21] establish that it is SQ (and low-degree) hard to distinguish between a standard multivariate Gaussian, $\mathcal{N}(0,I)$, and a distribution $P$ that behaves like $\mathcal{N}(0,I)$ (in terms of low-degree moments) but $P$ contains an $\alpha$-fraction of samples from a Gaussian $\mathcal{N}(0,\Sigma)$, where $\Sigma$ is *very thin* in some hidden direction $v$. Since the distribution $\mathcal{N}(0,\Sigma)$ is very thin along $v$, i.e., has very small variance, the obvious choice of $\mathcal{N}(0,I)$ essentially has no overlap with $\mathcal{N}(0,\Sigma)$ — making this kind of strong list-decoding guarantee (closeness in total variation distance) likely computationally intractable.

Interestingly, there are two possible ways that a pair of mean zero Gaussians can be separated [DHKK20; BDHKKK20]: (1) one could be much thinner than the other in some direction, or (2) they could have many orthogonal directions in which their variances differ, adding up to something more substantial. While the lower bounds of [DKS17; DKPPS21] seem to rule out being able to detect deviations of the former type in fully polynomial time (i.e., $\mathrm{poly}(d/\alpha)$), it does not rule out efficiently detecting deviations of the latter. In particular, we could hope to find (in $\mathrm{poly}(d/\alpha)$ time) a good hypothesis $\widetilde{\Sigma}$ such that $\|\widetilde{\Sigma}^{-1/2}\Sigma\widetilde{\Sigma}^{-1/2} - I\|_{\mathrm{F}}$ is not too big. While this does not exclude the possibility that $\Sigma$ is much thinner than $\widetilde{\Sigma}$ in a small number of independent directions, it does rule out the second kind of difference between the two. The main goal of this paper is to provide an elementary (relying only on spectral techniques) list-decoding algorithm of this form. Given corrupted samples from a Gaussian $\mathcal{N}(0,\Sigma)$, we give a $\mathrm{poly}(d/\alpha)$-time algorithm that returns a small list of hypotheses $\widetilde{\Sigma}$ such that for at least one of them we have that $\|\widetilde{\Sigma}^{-1/2}\Sigma\widetilde{\Sigma}^{-1/2} - I\|_{\mathrm{F}} < \mathrm{poly}(1/\alpha)$.

In addition to providing the best qualitative guarantee we could hope to achieve in fully polynomial time, the above kind of "weak" list-decoding algorithm has interesting implications for the well-studied problem of robustly learning Gaussian mixture models (GMMs). [BDJKKV22] gave a polynomial-time algorithm to robustly learn arbitrary mixtures of Gaussians (with a constant number

of components). One of the two main ingredients of their approach is a subroutine that could perform partial clustering of points into components that satisfy exactly this kind of weak closeness guarantee. [BDJKKV22] developed such a subroutine by making essential use of a Sum-of-Squares (SoS) relaxation, for the setting that the samples come from a mildly corrupted mixture of Gaussians (with a small constant fraction of outliers). As a corollary of our techniques, we obtain an elementary spectral algorithm for this partial clustering task (and, as already stated, our results work in the more general list-decoding setting). This yields the first SoS-free algorithm for robustly learning GMMs, answering an open problem in the literature [Vem22; Kot21].

## 1.1 Our Results

Our main result is a polynomial-time algorithm for list-decoding the covariance of a Gaussian in relative Frobenius norm, under adversarial corruption where more than half the samples could be outliers. Definition 1.1 makes precise the corruption model, and Theorem 1.2 states the guarantees of our main algorithm (Algorithm 1).

**Definition 1.1** (Corruption model for list-decoding). *Let the parameters $\epsilon, \alpha \in (0, 1/2)$ and a distribution family $\mathcal{D}$. The statistician specifies the number of samples $m$. Then a set of $n \geq \alpha m$ i.i.d. points are sampled from an unknown $D \in \mathcal{D}$. We call these $n$ samples the* inliers*. Upon inspecting the $n$ inliers, a (malicious and computationally unbounded) adversary can replace an arbitrary $\ell \leq \epsilon n$ of the inliers with arbitrary points, and further add $m - n$ arbitrary points to the dataset, before returning the entire set of $m$ points to the statistician. The parameter $\alpha$ is also known a-priori to the statistician, but the number $n$ chosen by the adversary is unknown. We refer to this set of $m$ points as an $(\alpha, \epsilon)$-corrupted set of samples from $D$.*

In our context, the notion of list-decoding is as follows: our algorithm will return a polynomially-sized list of matrices $H_i$, such that at least one $H_i$ is an "approximate square root" of the true covariance $\Sigma$ having bounded *dimension-independent* error $\|H_i^{-1/2} \Sigma H_i^{-1/2} - I\|_F$. As discussed earlier, the bound we guarantee is $\text{poly}(1/\alpha)$, which is in general larger than 1 and thus does not lead to non-trivial total variation bounds, thus circumventing related SQ lower bounds. Our main theorem is:

**Theorem 1.2** (List-Decodable Covariance Estimation in Relative Frobenius Norm). *Let $C' > 0$ be a sufficiently large constant and $\epsilon_0 > 0$ be a sufficiently small constant. Let the parameters $\alpha \in (0, 1/2)$, $\epsilon \in (0, \epsilon_0)$, and failure probability $\delta \in (0, 1/2)$ be known to the algorithm. Let $D$ be the Gaussian distribution $\mathcal{N}(\mu, \Sigma)$ with mean $\mu \in \mathbb{R}^d$ and full-rank covariance $\Sigma \in \mathbb{R}^{d \times d}$. There is an $\tilde{O}(m^2 d^2)$-time algorithm (Algorithm 1) such that, on input $\alpha, \delta$ and an $(\alpha, \epsilon)$-corrupted set of $m$ points from $D$ (Definition 1.1) for any $m > C' \frac{d^2 \log^5(d/\alpha\delta)}{\alpha^6}$, with probability at least $1 - \delta$, the algorithm returns a list of at most $O(1/\alpha)$ many sets $T_i$ which are disjoint subsets of samples, each of size at least $0.5\alpha m$, and there exists a $T_i$ in the output list such that:*

- *Recall the notation in the corruption model (Definition 1.1) where $n$ is the size of the original inlier set $S$ and $\ell$ is the number of points in $S$ that the adversary replaced—$n$ and $\ell$ are unknown to the algorithm except that $n \geq \alpha m$ and $\ell \leq \epsilon n$. The set $T_i$ in the returned list satisfies that $|T_i \cap S| \geq (1 - 0.01\alpha)(n - \ell)$.*

- *Denote $H_i := \mathbf{E}_{X \sim T_i}[XX^\top]$. The matrix $H_i$ satisfies $\|H_i^{-1/2} \Sigma H_i^{-1/2} - I\|_F \lesssim \frac{1}{\alpha^4} \log(\frac{1}{\alpha})$.*

Algorithm 1 is an iterative spectral algorithm, as opposed to involving large convex programs based on the sum-of-squares hierarchy. We state Algorithm 1 and give a high-level proof sketch of why it satisfies Theorem 1.2 in Section 3. The formal proof of Theorem 1.2 appears in Appendix F (where we allow $\Sigma$ to be rank-deficient as well).

We also briefly remark that, if we wish to list-decode pairs of (mean, covariance), then we can run the following straightforward augmentation to Algorithm 1: after running Algorithm 1, for each $\hat{\Sigma}$ in the output list, whiten the data using $\hat{\Sigma}$ and run a standard list-decoding algorithm for the mean [CSV17; DKK20b]. These algorithms work if the data is distributed with a bounded covariance: by the guarantees of Theorem 1.2, after whitening, the data has covariance bounded by $\text{poly}(1/\alpha) \cdot I$.

As a corollary of our main result, the same algorithm (but with a slightly higher sample complexity) also achieves outlier-robust list-decoding of the covariances for the components of a Gaussian mixture model, in relative Frobenius norm. Definition 1.3 and Theorem 1.4 state the corresponding corruption model and theoretical guarantees on Algorithm 1.

**Definition 1.3** (Corruption model for samples from Gaussian Mixtures). *Let $\epsilon \in (0, 1/2)$. Consider a Gaussian mixture model $\sum_p \alpha_p \mathcal{N}(\mu_p, \Sigma_p)$, where the parameters $\alpha_p, \mu_p$ and $\Sigma_p$ are unknown and satisfy $\alpha_p \geq \alpha$ for some known parameter $\alpha$. The statistician specifies the number of samples $m$, and $m$ i.i.d. samples are drawn from the Gaussian mixture, which are called the* inliers. *The malicious and computationally unbounded adversary then inspects the $m$ inliers and is allowed to replace an arbitrary subset of $\epsilon \alpha m$ many inlier points with arbitrary outlier points, before giving the modified dataset to the statistician. We call this the $\epsilon$-corrupted set of samples.*

**Theorem 1.4** (Outlier-Robust Clustering and Estimation of Covariances for GMM). *Let $C' > 0$ be a sufficiently large constant and $\epsilon_0 > 0$ be a sufficiently small constant. Let the parameters $\alpha \in (0, 1/2)$, $\epsilon \in (0, \epsilon_0)$, and failure probability $\delta \in (0, 1/2)$ be known. There is an $\tilde{O}(m^2 d^2)$-time algorithm such that, on input $\alpha, \delta$, and $m > C' \frac{d^2 \log^5(d/\alpha\delta)}{\alpha^6}$ many $\epsilon$-corrupted samples from an unknown $k$-component Gaussian mixture $\sum_{p=1}^k \alpha_p \mathcal{N}(\mu_p, \Sigma_p)$ over $\mathbb{R}^d$ as in Definition 1.3, where all $\Sigma_p$'s are full-rank and all $\alpha_p$ satisfies $\alpha_p \geq \alpha$ and $k \leq \frac{1}{\alpha}$ is unknown to the algorithm, with probability at least $1 - \delta$ over the corrupted samples and the randomness of the algorithm, the algorithm returns a list of at most $k$ many disjoint subsets of samples $\{T_i\}$ such that:*

- *For the $p^{th}$ Gaussian component, denote the set $S_p$ as the samples in the inlier set $S$ that were drawn from component $p$. Let $n_p$ be the size of $S_p$, and let $\ell_p$ be the number of points in $S_p$ that the adversary replaced—$n_p$ and $\ell_p$ are both unknown to the algorithm except that $\mathbf{E}[n_p] = \alpha_p m \geq \alpha m$ for each $p$ and $\sum_p \ell_p \leq \epsilon \alpha m$. Then, for every Gaussian component $p$ in the mixture, there exists a set $T_{i_p}$ in the returned list such that $|T_{i_p} \cap S_p| \geq (1 - 0.01\alpha)(n_p - \ell_p)$.*

- *For every component $p$, there is a set of samples $T_{i_p}$ in the returned list such that, defining $H_{i_p} = \mathbf{E}_{X \sim T_{i_p}}[XX^\top]$, we have $H_{i_p}$ satisfying $\|H_{i_p}^{-1/2} \Sigma_p H_{i_p}^{-1/2} - I\|_{\mathrm{F}} \lesssim (1/\alpha^4) \log(1/\alpha)$.*

- *Let $\Sigma$ be the (population-level) covariance matrix of the Gaussian mixture. For any two components $p \neq p'$ with $\|\Sigma^{-1/2}(\Sigma_p - \Sigma_{p'})\Sigma^{-1/2}\|_{\mathrm{F}} > C(1/\alpha)^5 \log(1/\alpha)$ for a sufficiently large constant $C$, the sets $T_{i_p}$ and $T_{i_{p'}}$ from the previous bullet are guaranteed to be different.*

The theorem states that, not only does Algorithm 1 achieve list-decoding of the Gaussian component covariances, but it also clusters samples according to separation of covariances in relative Frobenius norm. The recent result of [BDJKKV22] on robustly learning GMMs also involves an algorithmic component for clustering. Their approach is based on the sum-of-squares hierarchy (and thus requires solving large convex programs) while, Algorithm 1 is a purely spectral algorithm.

We also emphasize that the list size returned by Algorithm 1, in the case of a $k$-component Gaussian mixture model, is at most $k$ — instead of a weaker result such as $O(1/\alpha)$ or polynomial/exponentially large in $1/\alpha$. This is possible because Algorithm 1 keeps careful track of samples and makes sure that no more than a $0.01\alpha$-fraction of samples is removed from each component or mis-clustered into another component. We prove Theorem 1.4 in Appendix G.

## 1.2 Overview of Techniques

At a high level, our approach involves integrating robust covariance estimation techniques from [DKKLMS16] with the multifilter for list-decoding of [DKS18]. For a set $S$, we use $SS^\top$ to denote the set $\{xx^\top : x \in S\}$. [DKKLMS16] states that in order to estimate the true covariance $\Sigma$, it suffices to find a large subset $S$ of points with large overlap with the original set of good points, so that if $\Sigma' = \mathbf{Cov}(S)$ then $\mathbf{Cov}((\Sigma')^{-1/2} SS^\top (\Sigma')^{-1/2})$ has no large eigenvalues. This will imply that $\|(\Sigma')^{-1/2}(\Sigma - \Sigma')(\Sigma')^{-1/2}\|_{\mathrm{F}}$ is not too large (Lemma E.2).

As in [DKKLMS16], our basic approach for finding such a set $S$ is by the iteratively repeating the following procedure: We begin by taking $S$ to be the set of all samples and repeatedly check whether or not $\mathbf{Cov}((\Sigma')^{-1/2} SS^\top (\Sigma')^{-1/2})$ has any large eigenvalues. If it does not, we are done. If it has a large eigenvalue corresponding to a matrix $A$ (normalized to have unit Frobenius norm), we consider the values of $f(x) = \langle A, (\Sigma')^{-1/2} xx^\top (\Sigma')^{-1/2} \rangle$, for $x$ in $S$, and attempt to use them to find outliers. Unfortunately, as the inliers might comprise a minority of the sample, the values we get out of this formula might end-up in several reasonably large clusters, any one of which could plausibly contain the true samples; thus, not allowing us to declare any particular points to be outliers

with any degree of certainty. We resolve this issue by using the multifilter approach of [DKS18]—we either (i) iteratively remove outliers, or (ii) partition the data into clusters and recurse on each cluster. In particular, we note that if there is large variance in the $A$-direction, one of two things must happen, either: (i) The substantial majority of the values of $f(x)$ lie in a single cluster, with some extreme outliers. In this case, we can be confident that the extreme outliers are actual errors and remove them (Section 3.2). (ii) There are at least two clusters of values of $f(x)$ that are far apart from each other. In this case, instead of simply removing obvious outliers, we replace $S$ by two subsets $S_1$ and $S_2$ with the guarantee that at least one of the $S_i$ contains almost all of the inliers. Naïvely, this can be done by finding a value $y$ between the two clusters so that very few samples have $f(x)$ close to $y$, and letting $S_1$ be the set of points with $f(x) < y$ and $S_2$ the set of points with $f(x) > y$ (Section 3.3). In either case, we will have cleaned up our set of samples and can recurse on each of the returned subsets of $S$. Iterating this technique recursively on all of the smaller subsets returned ensures that there is always at least one subset containing the majority of the inliers, and that eventually once it stops having too large of a covariance, we will return an appropriate approximation to $\Sigma$.

We want to highlight the main point of difference where our techniques differ notably from [DKS18]. In order to implement the algorithm outlined above, one needs to have good a priori bounds for what the variance of $f(x)$ over the inliers ought to be. Since $f(\cdot)$ is a quadratic polynomial, the variance of $f$ over the inliers, itself depends on the covariance $\Sigma$, which is exactly what we are trying to estimate. This challenge of circular dependence does not appear in [DKS18]: their goal was to estimate the unknown mean of an identity-covariance Gaussian, and thus it sufficed to use a linear polynomial $f$ (instead of a quadratic polynomial). Importantly, the covariance of a linear polynomial does not depend on the (unknown) mean (it depends only on the covariance, which was known in their setting). In order to overcome this challenge, we observe that if $S$ contains most of the inliers, then the covariance of $S$ cannot be too much smaller than the true covariance $\Sigma$. This allows us to find an upper bound on $\Sigma$, which in turn lets us upper bound the variance of $f(x)$ over the good samples (Lemma 3.2).

**Related Work**    We refer the reader to [DK23] for an overview of algorithmic robust statistics. We mention the most relevant related work here and discuss additional related work in Appendix A. Algorithms for list-decodable covariance estimation were developed in the special cases of subspace estimation and linear regression in [KKK19; BK21; RY20b; RY20a; DJKS22]. On the other hand, [DKS17; DKPPS21] present SQ lower bounds for learning Gaussian mixture models and list-decodable linear regression (and thus list-decodable covariance estimation), respectively.

[IK22] gave the first algorithm for general list-decodable covariance estimation that achieves non-trivial bounds in total variation distance using the powerful sum-of-squares hierarchy. Their algorithm outputs an $\exp(\mathrm{poly}(1/\alpha))$-sized list of matrices containing an $H_i$ that is close to the true $\Sigma$ in two metrics (i) relative Frobenius norm: $\|H_i^{-1/2}\Sigma H_i^{-1/2} - I\|_{\mathrm{F}} = \mathrm{poly}(1/\alpha)$ *and* (ii) multiplicative spectral approximation: $\mathrm{poly}(\alpha)\Sigma \preceq H_i \preceq \mathrm{poly}(1/\alpha)\Sigma$. Their algorithm uses $d^{\mathrm{poly}(1/\alpha)}$ samples, which seems to be necessary for efficient (statistical query) algorithms achieving multiplicative spectral approximation [DKS17; DKPPS21]. In comparison, Theorem 1.2 uses only $\mathrm{poly}(d/\alpha)$ samples, returns a list of $O(1/\alpha)$ matrices, but approximates only in the relative Frobenius norm: $\|H_i^{-1/2}\Sigma H_i^{-1/2} - I\|_{\mathrm{F}} = \mathrm{poly}(1/\alpha)$.

## 2    Preliminaries

For a vector $v$, we let $\|v\|_2$ denote its $\ell_2$-norm. We use $I_d$ to denote the $d \times d$ identity matrix; We will drop the subscript when it is clear from the context. For a matrix $A$, we use $\|A\|_{\mathrm{F}}$ and $\|A\|_{\mathrm{op}}$ to denote the Frobenius and spectral (or operator) norms, respectively. We denote by $\langle v, u \rangle$, the standard inner product between the vectors $u, v$. For matrices $U, V \in \mathbb{R}^{d \times d}$, we use $\langle U, V \rangle$ to denote the trace inner product $\sum_{ij} U_{ij}V_{ij}$. For a matrix $A \in \mathbb{R}^{d \times d}$, we use $A^{\flat}$ to denote the flattened vector in $\mathbb{R}^{d^2}$, and for a $v \in \mathbb{R}^{d^2}$, we use $v^{\sharp}$ to denote the unique matrix $A$ such that $A^{\flat} = v^{\sharp}$. For a matrix $A$, we let $A^{\dagger}$ denote its pseudo-inverse. We use $\otimes$ to denote the Kronecker product. For a matrix $A$, we use $\ker(A)$ for the null space of $A$. We use $X \sim D$ to denote that a random variable $X$ is distributed according to the distribution $D$. We use $\mathcal{N}(\mu, \Sigma)$ for the Gaussian distribution with mean $\mu$ and covariance matrix $\Sigma$. For a set $S$, we use $X \sim S$ to denote that $X$ is distributed uniformly at random from $S$. We use $a \lesssim b$ to denote that there exists an absolute universal constant $C > 0$ (independent of the variables or parameters on which $a$ and $b$ depend) such that $a \leq Cb$.

## 2.1 Deterministic Conditions

Our algorithm will rely on the uncorrupted inliers satisfying a set of properties, similar to the "stability conditions" from [DKKLMS16]. Intuitively, these are are concentration properties for sets of samples, but with the added requirement that every large subset of the samples also satisfies these properties.

**Definition 2.1** $((\eta, \epsilon)$-Stable Set). *Let $D$ be a distribution with mean $\mu$ and covariance $\Sigma$. We say a set of points $A \subset \mathbb{R}^d$ is $(\eta, \epsilon)$-stable with respect to $D$, if for any subset $A' \subseteq A$ with $|A'| \geq (1 - \epsilon)|A|$, the following hold: for every $v \in \mathbb{R}^d$, symmetric $U \in \mathbb{R}^{d \times d}$, and every even degree-$2$ polynomial $p$:*

*(L.1)* $\left| \frac{1}{|A'|} \sum_{x \in A'} v^\top (x - \mu) \right| \leq 0.1 \sqrt{(v^\top \Sigma v)}$ .

*(L.2)* $\left| \left\langle \frac{1}{|A'|} \sum_{x \in A'} (x - \mu)(x - \mu)^\top - \Sigma, U \right\rangle \right| \leq 0.1 \left\| \Sigma^{1/2} U \Sigma^{1/2} \right\|_F$ .

*(L.3)* $\Pr_{X \sim A'} \left[ \left| p(X) - \mathbf{E}_{X \sim D}[p(X)] \right| > 10 \sqrt{\mathbf{Var}_{Y \sim D}[p(Y)]} \ln \left( \frac{2}{\eta} \right) \right] \leq \eta.$

*(L.4)* $\mathbf{Var}_{X \sim A'}[p(X)] \leq 4 \mathbf{Var}_{X \sim D}[p(X)].$

*(L.5)* *The null space of second moment matrix of $A'$ is contained in the null space of $\Sigma$, i.e.,* $\ker \left( \sum_{x \in A'} x x^\top \right) \subseteq \ker(\Sigma).$

A Gaussian dataset is "stable" with high probability [DKKLMS16]; formally, we have Lemma 2.2, proved in Appendix D.1. Moreover, Lemma 2.2 can be extended to a variety of distributions (cf. Remark D.1).

**Lemma 2.2** (Deterministic Conditions Hold with High Probability). *For a sufficiently small positive constant $\epsilon_0$ and a sufficiently large absolute constant $C$, a set of $m > C d^2 \log^5(d/(\eta\delta))/\eta^2$ samples from $\mathcal{N}(\mu, \Sigma)$, with probability $1 - \delta$, is $(\eta, \epsilon)$-stable set with respect to $\mu, \Sigma$ for all $\epsilon \leq \epsilon_0$.*

# 3 Analysis of a Single Recursive Call of the Algorithm

Our algorithm, Algorithm 1, filters and splits samples into multiple sets recursively, until we can certify that the empirical second moment matrix of the "current data set" is suitable to be included in the returned list of covariances. As a reference point, we define the notations and assumptions necessary to analyze each recursive call below. However, before moving to the formal analysis we will first give an informal overview of the algorithm's steps and the high-level ideas behind them.

---

**Assumption 3.1** (Assumptions and notations for a single recursive call of the algorithm).

- $S = \{x_i\}_{i=1}^n$ is a set of $n$ uncontaminated samples, which is assumed to be $(\eta, 2\epsilon_0)$-stable with respect to the inlier distribution $D$ having mean and covariance $\mu, \Sigma$ (c.f. Definition 2.1). We assume $\eta \leq 0.001$, $\epsilon_0 = 0.01$, and $\mathbf{Var}_{X \sim D}[X^\top A X] \leq C_1(\|\Sigma^{1/2} A \Sigma^{1/2}\|_F^2 + \|\Sigma^{1/2} A \mu\|_2^2)$ for all symmetric $d \times d$ matrices $A$ and a constant $C_1$.

- $T$ is the input set to the current recursive call of the algorithm (after the adversarial corruptions), which satisfies $|S \cap T| \geq (1 - 2\epsilon_0)|S|$ and $|T| \leq (1/\alpha)|S|$.

- We denote $H = \mathbf{E}_{X \sim T} \left[ X X^\top \right]$.

- We denote by $\widetilde{S}, \widetilde{T}$ the versions of $S$ and $T$ normalized by $H^{\dagger/2}$: $\widetilde{S} = \{H^{\dagger/2} x : x \in S\}$ and $\widetilde{T} = \{H^{\dagger/2} x : x \in T\}$. We use the notation $\widetilde{x}$ for elements in $\widetilde{S}$ and $\widetilde{T}$, and $x$ for elements in $S$ and $T$. Similarly, we use the notation $\widetilde{X}$ for random variables with support in $\widetilde{S}$ or $\widetilde{T}$.

- The mean and covariance of the inlier distribution $D$ after transformation with $H^{\dagger/2}$ are denoted by $\widetilde{\mu} := H^{\dagger/2} \mu$, $\widetilde{\Sigma} := H^{\dagger/2} \Sigma H^{\dagger/2}$. We denote the empirical mean and covariance of the transformed inliers in $T$ by $\hat{\mu} := \mathbf{E}_{\widetilde{X} \sim \widetilde{S} \cap \widetilde{T}}[\widetilde{X}]$ and $\hat{\Sigma} := \mathbf{Cov}_{\widetilde{X} \sim \widetilde{S} \cap \widetilde{T}}[\widetilde{X}]$.

---

Much of the algorithm uses the fact that, for a Gaussian, even quadratic polynomials have a small variance. We will leverage this for filtering and clustering samples. See Appendix E for the proof.

**Lemma 3.2.** *Make Assumption 3.1 and recall that* $H = \mathbf{E}_{X \sim T}[XX^\top]$, *where* $T$ *is the corrupted version of a* stable *inlier set* $S$. *For every symmetric matrix* $A$ *with* $\|A\|_F = 1$, *we have that* $\mathbf{Var}_{X \sim D}[(H^{\dagger/2}X)^\top A(H^{\dagger/2}X)] \leq 18C_1/\alpha^2$.

Armed with Lemma 3.2, we can now give a high-level overview of a recursive call of Algorithm 1:

1. In our notation, we call the current data set $T$. Denoting $H = \mathbf{E}_{X \sim T}[XX^\top]$ for its the empirical second moment matrix, we construct the normalized data set $\widetilde{T} = \{H^{-1/2}x \; : \; x \in T\}$. The normalization allows us to bound the covariance $\Sigma$ in terms of $H$.
2. Since we are trying to estimate a covariance, consider the vectors $\tilde{s} = \{(\tilde{x}\tilde{x}^\top)^\flat : \tilde{x} \in \widetilde{T}\}$, which are the second moment matrices of each data point flattened into vectors.
3. The first step is standard in filtering-based outlier-robust estimation: we test whether the covariance of the $\tilde{s}$ vectors is small. If so, we are able to prove that the current $H$ is a good approximate square root of $\Sigma$ (c.f. Section 3.1) hence we just return $H$.
4. If the first test fails, that would imply that the empirical covariance of the $\tilde{s}$ vectors is large in some direction. We want to leverage this direction to make progress, either by removing outliers through filtering or by bi-partitioning our samples into two clear clusters.
5. To decide between the 2 options, consider projecting the $\tilde{s}$ vectors onto their largest variance direction. Specifically, let $A$ be the matrix lifted from the largest eigenvector of the covariance of the $\tilde{s}$ vectors. Define the vectors $\tilde{y} = \tilde{x}^\top A\tilde{x} = \langle \tilde{x}\tilde{x}^\top, A \rangle$ for $\tilde{x} \in \widetilde{T}$, corresponding to the 1-dimensional projection of $\tilde{s}$ onto the $A^\flat$ direction. Since we have failed the first test, these $\tilde{y}$ elements must have a large variance. We will decide to filter or divide our samples, based on whether the $\alpha m$-smallest and $\alpha m$-largest elements of the $\tilde{y}$s are close to each other.
6. If they are close, yet we have large variance, we will use this information to design a *score function* and perform filtering that removes a random sample with probability proportional to its score. We will then go back to Step 1. This would work because by Lemma 3.2 and by stability (Definition 2.1), the (unfiltered) inliers have a small empirical variance within themselves, meaning that the large total empirical variance is mostly due to the outlier.

    Ideally, the score of a sample would be (proportional to) the squared distance between the sample and the mean of the inliers—the total inlier score would then be equal to the inlier variance. However, since we do not know which points are the inliers, we instead use the median of all the projected samples as a proxy for the unknown inlier mean. We show that the distance between the $\alpha m$-smallest and largest $\tilde{y}$s bounds the difference between the ideal and proxy scores.
7. Otherwise, $\alpha m$-smallest and $\alpha m$-largest elements of the $\tilde{y}$s are far apart. By the stability condition Definition 2.1 (specifically, Condition (L.3)), most of the inliers must be close to each other under this 1-dimensional projection. Therefore, the large quantile range necessarily means there is a threshold under this projection to divide the samples into two sets, such that each set has at least $\alpha m$ points and most of the inliers are kept within a single set.

The score function mentioned in Step 6 upper bounds the maximum variance we check in Step 3. For simplicity, in the actual algorithm (Algorithm 1) we use the score directly for the termination check instead of checking the covariance, but it does not matter technically which quantity we use.

**Remark 3.3** (Runtime of Algorithm 1). We claim that each "loop" in Algorithm 1 takes $\tilde{O}(md^2)$ time to compute. The number of times we run the "loop" is at most $O(m)$, since each loop either ends in termination, removes 1 element from the dataset, or splits the dataset, all of which can happen at most $O(m)$ times. From this, we can conclude a runtime of $\tilde{O}(m^2d^2)$. The sample complexity of our algorithm is also explicitly calculable to be $\tilde{O}(d^2/\alpha^6)$, which follows from Lemma 2.2 and the choice of parameter $\eta = \Theta(\alpha^3)$ from Theorem F.1 (the formal version of Theorem 1.2).

To see the runtime of a single loop: the most expensive operations in each loop are to compute $H_t$, its pseudo-inverse, and to compute the symmetric matrix $A$ in Line 7 that is the top eigenvector of a $d^2 \times d^2$ matrix. Computing $H_t$ trivially takes $O(md^2)$ time, resulting in a $d \times d$ matrix. Its pseudoinverse can be computed in $O(d^\omega)$ time, which is dominated by $O(md^2)$ since $m \gtrsim d$. Lastly, we observe that, instead of actually computing the top eigenvector in Line 7 to yield the matrix $A$, it suffices in our analysis to compute a matrix $B$ whose Rayleigh quotient $(B^\flat)^\top \left( \mathbf{Cov}_{\widetilde{X} \sim \widetilde{T}_t}[\widetilde{X}^{\otimes 2}] \right) B^\flat / ((B^\flat)^\top B^\flat)$ is at least $\frac{1}{2}$ times $(A^\flat)^\top \left( \mathbf{Cov}_{\widetilde{X} \sim \widetilde{T}_t}[\widetilde{X}^{\otimes 2}] \right) A^\flat / ((A^\flat)^\top A^\flat)$. We can do this via $O(\log d)$ many

---

**Algorithm 1** List-Decodable Covariance Estimation in Relative Frobenius Norm

---

1: **Constants**: $m, \alpha, \eta, R := C(1/\alpha^2)\log(1/(\epsilon_0\alpha))$ for $C > 6000\sqrt{C_1}$, $C' > 720/\epsilon_0$ (where $C_1, \epsilon_0$ are defined in Assumption 3.1).
2: **function** COVLISTDECODING($T_0$)
3:     $t \leftarrow 0$.
4:     **loop**
5:         Compute $H_t = \mathbf{E}_{X \sim T_t}[XX^\top]$.
6:         Let $\widetilde{T}_t = \{H_t^{\dagger/2}x \ : \ x \in T_t\}$ be the transformed set of samples.
7:         Let $A$ be the symmetric matrix corresponding to the top eigenvector of $\mathbf{Cov}_{\widetilde{X} \sim \widetilde{T}_t}[\widetilde{X}^{\otimes 2}]$.
8:         Normalize $A$ so that $\|A\|_{\mathrm{F}} = 1$.
9:         Compute the set $\widetilde{Y}_t = \{\widetilde{x}^\top A \widetilde{x} \ : \ \widetilde{x} \in \widetilde{T}_t\}$
10:        Compute the $\alpha m/9$-th smallest element $q_{\mathrm{left}}$ as well as the $\alpha m/9$-th largest element $q_{\mathrm{right}}$, as well as the median $\widetilde{y}_{\mathrm{median}}$ of $\widetilde{Y}_t$.
11:        Define the function $f(\widetilde{x}) = (\widetilde{x}^\top A \widetilde{x} - \widetilde{y}_{\mathrm{median}})^2$.
12:        **if** $\mathbf{E}_{\widetilde{X} \sim \widetilde{T}_t}[f(\widetilde{X})] \leq C'R^2/\alpha^3$ **then**                    ▷ c.f. Lemma 3.4
13:            **If** $|T_t| \geq 0.5\alpha m$ **then return** $\{T_t\}$ **else return** the empty list.
14:        **else if** $q_{\mathrm{right}} - q_{\mathrm{left}} \leq R$ **then**                    ▷ c.f. Lemma 3.5
15:            Let the probability mass function $p(\widetilde{x}) := f(\widetilde{x})/\sum_{\widetilde{x} \in \widetilde{T}_t} f(\widetilde{x})$.
16:            Pick $x_{\mathrm{removed}} \in T_t$ according to $p(H_t^{\dagger/2}x)$.
17:            $T_{t+1} \leftarrow T_t \setminus \{x_{\mathrm{removed}}\}$.
18:        **else**
19:            $\tau \leftarrow \mathrm{FindDivider}(\widetilde{Y}_t, \alpha m/9)$.                    ▷ c.f. Lemma 3.6
20:            $T' \leftarrow \{H_t^{1/2}\widetilde{x} : \widetilde{x} \in \widetilde{T}_t, \widetilde{x}^\top A \widetilde{x} \leq \tau\}$, $T'' \leftarrow \{H_t^{1/2}\widetilde{x} : \widetilde{x} \in T_t, \widetilde{x}^\top A \widetilde{x} > \tau\}$.
21:            $L_1 \leftarrow \mathrm{COVLISTDECODING}(T'), L_2 \leftarrow \mathrm{COVLISTDECODING}(T'')$.
22:            **return** $L_1 \cup L_2$.
23:        $t \leftarrow t + 1$.

---

power iterations. Since

$$\mathbf{Cov}_{\widetilde{X} \sim \widetilde{T}_t}[\widetilde{X}^{\otimes 2}] = \frac{1}{|\widetilde{T}_t|}\sum_{z \in \widetilde{T}_t} zz^\top - \left(\frac{1}{|\widetilde{T}_t|}\sum_{z \in \widetilde{T}_t} z\right)^\top \left(\frac{1}{|\widetilde{T}_t|}\sum_{z \in \widetilde{T}_t} z\right),$$

we can compute each matrix-vector product in $O(|T_t|d^2) \leq O(md^2)$ time, thus yielding an $\tilde{O}(m^2 d^2)$ runtime for the power iteration.

### 3.1   Certificate Lemma: Bounded Fourth Moment Implies Closeness

The first component of the analysis is our certificate lemma, which states that, if the empirical covariance of the (flattened) second moment matrices of current data set (after normalization) is bounded, then the empirical second moment matrix $H$ of the current data set is a good approximation to the covariance of the Gaussian component we want to estimate.

**Lemma 3.4** (Case when we stop and return). *Make Assumption 3.1. Let $w \in \mathbb{R}^{d^2}$ be the leading eigenvector of the $\mathbf{Cov}_{\widetilde{X} \sim \widetilde{T}}[\widetilde{X}^{\otimes 2}]$ with $\|w\|_2 = 1$, and let $A \in \mathbb{R}^{d \times d}$ be $w^\sharp$. Note that $\|w\|_2 = 1$ implies $\|A\|_{\mathrm{F}} = 1$. Then, we have $\left\|H^{\dagger/2}\Sigma H^{\dagger/2} - I\right\|_{\mathrm{F}}^2 \lesssim (1/\alpha)\mathbf{Var}_{\widetilde{X} \sim \widetilde{T}}[\widetilde{X}^\top A \widetilde{X}] + 1/\alpha^2$.*

See Appendix E.2 for the proof. Our termination check of Line 12 uses the score $f(\widetilde{x}) = (\widetilde{x}^\top A \widetilde{x} - \widetilde{y}_{\mathrm{median}})^2$ where $\widetilde{y}_{\mathrm{median}}$ is the median of $\{\widetilde{x}^\top A \widetilde{x} \ : \ \widetilde{x} \in \widetilde{T}\}$. Since $\mathbf{Var}_{\widetilde{X} \sim \widetilde{T}}[\widetilde{X}^\top A \widetilde{X}] \leq \mathbf{E}_{\widetilde{X} \sim \widetilde{T}}[f(\widetilde{X})]$ our check ensures that $\mathbf{Var}_{\widetilde{X} \sim \widetilde{T}}[\widetilde{X}^\top A \widetilde{X}] \leq \mathrm{poly}(1/\alpha)$ before returning.

### 3.2   Filtering: Removing Extreme Outliers

As discussed in the algorithm outline, if the termination check fails, namely the expected score over the entire set of $\widetilde{T}$ is large, then we proceed to either filter or bi-partition our samples. This subsection

states the guarantees of the filtering procedure, which assumes that the $\alpha m$-smallest and largest elements in the set $\{\widetilde{x}^\top A \widetilde{x} \;:\; \widetilde{x} \in \widetilde{T}\}$ have distance at most $R$ for some $R = \mathrm{poly}(1/\alpha)$.

Recall from Lemma 3.2 that $\mathbf{Var}_{\widetilde{X} \sim H^{\dagger/2}D}[\widetilde{X}^\top A \widetilde{X}] = \mathbf{E}_{\widetilde{X} \sim H^{\dagger/2}D}[(\widetilde{X}^\top A \widetilde{X} - \mathbf{E}_{\widetilde{X} \sim H^{\dagger/2}D}[\widetilde{X}^\top A \widetilde{X}])^2]$ is bounded by $O(1/\alpha^2)$. By stability, the same is true for $\mathbf{E}_{\widetilde{X} \sim \widetilde{S} \cap \widetilde{T}}[(\widetilde{X}^\top A \widetilde{X} - \mathbf{E}_{\widetilde{X} \sim \widetilde{S} \cap \widetilde{T}}[\widetilde{X}^\top A \widetilde{X}])^2]$. Notice that this looks almost like our score function $f(\widetilde{X})$, except that in $f(\widetilde{X})$ we use $\widetilde{y}_{\mathrm{median}}$ for centering instead of $\mathbf{E}_{\widetilde{X} \sim \widetilde{S} \cap \widetilde{T}}[\widetilde{X}^\top A \widetilde{X}]$, since the latter quantity is by definition unknown to the algorithm. In Lemma E.3, we show that the two quantities have distance upper bounded by $O(R)$, where $R$ is the quantile distance in our assumption, which in turn implies that the inliers in $\widetilde{S} \cap \widetilde{T}$ contribute very little to $\mathbf{E}_{\widetilde{X} \sim \widetilde{T}}[f(\widetilde{X})]$. Given that $\mathbf{E}_{\widetilde{X} \sim \widetilde{T}}[f(\widetilde{X})]$ is large, by virtue of having failed the termination check, we can then conclude that most of the score contribution comes from the outliers. Thus, we can safely use the score to randomly pick an element in the dataset for removal, with probability proportional to its score, and the element will be overwhelmingly more likely to be an outlier rather than an inlier. Lemma 3.5 below states the precise guarantees on the ratio of the total score of the inliers versus the total score over the entire dataset.

**Lemma 3.5** (Filtering)**.** *Make Assumption 3.1. Let $A$ be an arbitrary symmetric matrix with $\|A\|_{\mathrm{F}} = 1$. Let $R = C(1/\alpha)\log(1/\eta)$ for $C \geq 100\sqrt{C_1}$. Define $\widetilde{y}_{\mathrm{median}} = \mathrm{Median}(\{\widetilde{x}^\top A \widetilde{x} \mid \widetilde{x} \in \widetilde{T}\})$. Define the function $f(\widetilde{x}) := (\widetilde{x}^\top A \widetilde{x} - \widetilde{y}_{\mathrm{median}})^2$. Let $m_1$ be a number less than $|S|/3$. Denote by $q_i$ the $i$-th smallest point of $\{\widetilde{x}^\top A \widetilde{x} \mid \widetilde{x} \in \widetilde{T}\}$. If $q_{|T|-m_1} - q_{m_1} \leq R$ and $\mathbf{E}_{X \sim T}[f(x)] > C'R^2/\alpha^3$ for $C' \geq 720/\epsilon_0$, that is, in the case where the check in Line 12 fails, then, the function $f(\cdot)$ satisfies $\sum_{\widetilde{x} \in \widetilde{T}} f(\widetilde{x}) > \frac{40}{\epsilon_0}\frac{1}{\alpha^3} \sum_{\widetilde{x} \in \widetilde{S} \cap \widetilde{T}} f(\widetilde{x})$ .*

The score ratio determines (in expectation) the ratio between the number of outliers and inliers removed. In Lemma 3.5, the ratio is in the order of $1/\alpha^3$—this will allow us to guarantee that at the end of the entire recursive execution of Algorithm 1, we would have removed at most a $0.01\alpha$ fraction of the inliers. See Remark F.5 in Appendix F for more details.

### 3.3   Divider: Identifying Multiple Clusters and Recursing

The previous subsections covered the cases where (i) the expected score is small, or (ii) the expected score over $\widetilde{T}$ is large and the $\alpha$ and $1 - \alpha$ quantiles of $\{\widetilde{X}^\top A \widetilde{X} \;:\; \widetilde{x} \in \widetilde{T}\}$ are close to each other. What remains is the case when both the expected score is large yet the quantiles are far apart. In this instance, we will not be able to make progress via filtering using the above argument. This is actually an intuitively reasonable scenario, since the outliers may in fact have another $\approx \alpha m$ samples that are distributed as a different Gaussian with a very different covariance—the algorithm would not be able to tell which Gaussian is supposed to be the inliers. We will argue that, when both the expected score is large and the quantiles are far apart, the samples are in fact easy to bipartition into 2 clusters, such that the most of the inliers fall within 1 side of the bipartition. This allows us to make progress outside of filtering, and this clustering mechanism also allows us to handle Gaussian mixture models and make sure we (roughly) handle components separately.

The key intuition is that, by the stability Conditions (L.3) and (L.4), we know that the inliers under the 1-dimensional projection $\{\widetilde{X}^\top A \widetilde{X} \;:\; \widetilde{X} \in \widetilde{S} \cap \widetilde{T}\}$ must be well-concentrated, in fact lying in an interval of length $\widetilde{O}(1/\alpha)$. The fact that the quantile range is wide implies that there must be some point within the range that is close to very few samples $\widetilde{X}^\top A \widetilde{X}$, by an averaging argument. We can then use the point as a threshold to bipartition our samples. See below for the precise statement.

---

**Algorithm 2** Divider for list decoding

1: **function** FINDDIVIDER($T, n', m_1$)
2:      Let the $m_1$-th smallest point be $q_{m_1}$ and $m_1$-th largest point be $q_{|T|-m_1}$.
3:      Divide the interval $[q_{m_1}, q_{|T|-m_1}]$ into $2m'/n'$ equally-sized subintervals.
4:      Find a subinterval $I'$ with at most $n'/2$ points and return its midpoint.

---

**Lemma 3.6** (Divider Algorithm)**.** *Let $T = (y_1, \ldots, y_{m'})$ be a set of $m'$ points in $\mathbb{R}$. Let $S_{\mathrm{proj}} \subset T$ be a set of $n'$ points such that $S_{\mathrm{proj}}$ is supported on an interval $I$ of length $r$. For every $i \in [m']$, let*

*the $i$-th smallest point of the set $T$ be $q_i$. Suppose $q_{|T|-m_1} - q_{m_1} \geq R$ such that $R \geq 10(m'/n')r$.*
*Then, given $T$ and $n', m_1$, Algorithm 2 returns a point $t$ such that if we define $T_1 = \{x \in T : x \leq t\}$*
*and $T_2 = T \setminus T_1$ then: (i) $\min(|T_1|, |T_2|) \geq m_1$ and (ii) $S_{\mathrm{proj}} \subseteq T_1$ or $S_{\mathrm{proj}} \subseteq T_2$.*

*Proof.* The last step of the algorithm must succeed by an averaging argument. Consider the midpoint $t$ of the returned subinterval $I'$, which is at least $q_{m_1}$ and at most $q_{|T|-m_1}$. Since $T_1$ contains all points at most $q_{m_1}$, and $T_2$ contains all points at most $q_{|T|-m_1}$, we must have $\min(|T_1|, |T_2|) \geq m_1$. Lastly, we verify the second desideratum, which holds if $t \notin I$. For the sake of contradiction, if $t \in I$, then since $I$ has length $r$ and $I'$ has length at least $R/(2m/n') \geq 5r$, then $I \subseteq I'$. However, since $|I' \cap T| \leq n'/2$, we know that $I$ cannot be strictly contained in $I'$, reaching the desired contradiction. $\square$

## 4  High-Level Proof Sketch of Theorem 1.2

We discuss the proof strategy of Theorem 1.2 at a high level. See Appendix F for the complete proof.

**Proof Sketch**  Recall that Algorithm 1 is a recursive algorithm: each call repeatedly filters out samples before either terminating or splitting the dataset into two and recursively calling itself. The execution of Algorithm 1 can thus be viewed as a binary tree, with each node being a recursive call.

The high-level idea for proving Theorem 1.2 is straightforward, though involving technical calculations to implement. Consider the subtree grown from the root recursive call, up to and including a certain level $j$. We proceed by induction on the height $j$ of such subtree, and claim there must exists a leaf node in this subtree such that most of the inliers remain in the input dataset of the leaf node.

Concretely, let $\mathcal{T}_j$ be the subtree of height $j$ grown from the root node. We claim that there must be a leaf in this subtree, whose input set $T$ satisfies

$$\alpha^3(\epsilon_0/40)|T| + |S \setminus T| \leq (j+1)\alpha^3(\epsilon_0/20)m + \ell \,, \tag{1}$$

recalling that $\alpha$ is the proportion of inliers, $\epsilon_0$ is the maximum fraction of inliers removed by the adversary, $\ell$ is the actual (unknown) number of inliers removed and $m$ is the size of the original dataset returned by the adversary. The left hand side keeps track of both a) how many inliers we have accidentally removed, through the $|S \setminus T|$ term, and b) the relative proportions of the outliers we have removed versus the inliers we have removed, by comparing both terms on the left hand side.

For the induction step, we need to analyze the execution of a single recursive call. We show that (1) implies Assumption 3.1, and so the inductive step can follow the case analysis outlined in Section 3— either we terminate, or we decide to either filter or bipartition the sample set. To convert this into a high-probability statement, we use a standard (sub-)martingale argument in Appendix F.1. $\square$

## Acknowledgements

We thank the anonymous reviewers for insightful comments and suggestions on this work. Ilias Diakonikolas is supported by NSF Medium Award CCF-2107079, NSF Award CCF-1652862 (CAREER), and a DARPA Learning with Less Labels (LwLL) grant. Daniel M. Kane is supported by NSF Medium Award CCF-2107547 and NSF Award CCF-1553288 (CAREER). Jasper C.H. Lee is supported in part by the generous funding of a Croucher Fellowship for Postdoctoral Research, NSF award DMS-2023239, NSF Medium Award CCF-2107079 and NSF AiTF Award CCF-2006206. Ankit Pensia is supported by NSF Awards CCF-1652862, and CCF-1841190, and CCF-2011255; The majority of this work was done while Ankit Pensia was at UW–Madison. Thanasis Pittas is supported by NSF Medium Award CCF-2107079 and NSF Award DMS-2023239 (TRIPODS).

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

# A  Additional Related Work

**Gaussian Mixture Models**  Gaussian mixture models (GMM) have a long history in statistics and theoretical computer science. In theoretical computer science, a system investigation of efficient algorithms for GMM was initiated by [Das99], and then improved upon in a series of works, see, for example, [VW04; AK05; DS07; KK10]. However, these algorithms imposed additional assumptions on the parameters of GMM, for example, separation between the components. Later, [MV10; BS10] gave algorithms without any assumptions on the components but these algorithms were sensitive to outliers. The field of algorithmic robust statistics has developed in the last decade to develop outlier-robust algorithms. A recent line of work [BDHKKK20; LM21; LM22; BDJKKV22] has developed efficient algorithms for learning GMM robustly.

**List-decodable Learning**  The model of list-decodable learning was first introduced in [BBV08] and was first used in the context of high-dimensional estimation in [CSV17]. Regarding mean estimation in particular, a body of work presented algorithms with increasingly better guarantees [DKS18; KS17; RY20a; CMY20; DKK20a; DKKLT21; DKKLT22; DKKPP22; ZS22].

# B  Omitted Preliminaries

## B.1  Notation

We use $\mathbb{Z}_+$ to denote positive integers. For $n \in \mathbb{Z}_+$, we denote $[n] \stackrel{\text{def}}{=} \{1, \dots, n\}$ and use $\mathcal{S}^{d-1}$ for the $d$-dimensional unit sphere. For a vector $v$, we let $\|v\|_2$ denote its $\ell_2$-norm. We use $I_d$ to denote the $d \times d$ identity matrix. We will drop the subscript when it is clear from the context. For a matrix $A$, we use $\|A\|_\mathrm{F}$ and $\|A\|_\mathrm{op}$ to denote the Frobenius and spectral (or operator) norms respectively. We say that a symmetric $d \times d$ matrix $A$ is PSD (positive semidefinite) and write $A \succeq 0$ if for all vectors $x \in \mathbb{R}^d$ we have that $x^\top A x \geq 0$. We denote $\lambda_{\max}(A) := \max_{x \in \mathcal{S}^{d-1}} x^\top A x$. We write $A \preceq B$ when $B - A$ is PSD. For a matrix $A \in \mathbb{R}^{d \times d}$, $\mathrm{tr}(A)$ denotes the trace of the matrix A. We denote by $\langle v, u \rangle$, the standard inner product between the vectors $u, v$. For matrices $U, V \in \mathbb{R}^{d \times d}$, we generalize the inner product as follows: $\langle U, V \rangle = \sum_{ij} U_{ij} V_{ij}$. For a matrix $A \in \mathbb{R}^{d \times d}$, we use $A^\flat$ to denote the flattened vector in $\mathbb{R}^{d^2}$, and for a $v \in \mathbb{R}^{d^2}$, we use $v^\sharp$ to denote the unique matrix $A$ such that $A^\flat = v^\sharp$. For a matrix $A$, we let $A^\dagger$ denote its pseudo-inverse. We use $\otimes$ to denote the Kronecker product. For a matrix $A$, we use $\ker(A)$ for the null space of $A$ and use $\mathbb{P}_A$ to denote the projection onto the range of $A$, i.e., $\mathbb{P}_A = AA^\dagger$ [Ber18, Proposition 8.1.7 (xii)].

We use the notation $X \sim D$ to denote that a random variable $X$ is distributed according to the distribution $D$. For a random variable $X$, we use $\mathbf{E}[X]$ for its expectation. We use $\mathcal{N}(\mu, \Sigma)$ to denote the Gaussian distribution with mean $\mu$ and covariance matrix $\Sigma$. For a set $S$, we use $\mathcal{U}(S)$ to denote the uniform distribution on $S$ and use $X \sim S$ as a shortcut for $X \sim \mathcal{U}(S)$.

We use $a \lesssim b$ to denote that there exists an absolute universal constant $C > 0$ (independent of the variables or parameters on which $a$ and $b$ depend) such that $a \leq Cb$.

## B.2  Miscellaneous Facts

In this section, we state standard facts that will be used in our analysis.

**Fact B.1.** *For any matrix* $A \in \mathbb{R}^{d \times d}$, $\mathbf{Var}_{X \sim \mathcal{N}(\mu, \Sigma)}[X^\top A X] \leq 4\|\Sigma^{1/2} A \Sigma^{1/2}\|_\mathrm{F}^2 + 8\|\Sigma^{1/2} A \mu\|_2^2$.

*Proof.* We write $X \sim \mathcal{N}(\mu, \Sigma)$ as $X = \Sigma^{1/2} Z + \mu$ for $Z \sim \mathcal{N}(0, I)$. We have that

$$
\begin{aligned}
\mathbf{Var}_{X \sim \mathcal{N}(\mu, \Sigma)}[X^\top A X] &= \mathbf{Var}_{Z \sim \mathcal{N}(0, I)}[(\Sigma^{1/2} Z + \mu)^\top A (\Sigma^{1/2} Z + \mu)] \\
&= \mathbf{Var}_{Z \sim \mathcal{N}(0, I)}[Z^\top \Sigma^{1/2} A \Sigma^{1/2} Z + 2\mu^\top A \Sigma^{1/2} Z + \mu^\top A \mu] \\
&\leq 2 \left( \mathbf{Var}_{Z \sim \mathcal{N}(0, I)}[Z^\top \Sigma^{1/2} A \Sigma^{1/2} Z] + 4 \mathbf{Var}_{Z \sim \mathcal{N}(0, I)}[\mu^\top A \Sigma^{1/2} Z] \right) \\
&\leq 4\|\Sigma^{1/2} A \Sigma^{1/2}\|_\mathrm{F}^2 + 8\|\Sigma^{1/2} A \mu\|_2^2 \,,
\end{aligned}
\tag{2}
$$

where the first inequality line uses that $\mathbf{Var}[A + B] = \mathbf{Var}[A] + \mathbf{Var}[B] + 2\,\mathbf{Cov}[A, B] \leq \mathbf{Var}[A] + \mathbf{Var}[B] + 2\sqrt{\mathbf{Var}[A]\,\mathbf{Var}[B]} = (\sqrt{\mathbf{Var}[A]} + \sqrt{\mathbf{Var}[B]})^2 \leq 2(\mathbf{Var}[A] + \mathbf{Var}[B])$, and the last inequality uses the following: The first term uses the fact that for a symmetric matrix $B$ and a vector $v$, $\mathbf{Var}_{Z \sim \mathcal{N}(0,I)}[Z^\top B Z] = 2\|B\|_\mathrm{F}^2$ and $\mathbf{Var}_{Z \sim \mathcal{N}(0,I)}[v^\top Z] = \|v\|_2^2$, with $B := \Sigma^{1/2} A \Sigma^{1/2}$ and $v := \Sigma^{1/2} A \mu$. $\hfill\square$

**Fact B.2** (Azuma-Hoeffding Inequality). *Let $\{X_n\}_{n \in \mathbb{N}}$ be a submartingale or supermatingale with respect to a sequence $\{Y_n\}_{n \in \mathbb{N}}$. If for all $n = 1, 2, \ldots$ it holds $|X_n - X_{n-1}| \leq c_n$ almost surely, then for any $\epsilon > 0$ and $n \in \mathbb{N}$*

$$\Pr\left[X_n - X_t > \epsilon\right] \leq \exp\left(-\frac{2\epsilon^2}{\sum_{t=1}^\infty c_t}\right).$$

**Fact B.3** (See, e.g., [Ber18, Proposition 11.10.34]). *Let $R$ and $M$ be two square matrices, then $\|RM\|_\mathrm{F} \leq \min\left((\|R\|_\mathrm{F}\|M\|_\mathrm{op}, \|R\|_\mathrm{op}\|M\|_\mathrm{F}\right)$.*

**Fact B.4** (Frobenius norm and projection matrices). *Let $A$ be a symmetric matrix. If $L_1$ and $L_2$ are two PSD projection matrices satisfying $L_1 \preceq L_2$, then $\|L_1 A L_1^\top\|_\mathrm{F} \leq \|L_2 A L_2^\top\|_\mathrm{F}$.*

**Fact B.5** (Pseudo-inverse of a matrix). *Let $G$ be a square matrix and define $R = G^\dagger$. Let the SVD of $G$ be $U \Lambda V^\top$. Then $R = V \Lambda^\dagger U^\top$, $\mathbb{P}_G = U \Lambda \Lambda^\dagger U^\top$, and $R = R \mathbb{P}_G$. Moreover, Let $\Delta$ be any arbitrary diagonal matrix such that if $\Lambda_{i,i} = 0$, then $\Delta_{i,i} = 0$, and define $L = V \Delta V^\top$. Then $L = LRG$.*

*Proof.* The first follows from [Ber18, Chapter 8], the second follows from [Ber18, Proposition 8.1.7 (xii)], and the third follows from the first two and the fact that $\Lambda^\dagger \Lambda \Lambda^\dagger = \Lambda$. The final claim follows from the following series of equalities:

$$\begin{aligned} LRG &= V \Delta V^\top V \Lambda^\dagger U^\top U \Lambda V^\top \\ &= V \Delta V^\top V \Lambda^\dagger U^\top U \Lambda V^\top \\ &= V \Delta \Lambda^\dagger \Lambda V^\top \\ &= V \Delta V^\top = L, \end{aligned}$$

where we use that $\Delta$ is diagonal and for each $i$ such that $\Delta_{i,i} \neq 0$, $\Lambda_{i,i}^\dagger \Lambda_{i,i} = 1$.

$\hfill\square$

**Fact B.6** ([Ber18, Fact 8.4.16]). *Let $A$ and $B$ be two square matrices and suppose that $A$ is full rank. Then $\mathbb{P}_{BA} = \mathbb{P}_B$.*

# C  Linear Algebraic Results

In this section, we prove the proofs of certain linear algebraic results that we need. In particular, we provide the proofs of Lemmata C.1 and C.3 that will be used later.

## C.1  Relative Frobenius Norm to Frobenius Norm: Proof of Lemma C.1

In this section, we provide the proof of Lemma C.1.

**Lemma C.1.** *Consider two arbitrary full rank, positive definite matrices $\Sigma_1$ and $\Sigma_2$, and suppose there exists a full rank, positive definite matrix $H$ such that (i) $\|I - H^{\dagger/2} \Sigma_1 H^{\dagger/2}\|_\mathrm{F} \leq \rho$ and (ii) $\|I - H^{\dagger/2} \Sigma_2 H^{\dagger/2}\|_\mathrm{F} \leq \rho$. Then, for an arbitrary full rank, positive definite matrix $\Sigma$, we have*

$$\|\Sigma^{-1/2} \Sigma_1 \Sigma^{-1/2} - \Sigma^{-1/2} \Sigma_2 \Sigma^{-1/2}\|_\mathrm{F} \leq 5\rho \max(\|\Sigma^{-1/2} \Sigma_1 \Sigma^{-1/2}\|_\mathrm{op}, \|\Sigma^{-1/2} \Sigma_2 \Sigma^{-1/2}\|_\mathrm{op}).$$

*Proof of Lemma C.1.* Let $G = H^{-1/2} \Sigma^{1/2}$. Let $\widetilde{\Sigma}_1 = \Sigma^{-1/2} \Sigma_1 \Sigma^{-1/2}$ and $\widetilde{\Sigma}_2 = \Sigma^{-1/2} \Sigma_2 \Sigma^{-1/2}$, both of which are positive definite matrices. Since $\Sigma$ is full rank, we obtain that

$$\begin{aligned} \Sigma_1 &= \Sigma^{1/2} \Sigma^{-1/2} \Sigma_1 \Sigma^{-1/2} \Sigma^{1/2} = \Sigma^{1/2} \widetilde{\Sigma}_1 \Sigma^{1/2} \\ \Sigma_2 &= \Sigma^{1/2} \Sigma^{-1/2} \Sigma_2 \Sigma^{-1/2} \Sigma^{1/2} = \Sigma^{1/2} \widetilde{\Sigma}_2 \Sigma^{1/2}. \end{aligned}$$

This directly gives us that $G\widetilde{\Sigma}_1 G^\top = I - H\Sigma_1 H$ and $G\widetilde{\Sigma}_2 G^\top = I - H\Sigma_2 H$. Overall, we have obtained the following:

$$\|I - G\widetilde{\Sigma}_1 G^\top\|_{\mathrm{F}} \le \rho \quad \text{and} \quad \|I - G\widetilde{\Sigma}_2 G^\top\|_{\mathrm{F}} \le \rho.$$

We first state a result, proved in the end of this section, that transfers closeness in relative Frobenius norm to Frobenius norm of the difference (along most of the directions).

**Lemma C.2.** *Let $G$ be a $d \times d$ matrix and let $B$ be an arbitrary symmetric matrix of the same dimension. Recall that $\mathbb{P}_G$ denotes the projection matrix onto range of $G$. Suppose that*

$$\|\mathbb{P}_G - GBG^\top\|_{\mathrm{F}} \le \rho$$

*for some $\rho \ge 1$. Let the SVD of $G$ be $G = U\Lambda V^\top$ for some orthonormal matrices $U$ and $V$ and a diagonal matrix $\Lambda$.*

*Then there exists a $d \times d$ matrix $L$ such that*

1. *(L captures directions of G) $L = V\Delta_B V^\top$ for some binary diagonal matrix $\Delta_B \preceq \Lambda\Lambda^\dagger$. That is, the diagonal entries of $\Delta_B$ are all in $\{0, 1\}$, and furthermore, all the 1 diagonal entries of $\Delta_B$ are also non-zero diagonal entries in $\Lambda$.*

2. *(rank of L) The rank of $L$ satisfies $\|\Lambda - \Delta_B\|_{\mathrm{F}} \le 2\rho$ and*

3. *(closeness of $G^\dagger$ and $B$ along L) $\left\|L\left(RR^\top - B\right)L^\top\right\|_{\mathrm{F}} \le 2\rho\|B\|_{\mathrm{op}}$, where $R = G^\dagger$.*

We will now apply Lemma C.2 twice following the same notation as in its statement. In particular, the SVD decomposition of $G = U\Delta V^\top$ and $R = G^\dagger$. Lemma C.2 implies that there exist two binary diagonal matrices $\Delta_1$ and $\Delta_2$, where $\Delta_1$ is for $\widetilde{\Sigma}_1$ and $\Delta_2$ is for $\widetilde{\Sigma}_2$ such that (i) both $\|I - \Delta_1\|_{\mathrm{F}}$ and $\|I - \Delta_1\|_{\mathrm{F}}$ are at $2\rho$, and (ii) $V\Delta_1 V^\top(RR^\top - \widetilde{\Sigma}_1)V\Delta_1 V^\top$ and $V\Delta_2 V^\top(RR^\top - \widetilde{\Sigma}_2)V\Delta_2 V^\top$ have Frobenius norms at most $2\rho\|\widetilde{\Sigma}_1\|_{\mathrm{op}}$ and $2\rho\|\widetilde{\Sigma}_2\|_{\mathrm{op}}$, respectively. Let $\Delta = \Delta_1 \times \Delta_2$ and define $L = V\Delta V^\top$. By (i), we have that $\|I - L\|_{\mathrm{F}} = \|I - \Delta\|_{\mathrm{F}} \le 4\rho$. Since $L \preceq V\Delta_1 V^\top$ and $L$ is a projection matrix, Fact B.4 along with (ii) above implies that

$$\|L(RR^\top - \widetilde{\Sigma}_1)L^\top\|_{\mathrm{F}} \le 2\rho\|\widetilde{\Sigma}_1\|_{\mathrm{op}}.$$

Similarly for $\widetilde{\Sigma}_2$. By triangle inequality, we obtain that

$$\|L(\widetilde{\Sigma}_1 - \widetilde{\Sigma}_2)L^\top\|_{\mathrm{F}} \le 4\rho \max\left(\|\widetilde{\Sigma}_1\|_{\mathrm{op}}, \|\widetilde{\Sigma}_2\|_{\mathrm{op}}\right).$$

Since $L$ is a projection matrix, triangle inequality implies that

$$
\begin{aligned}
\left\|\left(\widetilde{\Sigma}_1 - \widetilde{\Sigma}_2\right)\right\|_{\mathrm{F}} &\le \left\|L\left(\widetilde{\Sigma}_1 - \widetilde{\Sigma}_2\right)L^\top\right\|_{\mathrm{F}} + \left\|(I - L)\left(\widetilde{\Sigma}_1 - \widetilde{\Sigma}_2\right)(I - L)^\top\right\|_{\mathrm{F}} \\
&\lesssim \left\|L\left(\widetilde{\Sigma}_1 - \widetilde{\Sigma}_2\right)L^\top\right\|_{\mathrm{F}} + \|I - L\|_{\mathrm{F}} \cdot \left\|\left(\widetilde{\Sigma}_1 - \widetilde{\Sigma}_2\right)\right\|_{\mathrm{op}} \\
&\lesssim \rho \cdot \max\left(\|\widetilde{\Sigma}_1\|_{\mathrm{op}}, \|\widetilde{\Sigma}_2\|_{\mathrm{op}}\right).
\end{aligned}
$$

This completes the proof.

$\square$

We now provide the proof of Lemma C.2.

**Lemma C.2.** *Let $G$ be a $d \times d$ matrix and let $B$ be an arbitrary symmetric matrix of the same dimension. Recall that $\mathbb{P}_G$ denotes the projection matrix onto range of $G$. Suppose that*

$$\|\mathbb{P}_G - GBG^\top\|_{\mathrm{F}} \le \rho$$

*for some $\rho \ge 1$. Let the SVD of $G$ be $G = U\Lambda V^\top$ for some orthonormal matrices $U$ and $V$ and a diagonal matrix $\Lambda$.*

*Then there exists a $d \times d$ matrix $L$ such that*

1. *(L captures directions of G)* $L = V\Delta_B V^\top$ *for some binary diagonal matrix* $\Delta_B \preceq \Lambda\Lambda^\dagger$. *That is, the diagonal entries of* $\Delta_B$ *are all in* $\{0,1\}$, *and furthermore, all the 1 diagonal entries of* $\Delta_B$ *are also non-zero diagonal entries in* $\Lambda$.

2. *(rank of L) The rank of L satisfies* $\|\Lambda - \Delta_B\|_F \le 2\rho$ *and*

3. *(closeness of* $G^\dagger$ *and B along L)* $\left\|L\left(RR^\top - B\right)L^\top\right\|_F \le 2\rho\|B\|_{\mathrm{op}}$, *where* $R = G^\dagger$.

*Proof.* Let $u_1, u_2, \ldots, u_d$ be the columns of $U$ and let $v_1, v_2, \ldots, v_d$ be the columns of $V$. Let the entries of $\Lambda$ be $\lambda_1, \lambda_2, \ldots, \lambda_d$, which are non-negative by SVD property. Let $\mathcal{J}$ be the set of $i$'s in $[d]$ such that $\lambda_i > 0$.

Define the matrices $F := U^\top GBG^\top U$ and $B' := V^\top BV$. Then since $G^\top = V\Lambda U^\top$, we have
$$F = U^\top GBG^\top U = U^\top U\Lambda V^\top BV\Lambda U^\top U = \Lambda V^\top BV\Lambda = \Lambda B'\Lambda.$$
Thus the $(i,j)$-th entry of $F$ is $B'_{i,j}\lambda_i\lambda_j$.

Now define the matrix $M := U^\top \mathbb{P}_G U$. From [Fact B.5], we have that $\mathbb{P}_G = U\Lambda\Lambda^\dagger U^\top$. And thus, $M = \Lambda\Lambda^\dagger$, which is the diagonal matrix with entries $M_{ii} = 1$ for each $i \in \mathcal{J}$, and $M_{ii} = 0$ otherwise.

Using the invariance of Frobenius norm under orthogonal transform, we obtain the following series of equalities:
$$
\begin{aligned}
&\|\mathbb{P}_G - GBG^\top\|_F^2 \\
&= \left\|U^\top\left(\mathbb{P}_G - GBG^\top\right)U\right\|_F^2 \\
&= \|M - F\|_F^2 \\
&\ge \sum_{i \in \mathcal{J}}\left(M_{i,i} - F_{i,i}\right)^2 \\
&= \sum_{i \in \mathcal{J}}\left(1 - B'_{i,i}\cdot\lambda_i^2\right)^2
\end{aligned}
\tag{3}
$$

Let $\mathcal{I} \subseteq \mathcal{J}$ be the set of $i$'s in $\mathcal{J}$ such that $\lambda_i^2 B'_{i,i} < (1/2)$. Then $\sum_{i\in\mathcal{I}}\left(1 - B'_{i,i}\cdot\lambda_i^2\right)^2 > |\mathcal{I}|/4$. Thus if $|\mathcal{I}|$ is larger than $4\rho^2$, then the Frobenius norm squared is larger than $r^2$, which is a contradiction.

If $i \in \mathcal{J} \setminus \mathcal{I}$, then $\lambda_i^2 \ge 1/(2B'_{i,i})$ and not just $\lambda_i > 0$.

Define the matrix $L$ to be $L := \sum_{i:i\in\mathcal{J}\setminus\mathcal{I}} v_iv_i^\top$, which is equal to $V\Delta_B V^\top$ where $\Delta_B$ is a diagonal matrix with the $i$-th diagonal entry equal to 1 if $i \in \mathcal{J}\setminus\mathcal{I}$ and 0 otherwise. By definition, $L$ has rank $|\mathcal{J}| - |\mathcal{I}|$ and furthermore, the 1 diagonal entries of $\Delta_B$ are always non-zero diagonal entries in $\Lambda$.

Let $R = G^\dagger$, and by [Fact B.5], we have that $R = V\Lambda^\dagger U^\top$. Thus $LR = V\Delta_B V^\top V\Lambda^\dagger U^\top = V\Delta_B\Lambda^\dagger U^\top$, and thus the singular values of $LR$ satisfy that they are at most
$$\frac{1}{\min_{i\in\mathcal{J}\setminus\mathcal{I}}\lambda_i} \le \max_i\sqrt{2B'_{i,i}} \le \sqrt{2\|B'\|_{\mathrm{op}}} = \sqrt{2\|V^\top BV\|_{\mathrm{op}}} = \sqrt{2\|B\|_{\mathrm{op}}}.$$

We first need two more observations: $L = LRG$ and $LRR^\top L^\top = LR\mathbb{P}_G R^\top L^\top$, both of which hold by [Fact B.5]. Finally, we show that $RR^\top$ and $B$ are close along the directions in $L$.
$$
\begin{aligned}
&\|L(RR^\top - B)L^\top\|_F \\
&= \|LRR^\top L^\top - LBL^\top\|_F \\
&= \|LR\mathbb{P}_G R^\top L^\top - LRGBG^\top R^\top L^\top\|_F \\
&= \|LR(\mathbb{P}_G - GBG^\top)R^\top L^\top\|_F \\
&\le \|LR\|_{\mathrm{op}}\|\mathbb{P}_G - GBG^\top\|_F\|R^\top L^\top\|_{\mathrm{op}} \\
&= \|LR\|_{\mathrm{op}}^2\|\mathbb{P}_G - GBG^\top\|_F \\
&\le 2\|B\|_{\mathrm{op}}\cdot\rho.
\end{aligned}
\tag{4}
$$
This completes the proof. $\qquad\square$

## C.2 Proof of Lemma C.3

We now restate and prove Lemma C.3.

**Lemma C.3.** *Let $A$ and $B$ be two PSD matrices with $\mathbb{P}_B \preceq \mathbb{P}_A$ or equivalently $\ker(A) \subseteq \ker(B)$. Then $\|ABA - \mathbb{P}_B\|_{\mathrm{F}} \leq 2\|ABA - \mathbb{P}_A\|_{\mathrm{F}}$.*

*Proof.* Applying triangle inequality, we have that

$$\|ABA - \mathbb{P}_B\|_{\mathrm{F}} \leq \|ABA - \mathbb{P}_A\|_{\mathrm{F}} + \|\mathbb{P}_A - \mathbb{P}_B\|_{\mathrm{F}}.$$

It thus suffices to show that the second term, $\|\mathbb{P}_B - \mathbb{P}_A\|_{\mathrm{F}}$, is less than $\|ABA - \mathbb{P}_A\|_{\mathrm{F}}$. Let $d_A$ be the rank of $A$ and $d_B$ be the rank of $B$. We will show the following two results, which imply the desired bound:

$$\|\mathbb{P}_B - \mathbb{P}_A\|_{\mathrm{F}}^2 = |d_B - d_A| \leq \|ABA - \mathbb{P}_B\|_{\mathrm{F}}^2 \tag{5}$$

In the rest of the proof, we will establish the above two inequalities.

We begin with the second inequality in (5). By the lemma assumptions, we have that the null space of $A$ being a subspace of the null space of $B$. This is equivalent to saying that the column space of $B$ being a subspace of the column space of $A$. In particular, this also implies that $d_B \leq d_A$. It is easy to see that $\mathrm{rank}(ABA) \leq \mathrm{rank}(B) \leq \mathrm{rank}(A) = \mathrm{rank}(\mathbb{P}_A)$, and the eigenvalues of $\mathbb{P}_A$ are either 0 or 1. We now state a simple result lower bounding the Frobenius norm by the difference of the ranks for such matrices:

**Claim C.4** (Bound on rank difference). *Let $A'$ be a symmetric matrix such that the eigenvalues of $A'$ belong to $\{0, 1, -1\}$. Let $B'$ be an arbitrary matrix. Then $\|B' - A'\|_{\mathrm{F}}^2 \geq \mathrm{rank}(A') - \mathrm{rank}(B')$.*

*Proof.* Let $u_1, \ldots u_d$ be the orthonormal eigenvectors of $A'$ with eigenvalues $\lambda_1, \lambda_2, \ldots, \lambda_d$ such that $|\lambda_i| = 1$ for $i \leq \mathrm{rank}(A')$ and $\lambda_i = 0$ otherwise. Without loss of generality, we assume $\mathrm{rank}(B') \leq \mathrm{rank}(A')$, otherwise the result is trivial. Let $\mathcal{I}$ be the set of $i$'s such that $i \leq \mathrm{rank}(A')$ and $B'u_i = 0$.

$$\|B' - A'\|_{\mathrm{F}}^2 = \sum_{i=1}^{d} \|B'u_i - A'u_i\|_2^2 \geq \sum_{i=1}^{\mathrm{rank}(A')} \|B'u_i - A'u_i\|_2^2$$
$$= \sum_{i=1}^{\mathrm{rank}(A')} \|B'u_i - \lambda_i u_i\|_2^2. \geq \sum_{i \in \mathcal{I}} \|\lambda_i u_i\|_2^2 = |\mathcal{I}|.$$

Since the rank of $B'$ is less than the rank of $A'$, it follows that $B'u_i$ is nonzero for at most $\mathrm{rank}(B')$ many $u_i$'s in $\{u_i : i \leq \mathrm{rank}(A')\}$. Thus $|\mathcal{I}|$ is at least $\mathrm{rank}(A') - \mathrm{rank}(B')$. $\square$

By Claim C.4, we have that $\mathrm{rank}(\mathbb{P}_B) - \mathrm{rank}(ABA) \leq \|\mathbb{P}_B - ABA\|_{\mathrm{F}}^2$. Thus, it implies the desired inequality, $|d_B - d_A| \leq \|ABA - \mathbb{P}_A\|_{\mathrm{F}}^2$.

We will now establish the first inequality in (5). Observe that both $\mathbb{P}_A$ and $\mathbb{P}_B$ are PSD matrices, whose eigenvalues are either 1 or 0. The following result upper bounds the Frobenius norm of $\mathbb{P}_A - \mathbb{P}_B$ in terms of the difference of their ranks.

**Claim C.5.** *If $A$ and $B$ are two PSD matrices such that $\mathbb{P}_B \preceq \mathbb{P}_A$, then $\|\mathbb{P}_A - \mathbb{P}_B\|_{\mathrm{F}}^2 = \mathrm{rank}(A) - \mathrm{rank}(B)$.*

*Proof.* Let $d_A$ be the rank of $A$ and $d_B$ be the rank of $B$. We have that $d_B = \mathrm{rank}(\mathbb{P}_B) \leq \mathrm{rank}(\mathbb{P}_A) = d_A$. Consider any orthonormal basis $v_1, \ldots, v_d$ in $\mathbb{R}^d$, such that $v_1, \ldots, v_{d_A}$ spans the column space of $\mathbb{P}_A$, and $v_1, \ldots, v_{d_B}$ spans the column space of $B$. Since the column space of $\mathbb{P}_B$ is a subspace of that of $\mathbb{P}_B$, such an orthonormal basis always exists.

Moreover, since the eigenvalues of $\mathbb{P}_A$ are either 0 or 1, we have that $\mathbb{P}_A v_i = v_i$ for $i \leq d_A$ and 0 otherwise. Similarly, $\mathbb{P}_B v_i = v_i$ for $i \leq d_B$ and 0 otherwise.

We will also use the following basic fact: for any square matrix $J$ and any set of orthonormal basis vectors $u_1, \ldots, u_d$, we have that $\|J\|_{\mathrm{F}}^2 = \sum_{i=1}^d \|Ju_i\|_2^2$. Applying this fact to $\mathbb{P}_A - \mathbb{P}_B$ and the orthonormal basis $v_1, \ldots, v_d$ above, we get the following:

$$\|\mathbb{P}_A - \mathbb{P}_B\|_{\mathrm{F}}^2 = \sum_{i=1}^d \|\mathbb{P}_A v_i - \mathbb{P}_B v_i\|_2^2 = \sum_{i=1}^{d_B} \|\mathbb{P}_A v_i - \mathbb{P}_B v_i\|_2^2 + \sum_{d_B+1}^{d_A} \|\mathbb{P}_A v_i - \mathbb{P}_B v_i\|_2^2 = \sum_{d_B+1}^{d_A} \|v_i\|_2^2$$

$$= d_A - d_B,$$

where the second equality uses that $\mathbb{P}_A v_i = \mathbb{P}_B v_i = 0$ for $i \geq d_A$, and the third equality uses that $\mathbb{P}_A v_i = \mathbb{P}_B v_i = v_i$ for $i \leq d_B$ and $\mathbb{P}_B v_i = 0$ and $\mathbb{P}_A v_i = v_i$ for $i \in \{d_B + 1, \ldots, d_A\}$. $\qquad\square$

By Claim C.5, we have that $\|\mathbb{P}_A - \mathbb{P}_B\|_{\mathrm{F}}^2 = |\operatorname{rank}(A) - \operatorname{rank}(B)|$.

$\qquad\square$

# D   Stability: Sample Complexity and Its Consequences

In this section, we prove results related to stability. Appendix D.1 proves the sample complexity for the stability condition to hold for Gaussians and Appendix D.2 focuses on some consequences of stability.

**Remark D.1.** We note that the deterministic conditions of Definition 2.1 are not specific to the Gaussian distribution but hold with polynomial sample complexity $\operatorname{poly}(d/\eta)$ for broader classes of distributions, roughly speaking, distributions with light tails for degree-4 polynomials.

- Condition (L.1) is satisfied with polynomial sample complexity by distributions that have bounded second moment.

- Condition (L.3) at the population level is a direct implication of hypercontractivity of degree-2 polynomials. The $\log(1/\eta)$ factor in the condition as stated currently is tailored to the Gaussian distribution but it is not crucial for the algorithm and could be relaxed to some polynomial of $1/\eta$; this would translate to incurring some $\operatorname{poly}(1/\eta)$ to the final error guarantee of the main theorem. After establishing the condition at the population level, it can be transferred to sample level with polynomial sample complexity.

- Regarding Condition (L.2), every entry of the Frobenius norm is a degree 2 polynomial, thus the property holds for distributions with light tails for these polynomials, e.g., hyper-contractive degree-2 distributions.

- Condition (L.4) can also be derived by hypercontractivity of degree 4 polynomials similarly.

- Condition (L.5) holds almost surely for continuous distributions.

## D.1   Sample Complexity of Stability Conditions

**Lemma 2.2** (Deterministic Conditions Hold with High Probability). *For a sufficiently small positive constant $\epsilon_0$ and a sufficiently large absolute constant $C$, a set of $m > Cd^2 \log^5(d/(\eta\delta))/\eta^2$ samples from $\mathcal{N}(\mu, \Sigma)$, with probability $1 - \delta$, is $(\eta, \epsilon)$-stable set with respect to $\mu, \Sigma$ for all $\epsilon \leq \epsilon_0$.*

*Proof.* Let $A$ be a set of i.i.d. samples from $\mathcal{N}(\mu, \Sigma)$. We check the four conditions from Definition 2.1 separately. Since $A$ is a set of i.i.d. samples from a Gaussian, it satisfies Condition (L.1) with high probability (see, e.g., [Li18, Corollary 2.1.13]).

We start with Condition (L.2). Let $X \sim \mathcal{N}(\mu, \Sigma)$, then $X - \mu \sim \mathcal{N}(0, \Sigma)$. Equivalently, $X - \mu = \Sigma^{1/2} Z$ with $Z \sim \mathcal{N}(0, I)$. Let $A$ be a set of $m$ i.i.d. samples from $\mathcal{N}(\mu, \Sigma)$ and like before, for every $x \in A$, write $x - \mu = \Sigma^{1/2} z$, where $z$ are corresponding i.i.d. samples from $\mathcal{N}(0, I)$. Let a subset $A' \subset A$ with $|A'| \geq |A|(1 - \epsilon)$. Then, we have the following

$$\operatorname{tr}\left(\left(\frac{1}{n}\sum_{x \in A'}(x - \mu)(x - \mu)^\top - \Sigma\right)U\right) = \operatorname{tr}\left(\Sigma^{1/2}\left(\frac{1}{n}\sum_z zz^\top - I\right)\Sigma^{1/2}U\right)$$

$$= \mathrm{tr}\left(\left(\frac{1}{n}\sum_{z} zz^{\top} - I\right)\Sigma^{1/2}U\Sigma^{1/2}\right)$$

$$\leq C\epsilon \log(1/\epsilon)\|\Sigma^{1/2}U\Sigma^{1/2}\|_{\mathrm{F}},$$

where the first line uses the rewriting that we introduced before, the second line uses the cyclic property of the trace operator, and the last line uses [Li18, Corollary 2.1.13]. As stated in that corollary, the inequality holds with probability $1 - \delta/4$ as long as the sample complexity is a sufficiently large multiple of $(d^2/\epsilon^2)\log(4/\delta)$. Fixing $\epsilon = \epsilon_0$ a sufficiently small positive absolute constant can make the right hand side of the above inequality $0.1\|\Sigma^{1/2}U\Sigma^{1/2}\|_{\mathrm{F}}$.

Condition Condition (L.1) can be shown identically using [Li18, Corollary 2.1.9].

We now turn our attention to Condition (L.3). [DKKLMS16, Lemma 5.17] implies that with the stated number of samples, with high probability, it holds that

$$\Pr_{X\sim A}\left[|p(x)| > \sqrt{\mathbf{Var}_{Y\sim\mathcal{N}(\mu,\Sigma)}[p(Y)]10\ln\left(\frac{2}{\eta}\right)}\right] \leq \frac{\eta}{2}.$$

Since $|A'| \geq |A|(1 - \epsilon_0) \geq |A|/2$, Condition (L.3) holds. Finally, Condition (L.4) follows by noting that $\mathbf{Var}_{X\sim A'}[p(X)] \leq 2\mathbf{Var}_{X\sim A}[p(X)]$ and using [DKKLMS16, Lemma 5.17] again with $\epsilon = \epsilon_0$. In order for the conclusion of that lemma to hold with probability $1 - \delta/4$, the number of samples of the Gaussian component is a sufficiently large multiple of $d^2 \log^5(4d/(\eta\delta))/\eta^2$. A union bound over the events corresponding to each of the four conditions concludes the proof.

Lastly, we show Condition (L.5). We will in fact show a stronger statement: suppose the rank of $\Sigma$ is $d_\Sigma$. Then, if we take $m \geq d_\Sigma + 1$ i.i.d. samples $\{\mu + z_1, \mu + z_2, \ldots, \mu + z_m\}$ from $\mathcal{N}(\mu, \Sigma)$, it must be the case that with probability 1, for all $A' \subseteq A$ with $|A'| \geq d_\Sigma + 1$, we have $\sum_{x\in A'}(v^{\top}x)^2 = 0$ (or equivalently, $v$ being orthogonal to all vectors $x \in A'$) implying $v^{\top}\Sigma v = 0$ for all vectors $v \in \mathbb{R}^d$.

We will need the following straightforward claim: if $A' = \{\mu + z'_0, \ldots, \mu + z'_{d_\Sigma}\}$ (of size $d_\Sigma + 1$) is such that 1) $\{z'_1 - z'_0, \ldots, z'_{d_\Sigma} - z'_0\}$ are linearly independent and 2) $z'_i$ all lie within the column space of $\Sigma$, then for any vector $v \in \mathbb{R}^d$, $v$ being orthogonal to all of $A'$ implies that $v^{\top}\Sigma v = 0$. To see why this is true, observe that $\{z'_1 - z'_0, \ldots, z'_{d_\Sigma} - z'_0\}$, which are linearly independent vectors that all lie in the column space of $\Sigma$, must span the entirety of the column space of $\Sigma$ by dimension counting. Thus, if $v$ is orthogonal to all the points in $A'$, then $v$ is orthogonal to all of $\{z'_1 - z'_0, \ldots, z'_{d_\Sigma} - z'_0\}$, which implies $v^{\top}\Sigma v = 0$.

We can now show the following inductive claim on the size of $A$, which combined with the above claim implies the strengthened version of Condition (L.5). Specifically, we show inductively that, with probability 1 over the sampling of $A$, for any subset $A' \subset A$ of size at most $d_\Sigma + 1$, the claim conditions hold for $A'$. Namely, if $A' = \{\mu + z'_0, \ldots, \mu + z'_j\}$ for some $j \leq d_\Sigma$, for some arbitrary ordering of the elements in $A'$, then 1) the set $\{z'_1 - z'_0, \ldots, z'_j - z'_0\}$ is linearly independent and 2) $z'_i$ all lie in the column space of $\Sigma$.

The base case of $|A| = 1$ is trivial. Now suppose the above statement is true for $|A_\ell| = \ell$, and consider sampling a new point $\mu + z_\ell$ and adding it to $A_\ell$ to form $A_{\ell+1}$. Take any subset $A'$ of $A_{\ell+1}$ of size at most $d_\Sigma + 1$. If $A' \subset A_\ell$ we are already done by the induction hypothesis. Otherwise, $A' = \{\mu + z_\ell\} \cup A'_\ell$, where $A'_\ell = \{\mu + z'_0, \ldots, \mu + z'_j\}$ has size at most $d_\Sigma$ and satisfies the conditions in the induction by the inductive hypothesis. The space of $\mu + z$ such that $z - z'_0$ lies in the span of $\{z'_1 - z'_0, \ldots, z'_j - z'_0\}$ has measure 0 under $\mathcal{N}(\mu, \Sigma)$, given there are strictly fewer than $d_\Sigma$ many linearly independent vectors in $\{z'_1 - z'_0, \ldots, z'_j - z'_0\}$, all of which lie in the column space of $\Sigma$. Furthermore, there is only a finite number of such $A'_\ell$. Thus, with probability 1, the new point $\mu + z_\ell$ will be sampled such that for all such $A'_\ell$, $\{\mu + z_\ell\} \cup A'_\ell$ remains linearly independent. Additionally, also with probability 1, $z_\ell$ will lie in the column space of $\Sigma$. This completes the proof of the inductive claim, which as we have shown implies Condition (L.5).

$\square$

## D.2 Facts Regarding Stability

The following corollaries will be useful later on. In particular, since our algorithm will perform transformations of the dataset, we will need Corollary D.2 to argue about the stability properties of these sets. Moreover, the next lemma (Lemma D.3) will be used to extract the final Frobenius norm guarantee of our theorem given the first bullet of that theorem. The proofs are deferred to Appendix D.4.

**Corollary D.2** (Stability of Transformed Sets). *Let $D$ be a distribution over $\mathbb{R}^d$ with mean $\mu$ and covariance $\Sigma$. Let $A$ be a set of points in $\mathbb{R}^d$ that satisfies Conditions (L.1) and (L.2) of Definition 2.1 of $(\eta, \epsilon)$-stability with respect to $D$. Then we have that for any symmetric transformation $P \in \mathbb{R}^{d \times d}$ and every $A' \subset A$ with $|A'| \geq (1 - \epsilon)|A|$, the following holds:*

1. $\left\| P \left( \frac{1}{|A'|} \sum_{x \in A'} x - \mu \right) \right\|_2 \leq 0.1\sqrt{\|P\Sigma P\|_{\text{op}}}.$

2. $\left\| P \left( \frac{1}{|A'|} \sum_{x \in A'} (x - \mu)(x - \mu)^\top - \Sigma \right) P \right\|_{\text{F}} \leq 0.1\|P\Sigma P\|_{\text{op}}.$

The following lemma (proved in Appendix D.4) will be needed to prove our certificate lemma to transfer the final guarantees of our algorithm from the empirical covariance of the sample set to the actual covariance of the Gaussians.

**Lemma D.3.** *Let $A$ be a set of points in $\mathbb{R}^d$ satisfying Conditions (L.1) and (L.2) of $(\eta, \epsilon)$-stability (Definition 2.1) with respect to $\mu, \Sigma$. Let $A' \subseteq A$ such that $|A'| \geq (1 - \epsilon)|A|$. If $P, L \in \mathbb{R}^{d \times d}$ are symmetric matrices with the property that $\| \mathbf{Cov}_{X \sim A'}[PX] - L\|_{\text{F}} \leq r$, then*

$$\|P\Sigma P - L\|_{\text{F}} = O(r + \|L\|_{\text{op}}) .$$

## D.3 Certificate Lemma

The following result shows that if the covariance matrix of $X^{\otimes 2}$ over the corrupted set of points is bounded in operator norm, then the empirical second moment matrix of $T$ is a good estimate of covariance of inliers in Frobenius norm. We prove a more general version below that allows for arbitrary linear transformation matrix $P$ and since we are interested in guarantees for relative Frobenius norm, our algorithm will chose $P$ so that $\mathbf{E}_{X \sim T}[PXX^\top P]$ is equal to identity.

**Lemma D.4** (Bounded fourth-moment and second moment imply closeness of covariance). *Let $T$ be a set of $n$ points in $\mathbb{R}^d$. Let $P$ be an arbitrary symmetric matrix. Let $w \in \mathbb{R}^{d^2}$ be the leading eigenvector of the $\mathbf{Cov}_{X \sim T}[(PX)^{\otimes 2}]$ with $\|w\|_2 = 1$, and let $A \in \mathbb{R}^{d \times d}$ be $w^\sharp$. Note that $\|w\|_2 = 1$ implies $\|A\|_{\text{F}} = 1$. For every subset $T' \subseteq T$ with $|T'| \geq (\alpha/2)|T|$, we have that*

$$\left\| \mathbf{E}_{X \sim T'}[PXX^\top P] - \mathbf{E}_{x \sim T}[PXX^\top P] \right\|_{\text{F}}^2 \leq \frac{2}{\alpha} \mathbf{Var}_{X \sim T}[X^\top PAPX] \quad \text{and}$$

$$\left\| \mathbf{Cov}_{X \sim T'}[PX] - \mathbf{E}_{x \sim T}[PXX^\top P] \right\|_{\text{F}}^2 \leq \frac{4}{\alpha} \mathbf{Var}_{X \sim T}[X^\top PAPX] + \frac{8}{\alpha^2} \left\| \mathbf{E}_{X \sim T}[PXX^\top P] \right\|_{\text{op}}^2 .$$

*In particular, suppose that there exists a set $S$ that satisfies Conditions (L.1) and (L.2) of $(\eta, \epsilon)$-stability (Definition 2.1) with respect to $\mu$ and $\Sigma$ for some $\eta$ and $\epsilon$, and suppose that $|S \cap T'| \geq \min(\alpha|T|/2, (1 - \epsilon)|S|)$. Then*

$$\left\| P\Sigma P - \mathbf{E}_{X \sim T'}[PXX^\top P] \right\|_{\text{F}} \lesssim \frac{1}{\alpha} \mathbf{Var}_{X \sim T}[X^\top PAPX] + \frac{1}{\alpha^2} \left\| \mathbf{E}_{X \sim T}[PXX^\top P] \right\|_{\text{op}}^2 .$$

*Proof.* Let $\widetilde{T}$ define the set $\{PX : X \in T\}$. In this notation, $w$ is the leading eigenvector of $\mathbf{Cov}_{\widetilde{x} \sim \widetilde{T}}[\widetilde{X}^{\otimes 2}]$ with $\|w\|_2 = 1$, and $A = w^\sharp$. Then, for any $\widetilde{T}' \subseteq \widetilde{T}$ with $|\widetilde{T}'| \geq \alpha|\widetilde{T}'|/2$, we have the following

$$\mathbf{Var}_{\widetilde{X} \sim \widetilde{T}}[\widetilde{X}^\top A \widetilde{X}] = \sup_{v \in \mathbb{R}^{d^2}, \|v\|_2^2 = 1} \mathbf{E}_{\widetilde{X} \sim \widetilde{T}} \left[ \left\langle \widetilde{X}^{\otimes 2} - \mathbf{E}_{\widetilde{Y} \sim \widetilde{T}}[\widetilde{Y}^{\otimes 2}], v \right\rangle^2 \right]$$

$$= \sup_{v \in \mathbb{R}^{d^2}, \|v\|_2^2 = 1} \frac{|\widetilde{T}'|}{|\widetilde{T}|} \mathbf{E}_{\widetilde{X} \sim \widetilde{T}'} \left[ \left\langle \widetilde{X}^{\otimes 2} - \mathbf{E}_{\widetilde{Y} \sim \widetilde{T}}[\widetilde{Y}^{\otimes 2}], v \right\rangle^2 \right]$$

$$+ \frac{|\widetilde{T} \setminus \widetilde{T}'|}{|\widetilde{T}|} \mathop{\mathbf{E}}_{\widetilde{X} \sim \widetilde{T} \setminus \widetilde{T}'} \left[ \left\langle \widetilde{X}^{\otimes 2} - \mathop{\mathbf{E}}_{\widetilde{Y} \sim \widetilde{T}} [\widetilde{Y}^{\otimes 2}], v \right\rangle^2 \right]$$

$$\geq \frac{\alpha}{2} \sup_{v \in \mathbb{R}^{d^2}, \|v\|_2^2 = 1} \mathop{\mathbf{E}}_{\widetilde{X} \sim \widetilde{T}'} \left[ \left\langle \widetilde{X}^{\otimes 2} - \mathop{\mathbf{E}}_{\widetilde{Y} \sim \widetilde{T}} [\widetilde{Y}^{\otimes 2}], v \right\rangle^2 \right] \qquad \text{(since } |\widetilde{T}'| \geq (\alpha/2)|\widetilde{T}|\text{)}$$

$$\geq \frac{\alpha}{2} \sup_{v \in \mathbb{R}^{d^2}, \|v\|_2^2 = 1} \left\langle \mathop{\mathbf{E}}_{\widetilde{X} \sim \widetilde{T}'} [\widetilde{X}^{\otimes 2}] - \mathop{\mathbf{E}}_{\widetilde{Y} \sim \widetilde{T}} [\widetilde{Y}^{\otimes 2}], v \right\rangle^2 \quad (\mathbf{E}[Y^2] \geq (\mathbf{E}[Y])^2 \text{ for any } Y)$$

$$= \frac{\alpha}{2} \sup_{J \in \mathbb{R}^{d \times d}, \|J\|_F^2 = 1} \left[ \left\langle \mathop{\mathbf{E}}_{\widetilde{X} \sim \widetilde{T}'} [\widetilde{X} \widetilde{X}^\top] - \mathop{\mathbf{E}}_{\widetilde{Y} \sim \widetilde{T}} [\widetilde{Y} \widetilde{Y}^\top], J \right\rangle^2 \right]$$

$$= \frac{\alpha}{2} \left\| \mathop{\mathbf{E}}_{\widetilde{X} \sim \widetilde{T}'} [\widetilde{X} \widetilde{X}^\top] - \mathop{\mathbf{E}}_{\widetilde{X} \sim \widetilde{T}} [\widetilde{X} \widetilde{X}^\top] \right\|_F^2,$$

where the penultimate step above holds by lifting the flattened matrix inner products in the previous line to the matrix inner product, and the last line uses the variation characterization of the Frobenius norm. This implies the first inequality since $\mathbf{E}_{\widetilde{X} \sim \widetilde{T}} [\widetilde{X} \widetilde{X}^\top] = \mathbf{E}_{X \sim T} [P X X^\top P]$.

We will now establish the second inequality. Using the relation between the second moment matrix and the covariance, we have the following:

$$\left\| \mathop{\mathbf{Cov}}_{\widetilde{X} \sim \widetilde{T}'} [\widetilde{X}] - \mathop{\mathbf{E}}_{\widetilde{X} \sim \widetilde{T}} [\widetilde{X} \widetilde{X}^\top] \right\|_F^2 \quad = \left\| \mathop{\mathbf{E}}_{\widetilde{X} \sim \widetilde{T}'} [\widetilde{X} \widetilde{X}^\top] - \mathop{\mathbf{E}}_{\widetilde{X} \sim \widetilde{T}'} [\widetilde{X}] \mathop{\mathbf{E}}_{\widetilde{X} \sim \widetilde{T}'} [\widetilde{X}^\top] - \mathop{\mathbf{E}}_{\widetilde{X} \sim \widetilde{T}} [\widetilde{X} \widetilde{X}^\top] \right\|_F^2$$

$$\leq 2 \left\| \mathop{\mathbf{E}}_{\widetilde{X} \sim \widetilde{T}'} [\widetilde{X} \widetilde{X}^\top] - \mathop{\mathbf{E}}_{\widetilde{X} \sim \widetilde{T}} [\widetilde{X} \widetilde{X}^\top] \right\|_F^2 + 2 \left\| \mathop{\mathbf{E}}_{\widetilde{X} \sim \widetilde{T}'} [\widetilde{X}] \mathop{\mathbf{E}}_{\widetilde{X} \sim \widetilde{T}'} [\widetilde{X}^\top] \right\|_F^2$$

$$\leq \frac{4}{\alpha} \mathop{\mathbf{Var}}_{\widetilde{X} \sim \widetilde{T}} [\widetilde{X}^\top A \widetilde{X}] + 2 \left\| \mathop{\mathbf{E}}_{\widetilde{X} \sim \widetilde{T}'} [\widetilde{X}] \mathop{\mathbf{E}}_{\widetilde{X} \sim \widetilde{T}'} [\widetilde{X}^\top] \right\|_{op}^2,$$

where the last step uses that $\mathbf{E}_{\widetilde{X} \sim \widetilde{T}'} [\widetilde{X}] \mathbf{E}_{\widetilde{X} \sim \widetilde{T}'} [\widetilde{X}^\top]$ has rank one and thus its Frobenius and operator norm match. Thus to establish the desired inequality, it suffices to prove that $\mathbf{E}_{\widetilde{X} \sim \widetilde{T}'} [\widetilde{X}] \mathbf{E}_{\widetilde{X} \sim \widetilde{T}'} [\widetilde{X}^\top] \preceq (2/\alpha) \cdot \mathbf{E}_{\widetilde{X} \sim \widetilde{T}} [\widetilde{X} \widetilde{X}^\top]$. Indeed, we have that

$$\mathop{\mathbf{E}}_{\widetilde{X} \sim \widetilde{T}'} [\widetilde{X}] \mathop{\mathbf{E}}_{\widetilde{X} \sim \widetilde{T}'} [\widetilde{X}^\top] \preceq \mathop{\mathbf{E}}_{\widetilde{X} \sim \widetilde{T}'} [\widetilde{X} \widetilde{X}^\top] = \frac{1}{|\widetilde{T}'|} \sum_{\widetilde{X} \in \widetilde{T}'} \widetilde{X} \widetilde{X}^\top = \frac{|\widetilde{T}|}{|\widetilde{T}'|} \frac{1}{|\widetilde{T}|} \sum_{\widetilde{X} \in \widetilde{T}'} \widetilde{X} \widetilde{X}^\top$$

$$\preceq \frac{2}{\alpha} \frac{1}{|\widetilde{T}|} \sum_{\widetilde{X} \in \widetilde{T}'} \widetilde{X} \widetilde{X}^\top \preceq \frac{2}{\alpha} \frac{1}{|\widetilde{T}|} \sum_{\widetilde{X} \in \widetilde{T}} \widetilde{X} \widetilde{X}^\top,$$

where the last expression is in fact equal to $(2/\alpha) \cdot \mathbf{E}_{\widetilde{X} \sim \widetilde{T}} [\widetilde{X} \widetilde{X}^\top]$. This completes the proof for the second statement.

We now consider the case when $T$ contains a large stable subset. Let $T' = T \cap S$, which satisfies that $|T'| \geq (\alpha/2)|T|$ and $|T'| \geq (1 - \epsilon)|S|$. By applying Lemma D.3 with $L = \mathbf{E}_{X \sim T} [P X X^\top P]$ to the second inequality in the statement with $T'$, we obtain the desired result.

$$\square$$

## D.4 Proofs of Corollary D.2 and lemma D.3

**Corollary D.2** (Stability of Transformed Sets). *Let $D$ be a distribution over $\mathbb{R}^d$ with mean $\mu$ and covariance $\Sigma$. Let $A$ be a set of points in $\mathbb{R}^d$ that satisfies Conditions (L.1) and (L.2) of Definition 2.1 of $(\eta, \epsilon)$-stability with respect to $D$. Then we have that for any symmetric transformation $P \in \mathbb{R}^{d \times d}$ and every $A' \subset A$ with $|A'| \geq (1 - \epsilon)|A|$, the following holds:*

1. $\left\| P \left( \frac{1}{|A'|} \sum_{x \in A'} x - \mu \right) \right\|_2 \leq 0.1 \sqrt{\|P \Sigma P\|_{op}}.$

2. $\left\| P \left( \frac{1}{|A'|} \sum_{x \in A'} (x - \mu)(x - \mu)^\top - \Sigma \right) P \right\|_F \leq 0.1 \|P \Sigma P\|_{op}.$

*Proof.* We explain each condition separately. We begin with the first condition as follows: For any unit vector $u$ in $\mathbb{R}^d$, we have that

$$\left| u^\top P \left( \frac{1}{|A'|} \sum_{x \in A'} x - \mu \right) \right| \leq 0.1 \sqrt{u^\top P \Sigma P u} \leq \sqrt{\|P \Sigma P\|_{\text{op}}} \,,$$

where the first inequality uses Condition (L.1) of Definition 2.1 for $v = Pu$.

We now focus our attention to the second condition. Denote $\hat{\Sigma} := \frac{1}{|A'|} \sum_{x \in A'} (x - \mu)(x - \mu)^\top$ for saving space. Let $Q$ be any symmetric matrix $Q \in \mathbb{R}^{d \times d}$ with $\|Q\|_F \leq 1$. Using the cyclical property of trace, we obtain

$$\text{tr} \left( P \left( \hat{\Sigma} - \Sigma \right) PQ \right) = \text{tr} \left( \left( \hat{\Sigma} - \Sigma \right) PQP \right) \leq 0.1 \left\| \Sigma^{1/2} PQP \Sigma^{1/2} \right\|_F \,,$$

where the last step applies Condition (L.2) of Definition 2.1 with $V = PQP$. Finally, using the cyclic properties of the trace operator,

$$\left\| \Sigma^{1/2} PQP \Sigma^{1/2} \right\|_F^2 = \text{tr} \left( \Sigma^{1/2} PQP \Sigma PQP \Sigma^{1/2} \right) = \text{tr}(QP \Sigma PQP \Sigma P)$$
$$= \|QP \Sigma P\|_F^2 \leq \|Q\|_F^2 \|P \Sigma P\|_{\text{op}}^2 \leq \|P \Sigma P\|_{\text{op}}^2,$$

where the first inequality uses Fact B.3 and the second inequality uses $\|Q\|_F \leq 1$. Thus we have the following:

$$\left\| P \left( \hat{\Sigma} - \Sigma \right) P \right\|_F = \sup_{Q : \|Q\|_F \leq 1} \text{tr} \left( P \left( \hat{\Sigma} - \Sigma \right) PQ \right)$$
$$= \sup_{\text{symmetric } Q : \|Q\|_F \leq 1} \text{tr} \left( P \left( \hat{\Sigma} - \Sigma \right) PQ \right)$$
$$\leq 0.1 \|P \Sigma P\|_{\text{op}}.$$

where the second equality is due to the fact that $P \left( \hat{\Sigma} - \Sigma \right) P$ is itself symmetric.

$\square$

**Lemma D.3.** *Let $A$ be a set of points in $\mathbb{R}^d$ satisfying Conditions (L.1) and (L.2) of $(\eta, \epsilon)$-stability (Definition 2.1) with respect to $\mu, \Sigma$. Let $A' \subseteq A$ such that $|A'| \geq (1 - \epsilon)|A|$. If $P, L \in \mathbb{R}^{d \times d}$ are symmetric matrices with the property that $\| \text{Cov}_{X \sim A'}[PX] - L \|_F \leq r$, then*
$$\|P \Sigma P - L\|_F = O(r + \|L\|_{\text{op}}) \,.$$

*Proof.* To simplify notation, let $\widehat{\Sigma} := \text{Cov}_{X \sim A'}[X]$ and $\overline{\Sigma} := \mathbb{E}_{X \sim A'}[(X - \mu)(X - \mu)^\top]$. Observe that $\overline{\Sigma} = \widehat{\Sigma} + (\mu - \hat{\mu})(\mu - \hat{\mu})^\top$. We apply triangle inequality and Corollary D.2 with to obtain the following:

$$\|P(\widehat{\Sigma} - \Sigma)P\|_F \leq \|P(\overline{\Sigma} - \Sigma)P\|_F + \|P(\mu - \hat{\mu})(\mu - \hat{\mu})^\top P\|_F$$
$$= \|P(\overline{\Sigma} - \Sigma)P\|_F + \|P(\mu - \hat{\mu})\|_2^2$$
$$\leq 0.1 \|P \Sigma P\|_{\text{op}} + 0.1 \|P \Sigma P\|_{\text{op}}$$
$$\leq 0.2 \|P \Sigma P\|_{\text{op}} \,, \tag{6}$$

where the first inequality follows from the triangle inequality and the second inequality uses Corollary D.2.

We now analyze the desired expression by using the triangle inequality and the lemma assumption as follows:

$$\|P \Sigma P - L\|_F \leq \|P \widehat{\Sigma} P - L\|_F + \|P(\widehat{\Sigma} - \Sigma)P\|_F \leq r + 0.2 \|P \Sigma P\|_{\text{op}},$$

where the first term was bounded by $r$ by assumption and the second term uses the bound (6). Thus it suffices to show that $\|P \Sigma P\|_{\text{op}} = O(r + \|L\|_{\text{op}})$. To this end, we again use triangle inequality as follows:

$$\|P \Sigma P\|_{\text{op}} \leq \|P \widehat{\Sigma} P - L\|_{\text{op}} + \|L\|_{\text{op}} + \|P(\widehat{\Sigma} - \Sigma)P\|_{\text{op}}$$
$$\leq r + \|L\|_{\text{op}} + 0.2 \|P \Sigma P\|_{\text{op}}, \tag{7}$$

where the second inequality uses the bound (6). Rearranging the above display inequality, we obtain $\|P \Sigma P\|_{\text{op}} = O(r + \|L\|_{\text{op}})$.

$\square$

# E   Analysis of a Single Recursive Call of the Algorithm: Proofs from Section 3

In this section, we present the details that were omitted from Section 3. We recall Assumption 3.1 from the main body.

---

**Assumption 3.1** (Assumptions and notations for a single recursive call of the algorithm).

- $S = \{x_i\}_{i=1}^n$ is a set of $n$ uncontaminated samples, which is assumed to be $(\eta, 2\epsilon_0)$-stable with respect to the inlier distribution $D$ having mean and covariance $\mu, \Sigma$ (c.f. Definition 2.1). We assume $\eta \leq 0.001$, $\epsilon_0 = 0.01$, and $\mathbf{Var}_{X \sim D}[X^\top A X] \leq C_1(\|\Sigma^{1/2} A \Sigma^{1/2}\|_F^2 + \|\Sigma^{1/2} A \mu\|_2^2)$ for all symmetric $d \times d$ matrices $A$ and a constant $C_1$.

- $T$ is the input set to the current recursive call of the algorithm (after the adversarial corruptions), which satisfies $|S \cap T| \geq (1 - 2\epsilon_0)|S|$ and $|T| \leq (1/\alpha)|S|$.

- We denote $H = \mathbf{E}_{X \sim T}\left[X X^\top\right]$.

- We denote by $\widetilde{S}, \widetilde{T}$ the versions of $S$ and $T$ normalized by $H^{\dagger/2}$: $\widetilde{S} = \{H^{\dagger/2} x : x \in S\}$ and $\widetilde{T} = \{H^{\dagger/2} x : x \in T\}$. We use the notation $\widetilde{x}$ for elements in $\widetilde{S}$ and $\widetilde{T}$, and $x$ for elements in $S$ and $T$. Similarly, we use the notation $\widetilde{X}$ for random variables with support in $\widetilde{S}$ or $\widetilde{T}$.

- The mean and covariance of the inlier distribution $D$ after transformation with $H^{\dagger/2}$ are denoted by $\widetilde{\mu} := H^{\dagger/2}\mu$, $\widetilde{\Sigma} := H^{\dagger/2}\Sigma H^{\dagger/2}$. We denote the empirical mean and covariance of the transformed inliers in $T$ by $\hat{\mu} := \mathbf{E}_{\widetilde{X} \sim \widetilde{S} \cap \widetilde{T}}[\widetilde{X}]$ and $\hat{\Sigma} := \mathbf{Cov}_{\widetilde{X} \sim \widetilde{S} \cap \widetilde{T}}[\widetilde{X}]$.

---

## E.1   Normalization and Proof of Lemma 3.2

We start with a simple result stating that after normalization with $H^{\dagger/2}$, the mean and covariance of the inliers, both empirical and population-level, are bounded. This is intuitive since the inliers constitute $\Omega(\alpha)$-fraction of the overall samples and the second moment of the complete set after normalization is bounded by identity.

**Lemma E.1** (Normalization). *Make Assumption 3.1 and recall the notations $\widetilde{\mu}, \widetilde{\Sigma}, \hat{\mu}, \hat{\Sigma}$ that were defined in Assumption 3.1. We have that:*

1. *$\|\hat{\Sigma}\|_{\mathrm{op}} \leq 2/\alpha$.*

2. *$\|\hat{\mu}\|_2 \leq \sqrt{2/\alpha}$.*

3. *$\|\widetilde{\Sigma}\|_{\mathrm{op}} \leq 3/\alpha$*

4. *$\|\widetilde{\mu}\|_2 \leq \sqrt{3/\alpha}$*

5. *$\| \mathbf{E}_{\widetilde{X} \sim \widetilde{S}}[\widetilde{X}]\|_2 \leq 2/\sqrt{\alpha}$ and $\| \mathbf{Cov}_{\widetilde{X} \sim \widetilde{S}}[\widetilde{X}]\|_{\mathrm{op}} \leq 4/\alpha$.*

6. *For every matrix $A$ with $\|A\|_F \leq 1$, $| \mathbf{E}_{\widetilde{X} \sim \widetilde{S}}[\widetilde{X}^\top A \widetilde{X}] - \mathbf{E}_{X \sim D}[(H^{\dagger/2} X)^\top A (H^{\dagger/2} X)]| < 1/\alpha$.*

*Proof.* We prove each part below:

**Establishing Item 1** The transformation $H$ is such that $\mathbf{E}_{\widetilde{X} \sim \widetilde{T}}[\widetilde{X}\widetilde{X}^\top] = \mathbf{E}_{X \sim T}[H^{\dagger/2} X X^\top H^{\dagger/2}] = H^{\dagger/2} H H^{\dagger/2} = \mathbb{P}_H$, which has operator norm at most 1. By assumption, we have that $\left|\widetilde{S} \cap \widetilde{T}\right| \geq (1 - 2\epsilon_0)\alpha|T| \geq 0.5\alpha|T|$. Thus, we obtain the following inequality:

$$\frac{1}{\left|\widetilde{S} \cap \widetilde{T}\right|} \sum_{\widetilde{x} \in \widetilde{S} \cap \widetilde{T}} \widetilde{x}\widetilde{x}^\top \preceq \frac{1}{\left|\widetilde{S} \cap \widetilde{T}\right|} \sum_{\widetilde{x} \in \widetilde{T}} \widetilde{x}\widetilde{x}^\top = \frac{|\widetilde{T}|}{\left|\widetilde{S} \cap \widetilde{T}\right|}\mathbb{P}_H \preceq \frac{2}{\alpha}\mathbb{P}_H \ . \tag{8}$$

By applying (8), we obtain

$$\hat{\Sigma} = \frac{1}{\left|\widetilde{S} \cap \widetilde{T}\right|} \sum_{\widetilde{x} \in \widetilde{S} \cap \widetilde{T}} (\widetilde{x} - \hat{\mu})(\widetilde{x} - \hat{\mu})^{\top} \preceq \frac{1}{\left|\widetilde{S} \cap \widetilde{T}\right|} \sum_{\widetilde{x} \in \widetilde{S} \cap \widetilde{T}} \widetilde{x}\widetilde{x}^{\top} \preceq \frac{2}{\alpha} \mathbb{P}_H,$$

implying that $\|\hat{\Sigma}\|_{\mathrm{op}} \le 2/\alpha$, since $\mathbb{P}_H$ is just a projection matrix with $0/1$ eigenvalues.

**Establishing Item 2**    Since, for any random vector $X$, we have that $\mathbf{E}[X]\,\mathbf{E}[X]^{\top} \preceq \mathbf{E}[XX^{\top}]$ (which is equivalent to $\mathbf{Cov}[X] \succeq 0$), we obtain

$$\hat{\mu}\hat{\mu}^{\top} \preceq \frac{1}{\left|\widetilde{S} \cap \widetilde{T}\right|} \sum_{\widetilde{x} \in \widetilde{S} \cap \widetilde{T}} \widetilde{x}\widetilde{x}^{\top} \preceq \frac{2}{\alpha} \mathbb{P}_H,$$

where the last inequality uses (8). This implies that $\|\hat{\mu}\|_2^2 \le 2/\alpha$.

**Establishing Item 3**    The goal is to bound the population-level covariance $\|\widetilde{\Sigma}\|_{\mathrm{op}}$. To this end, we will use the bounds from Items 1 and 2 which bounds their empirical versions and relate the empirical versions to the population ones via the deterministic conditions.

Consider an arbitrary subset $\widetilde{S}_1$ of $\widetilde{S}$ satisfying $\left|\widetilde{S}_1\right| \ge \left|\widetilde{S}\right|(1 - 2\epsilon_0)$. We first note that by Corollary D.2, we have that

$$\left\|\widetilde{\Sigma} - \mathbf{E}_{\widetilde{X} \sim \widetilde{S}_1}[(\widetilde{X} - \widetilde{\mu})(\widetilde{X} - \widetilde{\mu})^{\top}]\right\|_{\mathrm{F}} \le 0.1 \left\|\widetilde{\Sigma}\right\|_{\mathrm{op}} \quad \text{and} \quad \left\|\widetilde{\mu} - \mathbf{E}_{\widetilde{X} \sim \widetilde{S}_1}[\widetilde{X}]\right\|_2 \le 0.1\sqrt{\left\|\widetilde{\Sigma}\right\|_{\mathrm{op}}}. \quad (9)$$

Now define $\widetilde{\Sigma}_1 := \mathbf{E}_{\widetilde{X} \sim \widetilde{S} \cap \widetilde{T}}[(\widetilde{X} - \widetilde{\mu})(\widetilde{X} - \widetilde{\mu})^{\top}]$, the centered second moment matrix of $\widetilde{S} \cap \widetilde{T}$, which satisfies $\widetilde{\Sigma}_1 = \hat{\Sigma} + (\widetilde{\mu} - \hat{\mu})(\widetilde{\mu} - \hat{\mu})^{\top}$. We have that

$$
\begin{aligned}
\|\widetilde{\Sigma}\|_{\mathrm{op}} &\le \|\widetilde{\Sigma} - \hat{\Sigma}\|_{\mathrm{F}} + \|\hat{\Sigma}\|_{\mathrm{op}} && \text{(triangle inequality)} \\
&\le \|\widetilde{\Sigma} - \widetilde{\Sigma}_1\|_{\mathrm{F}} + \|\widetilde{\mu} - \hat{\mu}\|_2^2 + \|\hat{\Sigma}\|_{\mathrm{op}} && \text{(triangle inequality and } \widetilde{\Sigma}_1 = \hat{\Sigma} + (\mu - \hat{\mu})(\mu - \hat{\mu})^{\top}) \\
&\le 0.2\|\widetilde{\Sigma}\|_{\mathrm{op}} + \|\hat{\Sigma}\|_{\mathrm{op}},
\end{aligned}
$$

where in the last line we use (9) for $\widetilde{S}_1 = \widetilde{S} \cap \widetilde{T}$ and the fact that $\left|\widetilde{S} \cap \widetilde{T}\right| \ge (1 - 2\epsilon_0)|\widetilde{S}|$. Rearranging, we have that $\|\widetilde{\Sigma}\|_{\mathrm{op}} < 1.25\|\hat{\Sigma}\|_{\mathrm{op}}$ for which we can use Item 1 to further upper bound it by $3/\alpha$.

**Establishing Item 4**    We use a similar argument as in Item 3:

$$\|\widetilde{\mu}\|_2 \le \|\hat{\mu}\|_2 + \|\widetilde{\mu} - \hat{\mu}\|_2 \le \|\hat{\mu}\|_2 + 0.1\sqrt{\|\widetilde{\Sigma}\|_{\mathrm{op}}} \le \sqrt{3/\alpha}, \quad (10)$$

where the first step uses the triangle inequality, the second inequality uses Equation (9) for $\widetilde{S}_1 = \widetilde{S} \cap \widetilde{T}$, and the last inequality uses Items 2 and 3.

**Establishing Item 5**    For the covariance condition, we have the following:

$$
\begin{aligned}
\left\|\mathbf{Cov}_{\widetilde{X} \sim \widetilde{S}}[\widetilde{X}]\right\|_{\mathrm{op}} &\le \left\|\mathbf{E}_{\widetilde{X} \sim \widetilde{S}}\left[(\widetilde{X} - \widetilde{\mu})(\widetilde{X} - \widetilde{\mu})^{\top}\right]\right\|_{\mathrm{op}} && \text{(using } \mathbf{Cov}(\widetilde{X}) \preceq \mathbf{E}[\widetilde{X}\widetilde{X}^{\top}]) \\
&\le \left\|\mathbf{E}_{\widetilde{X} \sim \widetilde{S}}\left[(\widetilde{X} - \widetilde{\mu})(\widetilde{X} - \widetilde{\mu})^{\top}\right] - \widetilde{\Sigma}\right\|_{\mathrm{op}} + \|\widetilde{\Sigma}\|_{\mathrm{op}} && \text{(triangle inequality)} \\
&\le 1.1\|\widetilde{\Sigma}\|_{\mathrm{op}},
\end{aligned}
$$

where the last step upper bounds the first term using Equation (9) for $\widetilde{S}_1 = \widetilde{S}$, which trivially satisfies $|\widetilde{S}| \ge |\widetilde{S}|(1 - 2\epsilon_0)$. The overall expression is upper bounded by $(1.1 \times 3)/\alpha$ by Item 3.

The mean condition has a similar proof:

$$\left\|\mathbf{E}_{\widetilde{X} \sim \widetilde{S}}[\widetilde{X}]\right\|_2 \le \left\|\mathbf{E}_{\widetilde{X} \sim \widetilde{S}}[\widetilde{X}] - \widetilde{\mu}\right\|_2 + \|\widetilde{\mu}\|_2 \le 0.1\sqrt{\|\widetilde{\Sigma}\|_{\mathrm{op}}} + \sqrt{3/\alpha} \le 2/\sqrt{\alpha},$$

where we use Equations (9) and (10).

**Establishing Item 6** Consider an arbitrary square matrix $A$ with $\|A\|_F \leq 1$. We have that

$$
\left| \mathop{\mathbf{E}}_{\widetilde{X} \sim \widetilde{S}} [\widetilde{X}^\top A \widetilde{X}] - \mathop{\mathbf{E}}_{X \sim D} [(H^{\dagger/2} X)^\top A (H^{\dagger/2} X)] \right|
$$

$$
= \left| \left\langle A, \mathop{\mathbf{E}}_{\widetilde{X} \sim \widetilde{S}} [\widetilde{X} \widetilde{X}^\top] - \mathop{\mathbf{E}}_{X \sim D} [(H^{\dagger/2} X)(H^{\dagger/2} X)^\top] \right\rangle \right|
$$

$$
= \left| \left\langle A, \left( \mathop{\mathbf{Cov}}_{\widetilde{X} \sim \widetilde{S}} [\widetilde{X}] + \mathop{\mathbf{E}}_{\widetilde{X} \sim \widetilde{S}} [\widetilde{X}] \mathop{\mathbf{E}}_{\widetilde{X} \sim \widetilde{S}} [\widetilde{X}]^\top \right) - \left( \widetilde{\Sigma} + \widetilde{\mu} \widetilde{\mu}^\top \right) \right\rangle \right|
$$

$$
\leq \left| \left\langle A, \mathop{\mathbf{Cov}}_{X \sim \widetilde{S}} [X] - \widetilde{\Sigma} \right\rangle \right| + \left| \left\langle A, \mathop{\mathbf{E}}_{X \sim \widetilde{S}} [X] \mathop{\mathbf{E}}_{X \sim \widetilde{S}} [X]^\top - \widetilde{\mu} \widetilde{\mu}^\top \right\rangle \right| \qquad \text{(triangle inequality)}
$$

$$
\leq \left\| \mathop{\mathbf{Cov}}_{\widetilde{X} \sim \widetilde{S}} [\widetilde{X}] - \widetilde{\Sigma} \right\|_F + \left\| \mathop{\mathbf{E}}_{\widetilde{X} \sim \widetilde{S}} [\widetilde{X}] \mathop{\mathbf{E}}_{\widetilde{X} \sim \widetilde{S}} [\widetilde{X}]^\top - \widetilde{\mu} \widetilde{\mu}^\top \right\|_F \quad \text{(variational definition of Frobenius norm)}
$$

$$
= \left\| \mathop{\mathbf{E}}_{\widetilde{X} \sim \widetilde{S}} [(\widetilde{X} - \widetilde{\mu})(\widetilde{X} - \widetilde{\mu})^\top] - \widetilde{\Sigma} + \left( \mathop{\mathbf{E}}_{\widetilde{X} \sim \widetilde{S}} [\widetilde{X}] - \widetilde{\mu} \right) \left( \mathop{\mathbf{E}}_{\widetilde{X} \sim \widetilde{S}} [\widetilde{X}] - \widetilde{\mu} \right)^\top \right\|_F
$$

$$
+ \left\| \mathop{\mathbf{E}}_{\widetilde{X} \sim \widetilde{S}} [\widetilde{X}] \mathop{\mathbf{E}}_{\widetilde{X} \sim \widetilde{S}} [\widetilde{X}]^\top - \widetilde{\mu} \widetilde{\mu}^\top \right\|_F
$$

$$
\leq \left\| \mathop{\mathbf{E}}_{\widetilde{X} \sim \widetilde{S}} [(\widetilde{X} - \widetilde{\mu})(\widetilde{X} - \widetilde{\mu})^\top] - \widetilde{\Sigma} \right\|_F + \left\| \left( \mathop{\mathbf{E}}_{\widetilde{X} \sim \widetilde{S}} [\widetilde{X}] - \widetilde{\mu} \right) \left( \mathop{\mathbf{E}}_{\widetilde{X} \sim \widetilde{S}} [\widetilde{X}] - \widetilde{\mu} \right)^\top \right\|_F
$$

$$
+ \left\| \mathop{\mathbf{E}}_{\widetilde{X} \sim \widetilde{S}} [\widetilde{X}] \mathop{\mathbf{E}}_{\widetilde{X} \sim \widetilde{S}} [\widetilde{X}]^\top - \widetilde{\mu} \widetilde{\mu}^\top \right\|_F
$$

$$
\overset{(a)}{\leq} \left\| \mathop{\mathbf{E}}_{X \sim \widetilde{S}} [(X - \widetilde{\mu})(X - \widetilde{\mu})^\top] - \widetilde{\Sigma} \right\|_F + 3 \left\| \mathop{\mathbf{E}}_{X \sim \widetilde{S}} [X] - \widetilde{\mu} \right\|_2^2
$$

$$
\leq 0.1 \|\widetilde{\Sigma}\|_{op} + 0.03 \|\widetilde{\Sigma}\|_{op} \qquad\qquad\qquad\qquad\qquad\qquad\qquad\qquad \text{(using (9))}
$$

$$
< 1/\alpha \,, \qquad\qquad\qquad\qquad\qquad\qquad\qquad\qquad\qquad\qquad\qquad\qquad\quad \text{(using Item 3)}
$$

where the inequality (a) uses the fact that $\|uv^\top - wz^\top\|_F^2 \leq \|u - w\|_2^2 + \|v - z\|_2^2$ for the last term. $\qquad\square$

The previous result implies that the mean and covariance of the inlier distribution $D$ after transformation with $H^{\dagger/2}$ is bounded. Since the variance of degree-two polynomials under $D$ is bounded by assumption, we obtain the following bound on the variance:

**Lemma 3.2.** *Make Assumption 3.1 and recall that $H = \mathbf{E}_{X \sim T}[XX^\top]$, where $T$ is the corrupted version of a stable inlier set $S$. For every symmetric matrix $A$ with $\|A\|_F = 1$, we have that $\mathbf{Var}_{X \sim D}[(H^{\dagger/2} X)^\top A (H^{\dagger/2} X)] \leq 18 C_1 / \alpha^2$.*

*Proof.* By Assumption 3.1, we have that $\mathbf{Var}_{X \sim D}[X^\top B X] \leq C_1 (\|\Sigma^{1/2} B \Sigma^{1/2}\|_F^2 + \|\Sigma^{1/2} B \mu\|_2^2)$ for all $B \in \mathbb{R}^{d \times d}$. Applying this to $B = H^{\dagger/2} A H^{\dagger/2}$, we have that

$$
\mathop{\mathbf{Var}}_{X \sim D} [(H^{\dagger/2} X)^\top A (H^{\dagger/2} X)] \leq C_1 (\|\Sigma^{1/2} H^{\dagger/2} A H^{\dagger/2} \Sigma^{1/2}\|_F^2 + \|\Sigma^{1/2} H^{\dagger/2} A H^{\dagger/2} \mu\|_2^2)
$$

$$
= C_1 (\|\widetilde{\Sigma}^{1/2} A \widetilde{\Sigma}^{1/2}\|_F^2 + \|\widetilde{\Sigma}^{1/2} A \widetilde{\mu}\|_2^2) \,, \tag{11}
$$

where we use that $\|\widetilde{\Sigma}^{1/2} A \widetilde{\Sigma}^{1/2}\|_F^2 = \|\Sigma^{1/2} H^{\dagger/2} A H^{\dagger/2} \Sigma^{1/2}\|_F^2$ and $\|\Sigma^{1/2} H^{\dagger/2} A H^{\dagger/2} \mu\|_2^2 = \|\widetilde{\Sigma}^{1/2} A \widetilde{\mu}\|_2^2$ since $\widetilde{\Sigma} = H^{\dagger/2} \Sigma H^{\dagger/2}$ and $\widetilde{\mu} = H^{\dagger/2} \mu$. By Lemma E.1, $\|\widetilde{\Sigma}\|_{op} \leq 3/\alpha$. Since $\|A\|_F = 1$, we have that the first term in (11) is bounded as follows: $\|\widetilde{\Sigma}^{1/2} A \widetilde{\Sigma}^{1/2}\|_F^2 \leq \|\widetilde{\Sigma}\|_{op}^2 \|A\|_F^2 \leq 9/\alpha^2$, using the fact that $\|AB\|_F \leq \|A\|_{op} \|B\|_F$ (Fact B.3). We also have that $\|\widetilde{\Sigma}^{1/2} A \widetilde{\mu}\|_2^2 \leq \|\widetilde{\Sigma}\|_{op} \|A\|_{op}^2 \|\widetilde{\mu}\|_2^2 \leq 9/\alpha^2$, where we the bounds from Lemma E.1. The desired conclusion then follows by applying these upper bounds in (11). $\qquad\square$

## E.2 Certificate Lemma: Proof of Lemma 3.4

**Lemma E.2** (Case when we stop and return; Formal version of Lemma 3.4). *Make Assumption 3.1. Let $w \in \mathbb{R}^{d^2}$ be the leading eigenvector of the $\mathbf{Cov}_{\widetilde{X} \sim \widetilde{T}}[\widetilde{X}^{\otimes 2}]$ with $\|w\|_2 = 1$, and let $A \in \mathbb{R}^{d \times d}$ be $w^\sharp$. Note that $\|w\|_2 = 1$ implies $\|A\|_F = 1$. For every subset $\widetilde{T}' \subseteq \widetilde{T}$ with $|\widetilde{T}'| \geq (\alpha/2)|\widetilde{T}|$, we have that*

$$\left\| \mathbf{Cov}_{\widetilde{X} \sim \widetilde{T}'}[\widetilde{X}] - \mathbb{P}_H \right\|_F^2 \lesssim \frac{1}{\alpha} \mathbf{Var}_{\widetilde{X} \sim \widetilde{T}}[\widetilde{X}^\top A \widetilde{X}] + \frac{1}{\alpha^2} .$$

*In particular, we have that* $\left\| H^{\dagger/2} \Sigma H^{\dagger/2} - \mathbb{P}_H \right\|_F^2 \lesssim \frac{1}{\alpha} \mathbf{Var}_{\widetilde{X} \sim \widetilde{T}}[\widetilde{X}^\top A \widetilde{X}] + \frac{1}{\alpha^2}$ *and* $\left\| H^{\dagger/2} \Sigma H^{\dagger/2} - \mathbb{P}_\Sigma \right\|_F^2 \lesssim \frac{1}{\alpha} \mathbf{Var}_{\widetilde{X} \sim \widetilde{T}}[\widetilde{X}^\top A \widetilde{X}] + \frac{1}{\alpha^2}$ .

*Proof.* We will apply the main certificate lemma, Lemma D.4, with $P = H^{\dagger/2}$. The desired result then follows by noting that $\mathbf{E}_{X \sim T}[PXX^\top P] = H^{\dagger/2} H H^{\dagger/2} = \mathbb{P}_H$ and the operator norm of $\mathbb{P}_H$ is at most 1. We will use the following result, proved in Appendix C, to upper bound the frobenius norm of $H^{\dagger/2} \Sigma H^{\dagger/2} - \mathbb{P}_\Sigma$:

**Lemma C.3.** *Let $A$ and $B$ be two PSD matrices with $\mathbb{P}_B \preceq \mathbb{P}_A$ or equivalently $\ker(A) \subseteq \ker(B)$. Then $\|ABA - \mathbb{P}_B\|_F \leq 2\|ABA - \mathbb{P}_A\|_F$.*

Before providing the proof sketch of Lemma C.3 below, we show that it suffices to prove the desired result. Since $H$ is the second moment matrix of $T$, we have that $\ker(H) \subseteq \ker(\sum_{x \in S \cap T} xx^\top)$, which is further contained in $\ker(\Sigma)$ by Condition (L.5) in Definition 2.1. We apply Lemma C.3 with $B = \Sigma$ and $A = H$, which gives the desired result.

We now give a brief proof sketch of Lemma C.3. By triangle inequality, it suffices to show that $\|\mathbb{P}_A - \mathbb{P}_B\|_F \leq \|ABA - \mathbb{P}_A\|_F$. Since $\ker(A) \subset \ker(B)$, $\mathbb{P}_A - \mathbb{P}_B$ is again a projection matrix of rank equal to $\mathrm{rank}(A) - \mathrm{rank}(B)$ and thus its frobenius norm is equal to square root of $\mathrm{rank}(A) - \mathrm{rank}(B)$. On the other hand, the matrices $ABA$ and $\mathbb{P}_A$ have a rank difference of at least $\mathrm{rank}(A) - \mathrm{rank}(B)$. Combining this observation with the fact that $\mathbb{P}_A$ has binary eigenvalues, we can lower bound $\|ABA - \mathbb{P}_A\|_F$ by square root of $\mathrm{rank}(A) - \mathrm{rank}(B)$. $\qquad \square$

## E.3 Filtering: Proof of Lemma 3.5

We first show that if we take the median of a quadratic polynomial $p$ over the corrupted set, then the sample mean of $p$ over the inliers is not too far from the median if the left and right quantiles are close.

**Lemma E.3** (Quantiles of quadratics after normalization). *Make Assumption 3.1. Let $A$ be an arbitrary symmetric matrix with $\|A\|_F = 1$. Define $\widetilde{y}_{\mathrm{median}} = \mathrm{Median}\left( \left\{ \widetilde{x}^\top A \widetilde{x} \mid \widetilde{x} \in \widetilde{T} \right\} \right)$. Let $m_1$ be any number less than $|S|/3$. Denote by $q_i$ the $i$-th smallest point of $\left\{ \widetilde{x}^\top A \widetilde{x} \mid \widetilde{x} \in \widetilde{T} \right\}$.*

*Suppose that $q_{|T|-m_1} - q_{m_1} \leq R$ for $R := C(1/\alpha) \log(1/\eta)$ with $C > 100\sqrt{C_1}$ (recall that the absolute constant $C_1$ is defined in Assumption 3.1). Then, $\left| \mathbf{E}_{\widetilde{X} \sim \widetilde{S}}[\widetilde{X}^\top A \widetilde{X}] - \widetilde{y}_{\mathrm{median}} \right| \leq 2R$.*

*Proof.* Let $\mu' = \mathbf{E}_{\widetilde{X} \sim \widetilde{S}}[\widetilde{X}^\top A \widetilde{X}]$. (Not to be confused with $\hat{\mu}$ and $\widetilde{\mu}$ in Assumption 3.1, which are expectations of $\widetilde{X}$ in the samples and at the population level, instead of the quadratic form in the definition of $\mu'$.)

Given that $\widetilde{y}_{\mathrm{median}}$ (by definition) lies within the interval $[q_{m_1}, q_{|T|-m_1}]$ which has length at most $R$, it suffices to argue that $\mu'$ also lies within distance $R$ of that interval. Namely, $\mu' \geq q_{m_1} - R$ and $\mu' \leq q_{|T|-m_1} + R$.

Let $\sigma^2 := \mathbf{Var}_{X \sim D}[(H^{\dagger/2} X)^\top A (H^{\dagger/2} X)]$. The $(\eta, 2\epsilon_0)$-stability of $S$ in Assumption 3.1 (see Condition (L.3)) implies that all but an $\eta$-fraction of $\{\widetilde{x}^\top A \widetilde{x} : \widetilde{x} \in \widetilde{S} \cap \widetilde{T}\}$ lies in the interval

$\mathbf{E}_{X \sim D}[(H^{\dagger/2}X)^\top A (H^{\dagger/2}X)] \pm 10\sigma \log(1/\eta)$ since the size of the set $\widetilde{S} \cap \widetilde{T}\}$ is at least $|S|(1 - 2\epsilon_0)$ (by Assumption 3.1).

Thus, for at least $(1-\eta)|S \cap T| \geq |S|(1 - 2\epsilon_0 - \eta)|S|$ points in $T$, the value of $\widetilde{x}^\top A \widetilde{x}$ lies in the interval $\mathbf{E}_{X \sim D}[(H^{\dagger/2}X)^\top A (H^{\dagger/2}X)] \pm 10\sigma \log(1/\eta)$. Since $\eta \leq 0.01$ and $\epsilon_0 < 1/4$, the number of such points is at least $|S|/3$.

Rephrasing, there is an interval $[y_{\text{left}}, y_{\text{right}}]$ that contains at least $|S|/3$ points in $\widetilde{T}$, where

$$y_{\text{left}} = \mathbf{E}_{X \sim D}[(H^{\dagger/2}X)^\top A (H^{\dagger/2}X)] - 10\sigma \log(1/\eta) \ ,$$

$$y_{\text{right}} = \mathbf{E}_{X \sim D}[(H^{\dagger/2}X)^\top A (H^{\dagger/2}X)] + 10\sigma \log(1/\eta) \ .$$

Therefore, there are at most $|T| - |S|/3$ points in $\widetilde{T}$ that are less than $y_{\text{left}}$, implying that $q_{|T|-|S|/3} \geq y_{\text{left}}$. Furthermore, since, by assumption we have $m_1 \leq |S|/3$, this implies that $q_{|T|-m_1} \geq y_{\text{left}}$. By a symmetric argument, we also have that $q_{m_1} \leq y_{\text{right}}$.

Next, recall that by Item 6 of Lemma E.1, we have that $|\mathbf{E}_{X \sim D}[(H^{\dagger/2}X)^\top A (H^{\dagger/2}X)] - \mu'| = |\mathbf{E}_{X \sim D}[(H^{\dagger/2}X)^\top A (H^{\dagger/2}X)] - \mathbf{E}_{\widetilde{X} \sim \widetilde{S}}[\widetilde{X}^\top A \widetilde{X}]| < 1/\alpha$. Thus, we have $q_{|T|-m_1} \geq y_{\text{left}} \geq \mu' - 10\sigma \log(1/\eta) - 1/\alpha$. Rearranging, we get $\mu' \leq q_{|T|-m_1} + 10\sigma \log(1/\eta) + 1/\alpha$. Symmetrically, we get that $\mu' \geq q_{m_1} - 10\sigma \log(1/\eta) - 1/\alpha$.

Finally, we argue that $10\sigma \log(1/\eta) + 1/\alpha \leq R$, showing that $\mu' \leq q_{|T|-m_1} + R$ and $\mu' \geq q_{m_1} - R$, which as we argued at the beginning of the proof is sufficient to show the lemma statement. To argue this, by Lemma 3.2, we have that $\sigma \leq 5\sqrt{C_1}/\alpha$. Thus, it suffices to choose, as in the lemma statement, that $R = (C/\alpha) \log(1/\eta)$ for some sufficiently large $C > 0$.

$\square$

The above result implies that if we calculate the "squared deviation" of the projections of inliers centered around the *empirical median of the corrupted set*, then it will be of the order of the empirical variance of inliers up to a factor of $R^2$ (by triangle inequality). Moreover, the empirical variance of the inliers is of the order $O(1/\alpha^2)$ by Lemma 3.2. Thus if the variance of the corrupted set is much larger than the upper bound, then we can assign score to each point, the function $f$ below, such that the contribution from outliers is much larger than the inliers.

**Lemma E.4** (Filtering; Formal version of Lemma 3.5)**.** *Make Assumption 3.1. Let $A$ be an arbitrary symmetric matrix with $\|A\|_F = 1$. Let $R = C(1/\alpha) \log(1/\eta)$ for $C \geq 100\sqrt{C_1}$. Define $\widetilde{y}_{\text{median}} = \text{Median}(\{\widetilde{x}^\top A \widetilde{x} \mid \widetilde{x} \in \widetilde{T}\})$. Define the function $f(\widetilde{x}) := (\widetilde{x}^\top A \widetilde{x} - \widetilde{y}_{\text{median}})^2$. Let $m_1$ be a number less than $|S|/3$. Denote by $q_i$ the $i$-th smallest point of $\{\widetilde{x}^\top A \widetilde{x} \mid \widetilde{x} \in \widetilde{T}\}$.*

*If $q_{|T|-m_1} - q_{m_1} \leq R$ and $\mathbf{E}_{X \sim T}[f(x)] > C'R^2/\alpha^3$ for $C' \geq 720/\epsilon_0$, that is, in the case where the check in Line 12 fails, then, the function $f(\cdot)$ satisfies*

$$\sum_{\widetilde{x} \in \widetilde{T}} f(\widetilde{x}) > \frac{40}{\epsilon_0} \frac{1}{\alpha^3} \sum_{\widetilde{x} \in \widetilde{S} \cap \widetilde{T}} f(\widetilde{x}) \ .$$

*Proof.* The core argument for the proof is that $\sum_{\widetilde{x} \in \widetilde{S} \cap \widetilde{T}} f(\widetilde{x})$ can be upper bounded under the assumption that $q_{|T|-m_1} - q_{m_1} \leq R$, as shown in the following claim.

**Claim E.5.** *Assuming the conditions in Lemma E.4, we have $\sum_{\widetilde{x} \in \widetilde{S} \cap \widetilde{T}} f(\widetilde{x}) \leq 9|S|R^2$.*

*Proof.* Let $\mu' = \mathbf{E}_{\widetilde{X} \sim \widetilde{S}}[\widetilde{X}^\top A \widetilde{X}]$. We have the following series of inequalities:

$$\sum_{\widetilde{x} \in \widetilde{S} \cap \widetilde{T}} f(\widetilde{x}) \leq \sum_{\widetilde{x} \in \widetilde{S}} f(\widetilde{x}) \qquad \text{(since } f(x) \geq 0\text{)}$$

$$= \sum_{\widetilde{x} \in \widetilde{S}} (\widetilde{x}^\top A \widetilde{x} - \widetilde{y}_{\text{median}})^2$$

$$\leq 2 \sum_{\widetilde{x} \in \widetilde{S}} \left( \left( x^\top A x - \mu' \right)^2 + \left( \mu' - \widetilde{y}_{\text{median}} \right)^2 \right) \qquad \text{(by } (a+b)^2 \leq 2a^2 + 2b^2 \text{)}$$

$$= 2|S| \operatorname*{\mathbf{Var}}_{\widetilde{X} \sim \widetilde{S}} [\widetilde{X}^\top A \widetilde{X}] + 2 \sum_{\widetilde{x} \in \widetilde{S}} (\mu' - \widetilde{y}_{\text{median}})^2$$

$$\leq 2|S| \operatorname*{\mathbf{Var}}_{\widetilde{X} \sim \widetilde{S}} [\widetilde{X}^\top A \widetilde{X}] + 8R^2|S| \qquad \text{(by Lemma E.3)}$$

$$\leq 2|S| \left( \frac{72C_1}{\alpha^2} \right) + 8R^2|S| \qquad \text{(by Lemma 3.2 and Condition (L.4))}$$

$$\leq 9|S|R^2 , \qquad \text{(by definition of } R\text{)}$$

where the last line uses that $R \geq 12\sqrt{C_1}/\alpha$, which is satisfied for $C \geq 12\sqrt{C_1}$. This completes the proof of Claim E.5. $\qquad\square$

On the other hand, we have that

$$\sum_{\widetilde{x} \in \widetilde{T}} f(\widetilde{x}) > C' R^2 |T| / \alpha^3 \qquad \text{(by assumption)}$$

$$\geq 0.5 C' R^2 |S| / \alpha^3 \qquad \text{(since } |T| \geq |S \cap T| \geq |S|(1 - 2\epsilon_0) \geq |S|/2\text{)}$$

$$> (360/\epsilon_0) R^2 |S| / \alpha^3 \qquad \text{(since } C' \geq 720/\epsilon_0\text{)}$$

$$\geq 40/(\epsilon_0 \alpha^3) \sum_{\widetilde{x} \in \widetilde{S} \cap \widetilde{T}} f(\widetilde{x}) . \qquad \text{(using Claim E.5)}$$

This completes the proof of Lemma E.4.

$\qquad\square$

# F   Proof of Theorem 1.2

In this section, we present the main algorithmic result of the paper. As previously stated, the theorem holds for distributions beyond Gaussians, as long as the input set of points satisfies the deterministic conditions from Section 2.1 and the distribution meets a mild requirement. We now restate and prove the theorem in this more general form.

**Theorem F.1** (Weak List Decodable Covariance Estimation). *Let the ambient dimension be $d \in \mathbb{Z}_+$, let $C$ be a sufficiently large absolute constant, and let the parameters $\alpha \in (0, 1/2)$, $\epsilon \in (0, \epsilon_0)$ for a sufficiently small positive constant $\epsilon_0$, and failure probability $\delta \in (0, 1/2)$ be known to the algorithm. Let $D$ be a distribution with mean $\mu \in \mathbb{R}^d$, covariance $\Sigma \in \mathbb{R}^{d \times d}$, and assume that $\mathbf{Var}_{X \sim D}[X^\top A X] = O(\|\Sigma^{1/2} A \Sigma^{1/2}\|_F^2 + \|\Sigma^{1/2} A \mu\|_2^2)$. There is a polynomial-time algorithm such that, on input $\alpha, \delta$ as well as an $(\alpha, \epsilon)$-corruption of a set $S$ (Definition 1.1) with $|S| > C/(\alpha^6 \epsilon_0^2) \log(1/(\alpha\delta))$ which is $((\epsilon_0/40)\alpha^3, 2\epsilon_0)$-stable with respect to $D$ (Definition 2.1), with probability at least $1 - \delta$ over the randomness of the algorithm, the algorithm returns a list of at most $O(1/\alpha)$ many sets $T_i$ which are disjoint subsets of samples in the input, each $T_i$ has size at least $0.5\alpha m$, and there exists a $T_i$ in the output list such that:*

- *Recall the notation in the corruption model (Definition 1.1) where $n$ is the size of the original inlier set $S$ and $\ell$ is the number of points in $S$ that the adversary replaced—$n$ and $\ell$ are unknown to the algorithm except that $n \geq \alpha m$ and $\ell \leq \epsilon n$. The set $T_i$ satisfies that $|T_i \cap S| \geq (1 - 0.01\alpha)(n - \ell)$.*

- *Denote $H_i := \mathbf{E}_{X \sim T_i}[X X^\top]$. The matrix $H_i$ satisfies*

$$\max \left( \|(H_i^\dagger)^{1/2} \Sigma (H_i^\dagger)^{1/2} - \mathbb{P}_H\|_F \ , \ \|(H_i^\dagger)^{1/2} \Sigma (H_i^\dagger)^{1/2} - \mathbb{P}_\Sigma\|_F \right) \leq O((1/\alpha^4) \log(1/\alpha)).$$

The Gaussian distribution $\mathcal{N}(\mu, \Sigma)$ satisfies the distributional assumption of Theorem F.1 (c.f. Fact B.1), and a set of size $m = C' \frac{d^2 \log^5(d/\alpha\delta)}{\alpha^6}$ from $\mathcal{N}(\mu, \Sigma)$ is $((\epsilon_0/40)\alpha^3, 2\epsilon_0)$-stable with probability $1 - \delta$ for a constant $\epsilon_0$ (c.f. Lemma 2.2). Thus, Theorem F.1 theorem covers the Gaussian

case $D = \mathcal{N}(\mu, \Sigma)$ with polynomial sample complexity, and it directly implies Theorem 1.2, the main result of this work.

We achieve the guarantee of Theorem F.1 by combining the procedure of the previous section in a recursive algorithm (Algorithm 1). We maintain the notation of Assumption 3.1 with the addition of subscripts that indicate the number of filtering steps performed.

We restate the algorithm below, before proving Theorem F.1:

---

**Algorithm 3** List-Decodable Covariance Estimation in Relative Frobenius Norm

---

1: **Constants:** $m, \alpha, \eta, R := C(1/\alpha^2) \log(1/(\epsilon_0 \alpha))$ for $C > 6000\sqrt{C_1}$, $C' > 720/\epsilon_0$ (where $C_1, \epsilon_0$ are defined in Assumption 3.1).
2: **function** COVLISTDECODING($T_0$)
3:     $t \leftarrow 0$.
4:     **loop**
5:         Compute $H_t = \mathbf{E}_{X \sim T_t}[XX^\top]$.
6:         Let $\widetilde{T}_t = \{H_t^{\dagger/2} x \ : \ x \in T_t\}$ be the transformed set of samples.
7:         Let $A$ be the symmetric matrix corresponding to the top eigenvector of $\mathbf{Cov}_{\widetilde{X} \sim \widetilde{T}_t}[\widetilde{X}^{\otimes 2}]$.
8:         Normalize $A$ so that $\|A\|_{\mathrm{F}} = 1$.
9:         Compute the set $\widetilde{Y}_t = \{\widetilde{x}^\top A \widetilde{x} \ : \ \widetilde{x} \in \widetilde{T}_t\}$
10:        Compute the $\alpha m/9$-th smallest element $q_{\text{left}}$ as well as the $\alpha m/9$-th largest element $q_{\text{right}}$, as well as the median $\widetilde{y}_{\text{median}}$ of $\widetilde{Y}_t$.
11:        Define the function $f(\widetilde{x}) = (\widetilde{x}^\top A \widetilde{x} - \widetilde{y}_{\text{median}})^2$.
12:        **if** $\mathbf{E}_{\widetilde{X} \sim \widetilde{T}_t}[f(\widetilde{X})] \leq C'R^2/\alpha^3$ **then**         ▷ c.f. Lemma 3.4
13:            **If** $|T_t| \geq 0.5\alpha m$ **then return** $\{T_t\}$ **else return** the empty list.
14:        **else if** $q_{\text{right}} - q_{\text{left}} \leq R$ **then**         ▷ c.f. Lemma 3.5
15:            Let the probability mass function $p(\widetilde{x}) := f(\widetilde{x})/\sum_{\widetilde{x} \in \widetilde{T}_t} f(\widetilde{x})$.
16:            Pick $x_{\text{removed}} \in T_t$ according to $p(H_t^{\dagger/2} x)$.
17:            $T_{t+1} \leftarrow T_t \setminus \{x_{\text{removed}}\}$.
18:        **else**
19:            $\tau \leftarrow$ FindDivider($\widetilde{Y}_t, \alpha m/9$).         ▷ c.f. Lemma 3.6
20:            $T' \leftarrow \{H_t^{1/2} \widetilde{x} : \widetilde{x} \in \widetilde{T}_t, \widetilde{x}^\top A \widetilde{x} \leq \tau\}$, $T'' \leftarrow \{H_t^{1/2} \widetilde{x} : \widetilde{x} \in T_t, \widetilde{x}^\top A \widetilde{x} > \tau\}$.
21:            $L_1 \leftarrow$ COVLISTDECODING($T'$), $L_2 \leftarrow$ COVLISTDECODING($T''$).
22:            **return** $L_1 \cup L_2$.
23:     $t \leftarrow t + 1$.

---

*Proof of Theorem F.1.* Since the final error guarantees do not depend on $\epsilon$, without loss of generality we use the maximum level of corruptions $\epsilon = \epsilon_0 = 0.01$ in the following proof. We will also use the letter $\eta$ to denote the expression $(\epsilon_0/40)\alpha^3$ as in the stability assumption in the theorem statement on the inlier set. We will proceed to prove Theorem F.1 by induction, and crucially use the following fact that "union bounds also work under conditioning".

**Fact F.2.** *If event $A$ happens with probability $1 - \tau_1$ and event $B$ happens with probability $1 - \tau_2$ conditioned on event $A$, then the probability of both $A$ and $B$ happening is at least $1 - \tau_1 - \tau_2$.*

The technical bulk for proving Theorem F.1 is Lemma F.4, an inductive claim over the levels of recursion, that (in addition to some other basic properties) with high probability over any level, either (i) there was a recursive call at a prior level whose input contains most of the inliers, and by definition that call has terminated, or (ii) there exists some recursive call at this current level whose input contains most of the inliers. We will use the following notation (Definition F.3) to denote various objects in the recursive execution of our algorithm.

**Definition F.3** (Notation for Recursion Tree). *We define a rooted binary tree $\mathcal{T}$ that corresponds to the execution of our recursive algorithm (Algorithm 1). The root node at level 0 corresponds to the top-level call of* COVLISTDECODING *(Algorithm 1) and for every non-leaf node, its left child corresponds to a call of* COVLISTDECODING *from Line 21 and its right child corresponds to a call*

*from Line 21. Thus, every node is uniquely specified by the level in the tree and its position within that level. We denote by $\mathcal{T}_{i,j}$ the $i^{th}$ node of level $j$, where the node numbers are also 0-indexed (e.g., the root node is $\mathcal{T}_{0,0}$). For the node $\mathcal{T}_{i,j}$, we denote by $T_{(0)}^{(i,j)}$ the input data set to the corresponding recursive call. In order to refer to the working data set at the $t^{th}$ iteration of the main loop in the execution of node $\mathcal{T}_{i,j}$, we use the notation $T_{(t)}^{(i,j)}$ and $\widetilde{T}_{(t)}^{(i,j)}$ exactly in the same way as $T_{(t)}$ and $\widetilde{T}_{(t)}$ are used in Algorithm 1. Finally, $\mathcal{T}_j$ (i.e., using a single subscript) refers to the subtree growing from the root and including all the nodes up to and including the $j^{th}$ level.*

**Lemma F.4.** *In the context of Theorem F.1, consider the recursion tree of* COVLISTDECODING *(Definition F.3). Then, for any $j \in \{0, \ldots, 9/\alpha\}$, we have that with probability at least $1 - 0.01\alpha j\delta$ the following holds:*

1. *(Size decrease under recursion) The input of every node $\mathcal{T}_{i,j}$ of level $j$ satisfies $|T_{(0)}^{(i,j)}| \leq m - j\alpha m/9$*

2. *(Disjoint inputs) Consider the subtree $\mathcal{T}_j$ growing from the root and truncated at (but including) level $j$. All the leaves in $\mathcal{T}_j$ have disjoint input sets. Note that the leaves may not all be at level $j$, and some might be at earlier levels.*

3. *(Bound on inliers removed) Consider the subtree $\mathcal{T}_j$ growing from the root and truncated at (but including) level $j$. There exists a leaf $\mathcal{T}_{i',j'}$ in $\mathcal{T}_j$ (but the level of the leaf $j'$ is not necessarily equal to $j$) such that its input $T_{(0)}^{(i',j')}$ satisfies*

$$\alpha^3(\epsilon_0/40)|T_{(0)}^{(i',j')}| + |S \setminus T_{(0)}^{(i',j')}| \leq (j+1)\alpha^3(\epsilon_0/20)m + \ell \tag{12}$$

*where as in Definition 1.1, $m$ is the size of the original input set of the root node $\mathcal{T}_{0,0}$ and $\ell \leq \epsilon_0 n$ is the number of samples that the adversary replaced in $S$. In particular, the number of inliers removed by the algorithm until $T_{(0)}^{(i',j')}$ is at most $(j+1)\alpha^3(\epsilon_0/20)m$.*

**Remark F.5.** We remark on why (12) uses a cubic power of $\alpha$, which is the same cubic power appearing in Lemma E.4 that states that the total score of all the outliers is a $1/\alpha^3$ factor more than the total score of the inliers. The three powers of $\alpha$ are due to the following reasons: 1) the inliers can be as few as an $\alpha$-fraction of all the samples, 2) the depth of the recursion tree can be as much as $O(1/\alpha)$, meaning that a set can be filtered $O(1/\alpha)$ times, and 3) Theorem F.1 guarantees that we remove at most an $O(\alpha)$ fraction of inliers. We also remark that, by tweaking parameters in the algorithm and increasing the power of $\text{poly}(1/\alpha)$ in our covariance estimation guarantees, we can make the score function ratio (and hence the $\text{poly}(\alpha)$ factor in (12)) as large a power of $\alpha$ as we desire. This would guarantee that we remove at most a smaller $\text{poly}(\alpha)$ fraction of inliers.

*Proof.* We prove Lemma F.4 by induction on the level number $j$. For the base case of $j = 0$, Conditions 1 and 2 are trivial. As for Condition 3, at $j = 0$ we have $|T_{(0)}^{(0,0)}| = m$ and $|S \setminus T_{(0)}^{(0,0)}| \leq \ell$ by the definition of the contamination model (Definition 1.1), which directly implies $\alpha^3(\epsilon_0/40)|T_{(0)}^{(0,0)}| + |S \setminus T_{(0)}^{(0,0)}| \leq \alpha^3(\epsilon_0/40)m + \ell \leq \alpha^3(\epsilon_0/20)m + \ell$ as desired.

To show the inductive case, we assume the lemma statement is true for some $j$ and we will show the statement for the case $j + 1$.

**Conditions 1 and 2** These are trivially true, by virtue of the fact that the recursive algorithm, after it is done with removing points via filtering, it partitions the input set using FINDDIVIDER which guarantees that both output sets have size at least $m_1 = \alpha m/9$ (Lemma 3.6).

**Condition 3** Recall the notation $T_{(t)}^{(i,j)}$ from Definition F.3 that is used to denote the set of points in the variable $T_t$ after the $t^{th}$ iteration in the $i^{th}$ recursive call in level $j$.

By Fact F.2, we will condition on Condition 3 being true for $\mathcal{T}_j$, the subtree truncated at (but including) level $j$ and show that Condition 3 holds also for $\mathcal{T}_{j+1}$, the subtree truncated at level $j + 1$, except with probability $0.01\alpha\delta$. Concretely, the conditioning implies that there exists a leaf $\mathcal{T}_{i',j'}$ of the subtree

$\mathcal{T}_j$ whose input set $T_{(0)}^{(i',j')}$ satisfies the inequality in Condition 3 for $j' \leq j$. If $j' < j$, then we are done (since it continues to be a leaf and the desired bound on the right hand size in Equation (12) is only larger). Otherwise, in the case of $j' = j$, we have to analyze the execution of the recursive call associated with this leaf node in the recursion tree $\mathcal{T}_j$.

To begin with the analysis, we verify that the input set $T_{(0)}^{(i',j)}$ satisfies Assumption 3.1. We first check that $|S \cap T_{(0)}^{(i',j)}| \geq n(1 - 2\epsilon_0)$:

$$
\begin{aligned}
|S \cap T_{(0)}^{(i',j)}| &= |S| - |S \setminus T_{(0)}^{(i',j)}| \\
&\geq n - (j+1)\alpha^3(\epsilon_0/20)m - \ell && \text{(by the inductive hypothesis)} \\
&\geq n - (9/\alpha + 1)\alpha^3(\epsilon_0/20)m - \ell && (j \leq 9/\alpha) \\
&\geq n - \alpha^2\epsilon_0 m - \epsilon_0 n && (\ell \leq \epsilon_0 n) \\
&\geq n - \alpha\epsilon_0 n - \epsilon_0 n && (n \geq \alpha m) \\
&\geq n - 1.5\epsilon_0 n\ , && (13)
\end{aligned}
$$

where the last line uses that $\alpha \leq 1/2$. Then it remains to check that $|T_{(0)}^{(i',j)}| \leq (1/\alpha)|S|$, but this is trivially satisfied since it holds for the input set $T_{(0)}^{(0,0)}$ that we start with: $|T_{(0)}^{(i',j)}| \leq |T_{(0)}^{(0,0)}| = m \leq (1/\alpha)n = (1/\alpha)|S|$.

In a recursive call, the algorithm iteratively removes points (through Line 16) before either terminating (Line 12) or calling FINDDIVIDER and recursing (Line 21). Denote the iteration just before either terminating or running FINDDIVIDER by $t_{i'}^*$. We need to argue that the iterative filtering of points prior to iteration $t_{i'}^*$ still roughly preserves Condition 3. Lemma F.6, that is stated below and proved in Lemma F.6, captures the standard martingale argument for filtering-based robust algorithms, and it informally states that if the input set $T_{(0)}^{(i',j)}$ satisfies Assumption 3.1, then Condition 3 is roughly preserved.

**Lemma F.6.** *Recall the notations and assumptions from Assumption 3.1. Consider an execution of* COVLISTDECODING($T_0$) *with $T_0 = T$ (with $T$ satisfying the assumptions from Assumption 3.1), and suppose that $|T_0| \geq C'/(\alpha^6\epsilon_0^2)\log(1/(\alpha\delta))$. Further assume that $|S \setminus T_0| \leq 1.5\epsilon_0|S|$ (which is a strengthening over Assumption 3.1). Moreover, denote by $T_t$ the dataset through the loop iterations of Algorithm 1. Then, with probability at least $1 - 0.01\alpha\delta$, it holds that*

$$\alpha^3(\epsilon_0/40)|T_t| + |S \setminus T_t| \leq \alpha^3(\epsilon_0/40)|T_0| + |S \setminus T_0| + \alpha^3(\epsilon_0/40)|T_0|\ ,$$

*simultaneously for all iterations $t$ until the execution of* COVLISTDECODING *enters either Line 13 or Line 19.*

Concretely, applying Lemma F.6 for input set $T_{(0)}^{(i',j)}$ and with failure probability $0.01\alpha\delta$ in place of $\delta$. The lemma is applicable since its first requirement $|S \setminus T_{(0)}^{(i',j)}| \leq 1.5\epsilon_0|S|$ has been already checked in (13), and for its second requirement we have that $|T_{(0)}^{(i',j)}| \geq |T_{(0)}^{(i',j)} \cap S| \geq (1 - 1.5\epsilon_0)|S| \geq C'/(\alpha^6\epsilon_0^2)\log(1/(\alpha\delta))$, where the last one uses (13) and the next step uses our assumption $|S| > C/(\alpha^6\epsilon_0^2)\log(1/(\alpha\delta))$. The lemma then yields that, with probability at least $1 - 0.01\alpha\delta$, $T_{(t_{i'}^*)}^{(i',j)}$ is such that

$$\alpha^3(\epsilon_0/40)|T_{(t_{i'}^*)}^{(i',j)}| + |S \setminus T_{(t_{i'}^*)}^{(i',j)}| \leq (j+1)\alpha^3(\epsilon_0/20)m + \ell + (\epsilon_0/40)\alpha^3 m\ . \qquad (14)$$

Now, either the recursive call terminates at iteration $t_{i'}^*$ or it goes into Line 19. In the former case, we are done (since it is now a leaf node). Otherwise, the recursive call uses FINDDIVIDER to partition $T_{(t_{i'}^*)}^{(i',j)}$ into $T'$ and $T''$. We need to show that at least one of $T'$ and $T''$ satisfies Condition 3 (for case $j + 1$), using Lemma 3.6. Let us now derive what parameters and set to invoke Lemma 3.6 with.

Using (i) Condition (L.3) of the stability property of the original inlier set $S$, and (ii) the fact that $(H^{\dagger/2}X)^\top A(H^{\dagger/2}X) - \mathbf{E}_{X \sim \mathcal{N}(\mu,\Sigma)}[(H^{\dagger/2}X)^\top A(H^{\dagger/2}X)]$ is an even quadratic in $X$ for any $H$

and $A$, it must be the case that a $1 - \eta$ fraction (recall $\eta$ is a shorthand for $(\epsilon_0/40)\alpha^3$) of the points $x \in S$ satisfies

$$\left| (H^{\dagger/2}x)^\top A(H^{\dagger/2}x) - \mathop{\mathbf{E}}_{X \sim \mathcal{N}(\mu,\Sigma)} [(H^{\dagger/2}X)^\top A(H^{\dagger/2}X)] \right| \tag{15}$$

$$\leq 10\log(2/\eta)\sqrt{\mathop{\mathbf{Var}}_{X \sim D}[(H^{\dagger/2}X)^\top A(H^{\dagger/2}X)]}$$

$$\leq 50\log(2/\eta)\sqrt{\frac{C_1}{\alpha^2}} . \tag{Lemma 3.2}$$

Denote the set of points $x \in S$ with the above property by $S_{H,A} \subseteq S$, and let $S_{\text{core}}^{(i',j)} = T_{(t_{i'}^*)}^{(i',j)} \cap S_{H,A}$. We will therefore apply Lemma 3.6 with $S_{\text{proj}} = \{(H^{\dagger/2}x)^\top A(H^{\dagger/2}x) : x \in S_{\text{core}}^{(i',j)}\}$, $T = \{(H^{\dagger/2}x)^\top A(H^{\dagger/2}x) : x \in T_{(t_{i'}^*)}^{(i',j)}\}$ and diameter $r = 100\log(2/\eta)\sqrt{C_1}/\alpha$. It remains to check that for $q_{m_1} = q_{\text{left}}$ and $q_{|T_{(t_{i'}^*)}^{(i',j)}| - m_1} = q_{\text{right}}$ of $T_{(t_{i'}^*)}^{(i',j)}$, we have $q_{|T_{(t_{i'}^*)}^{(i',j)}| - m_1} - q_{m_1} \geq 10(|T_{(t_{i'}^*)}^{(i',j)}|/|S_{\text{proj}}|)r$, to make sure that the application of Lemma 3.6 is valid.

To show this, recall that Line 14 fails whenever FINDDIVIDER is called, meaning that $q_{|T_{(t_{i'}^*)}^{(i',j)}| - m_1} - q_{m_1} \geq R$. Thus, we need to verify that the definition of $R$ in Algorithm 1 satisfies $R \geq 10(|T_{(t_{i'}^*)}^{(i',j)}|/|S_{\text{proj}}|)r$. The main step is to lower bound $|S_{\text{proj}}| = |S_{\text{core}}^{(i',j)}|$. Since $S_{\text{core}}^{(i',j)} \subset S$, we can lower bound the above by

$$|S_{\text{core}}^{(i',j)}| \geq |S| - |S \setminus S_{H,A}| - |S \setminus T_{(t_{i'}^*)}^{(i',j)}| \qquad (S_{\text{core}}^{(i',j)} = T_{(t_{i'}^*)}^{(i',j)} \cap S_{H,A})$$

$$\geq (1-\eta)|S| - |S \setminus T_{(t_{i'}^*)}^{(i',j)}| \qquad \text{(by the definition of } S_{H,A})$$

$$\geq (1-\eta)|S| - (j+1)\alpha^3(\epsilon_0/20)m - \ell - (\epsilon_0/40)\alpha^3 m \qquad \text{(by (14))}$$

$$\geq (1-\eta)n - (j+2)\alpha^3(\epsilon_0/20)m - \ell \qquad (|S| = n \text{ and basic inequality})$$

$$\geq (1-\eta)n - (j+2)\alpha^2(\epsilon_0/20)n - \ell \qquad (n \geq \alpha m)$$

$$\geq (1-\eta)n - \alpha\epsilon_0 n - \ell \qquad (j \leq 9/\alpha \text{ and } \alpha < 1/2)$$

$$\geq (1-\eta)n - \alpha\epsilon_0 n - \epsilon_0 n \qquad (\ell \leq \epsilon_0 n)$$

$$\geq (1 - \eta - 2\epsilon_0)n \geq n/2 \qquad (\eta \leq 0.001 \text{ and } \epsilon_0 = 0.01)$$

Combining with the fact that $|T_{(t_{i'}^*)}^{(i',j)}| \leq m$, we have that $10(|T_{(t_{i'}^*)}^{(i',j)}|/|S_{\text{proj}}|)r \leq 20(m/n)r \leq (20/\alpha)r$. Recalling that $r = 100\log(2/\eta)\sqrt{C_1}/\alpha$, $\eta = (\epsilon_0/40)\alpha^3$ and $\epsilon_0 = 0.001$, the definition of $R$ in Algorithm 1 satisfies $R \geq 6000\sqrt{C_1}(1/\alpha^2)\log(1/(\epsilon_0\alpha)) \geq (20/\alpha)r \geq 10(|T_{(t_{i'}^*)}^{(i',j)}|/|S_{\text{proj}}|)r$.

Knowing that the application of Lemma 3.6 is valid, the lemma then guarantees that either $T'$ or $T''$ contains all of $S_{\text{core}}^{(i',j)}$. Without loss of generality, we assume this happens for $T'$. We now check Condition 3 for case $j + 1$ on $T'$:

$$\alpha^3(\epsilon_0/40)|T'| + |S \setminus T'| \leq \alpha^3(\epsilon_0/40)|T_{(t_{i'}^*)}^{(i',j)}| + |S \setminus T'| \qquad \text{(since } T' \subset T_{(t_{i'}^*)}^{(i',j)})$$

$$\leq \alpha^3(\epsilon_0/40)|T_{(t_{i'}^*)}^{(i',j)}| + |S \setminus S_{\text{core}}^{(i',j)}| \qquad \text{(by Lemma 3.6)}$$

$$\leq \alpha^3(\epsilon_0/40)|T_{(t_{i'}^*)}^{(i',j)}| + |S \setminus T_{(t_{i'}^*)}^{(i',j)}| + |S \setminus S_{H,A}|$$

$$\text{(by the definition of } S_{\text{core}}^{(i',j)} \text{ and a union bound)}$$

$$\leq \alpha^3(\epsilon_0/40)|T_{(t_{i'}^*)}^{(i',j)}| + |S \setminus T_{(t_{i'}^*)}^{(i',j)}| + \eta n \qquad \text{(by the definition of } S_{H,A})$$

$$\leq (j+1)\alpha^3(\epsilon_0/20)m + \ell + (\epsilon_0/40)\alpha^3 m + \eta n \qquad \text{(by Equation 14)}$$

$$\leq (j+1)\alpha^3(\epsilon_0/20)m + \ell + (\epsilon_0/20)\alpha^3 m$$

$$\text{(since } \eta = (\epsilon_0/40)\alpha^3 \text{ and } n \leq m)$$

$$= (j+2)\alpha^3(\epsilon_0/20)m + \ell . \tag{16}$$

Thus Condition 3 is satisfied by $T'$ for case $j+1$, completing the inductive proof of this lemma.

$\square$

We now use the lemma to conclude the proof of Theorem F.1.

By Lemma F.4, we know that with probability at least $1 - 0.09\delta$, the recursion tree generated up to level $9/\alpha$, namely $\mathcal{T}_{9/\alpha}$, satisfies the lemma guarantees. In particular, since (i) by Line 19 in Algorithm 1, the input to any recursive call must have size at least $m_1 = \alpha m/9$, and (ii) Condition 1 of Lemma F.4 applied to $j = m/m_1 = 9/\alpha$ yields an input size upper bound of 0, we can conclude that the execution of the algorithm must have completely terminated by level $9/\alpha$. We know by Lemma F.4 that there exists a leaf node $i$ in the recursion tree whose input satisfies Condition 3 of the third part of the lemma statement. That is, letting $j$ denote the level of that leaf, we have that

$$\alpha^3(\epsilon_0/40)|T_{(0)}^{(i,j)}| + |S \setminus T_{(0)}^{(i,j)}| \le (j+1)\alpha^3(\epsilon_0/20)m + \ell . \tag{17}$$

The recursion node $\mathcal{T}_{i,j}$ starts with that input set $T_{(0)}^{(i,j)}$ and, before terminating, it may perform some filtering steps. As in the proof of Lemma F.4, we will use Lemma F.6 to analyze the "working set" right before the recursive call terminates. If we denote by $t^*$ the number of filtering steps before termination, Lemma F.6 yields that except with probability $0.01\alpha\delta$, we have

$$\alpha^3(\epsilon_0/40)|T_{(t^*)}^{(i,j)}| + |S \setminus T_{(t^*)}^{(i,j)}| \le (j+1)\alpha^3(\epsilon_0/20)m + \ell + (\epsilon_0/40)\alpha^3 m . \tag{18}$$

Note that by Fact F.2, the total failure probability is upper bounded by $0.09\delta + 0.01\alpha\delta$ which is less than $\delta$.

We are now ready to prove the first bullet of Theorem F.1. The above inequality implies that

$$
\begin{aligned}
|T_{(t^*)}^{(i,j)} \cap S| = |S| - |S \setminus T_{(t^*)}^{(i,j)}| \\
\ge n - \ell - (\epsilon_0/20)(j+1)\alpha^3 m - (\epsilon_0/40)\alpha^3 m && \text{(by (18))}\\
\ge n - \ell - (\epsilon_0/20)(j+2)\alpha^3 m \\
\ge n - \ell - (\epsilon_0/20)(j+2)\alpha^2 n && (n \ge \alpha m)\\
\ge n - \ell - (\epsilon_0/2)\alpha n && (j \le 9/\alpha \text{ and } \alpha < 1/2)\\
\ge n - \ell - \epsilon_0(1-\epsilon_0)\alpha n && (1/2 \le 1 - \epsilon_0)\\
\ge n - \ell - \epsilon_0\alpha(n-\ell) && (\ell \le \epsilon_0 n)\\
= (n-\ell)(1 - \epsilon_0\alpha) . && (19)
\end{aligned}
$$

(19) further implies that $|T_{(t^*)}^{(i,j)} \cap S| \ge (n-\ell)(1-\epsilon_0\alpha) \ge n(1-\epsilon_0)(1-\epsilon_0\alpha) \ge (\alpha/2)m$ (since $n \ge \alpha m$ and $\epsilon_0$ and $\alpha$ are small), meaning that the set $T_{(t^*)}^{(i,j)}$ will indeed be returned. This completes the proof of the first bullet of Theorem F.1.

We now move to the second bullet. Defining $H = \mathbf{E}_{X \sim T_{(t^*)}^{(i,j)}}[XX^\top]$, we need to show that $\|H^{\dagger/2}\Sigma H^{\dagger/2} - HH^\dagger\|_F \lesssim (1/\alpha^4)\log^2(1/\alpha)$. We will apply Lemma E.2 to show the bounds in the second bullet, which requires checking that Assumption 3.1 is satisfied by $T_{(t^*)}^{(i,j)}$. Earlier, right below (19), we have already checked that $|T_{(t^*)}^{(i,j)} \cap S| \ge (\alpha/2)|T_{(t^*)}^{(i,j)}|$. The only remaining condition to check is $|T_{(t^*)}^{(i,j)}| \le |S|/\alpha$, but this is trivially true since $T_{(t^*)}^{(i,j)} \subset T_{(0)}^{(0,0)}$ (the original input set to the recursive algorithm) and $|T_{(0)}^{(0,0)}| \le |S|/\alpha$ by Definition 1.1. Thus, Lemma E.2 yields that

$$
\begin{aligned}
\left\| H^{\dagger/2}\Sigma H^{\dagger/2} - \mathbb{P}_H \right\|_F^2 &\lesssim \frac{1}{\alpha} \mathbf{Var}_{\widetilde{X} \sim T_{(t^*)}^{(i,j)}}[\widetilde{X}^\top A \widetilde{X}] + \frac{1}{\alpha^2} \\
&\le \frac{1}{\alpha} \mathbf{E}_{\widetilde{X} \sim T_{(t^*)}^{(i,j)}}[f(\widetilde{X})] + \frac{1}{\alpha^2} && (\mathbf{Var}(Y) \le \mathbf{E}[(Y-c)^2] \text{ for any } c \in \mathbb{R})
\end{aligned}
$$

$$\lesssim \frac{R^2}{\alpha^4} \qquad\qquad\qquad\qquad \text{(c.f. Line 12 of Algorithm 1)}$$

$$\lesssim \frac{1}{\alpha^8} \log^2\left(\frac{1}{\alpha}\right) , \quad (R = \Theta((1/\alpha^2)\log(1/\alpha)), \text{ noting that } \epsilon_0 = 0.01)$$

where the penultimate inequality uses the fact that, if the algorithm terminated, it must be the case that $\mathbf{E}_{\widetilde{X} \sim T_{(t^*)}^{(i,j)}}[f(\widetilde{X})] \le C'R^2/\alpha^3$ by Line 12 in Algorithm 1. Taking a square root on both sides implies that

$$\left\| H^{\dagger/2}\Sigma H^{\dagger/2} - \mathbb{P}_H \right\|_{\mathrm{F}} \lesssim \frac{1}{\alpha^4}\log\left(\frac{1}{\alpha}\right)$$

The same guarantee holds for $\left\| H^{\dagger/2}\Sigma H^{\dagger/2} - \mathbb{P}_\Sigma \right\|_{\mathrm{F}}$, also following from Lemma E.2.

Lastly, we check that we only return $O(1/\alpha)$ sets in the output list. This is true by construction: (i) Line 13 only allows sets to be output if they have size at least $\Omega(\alpha m)$, (ii) there were only $m$ points to begin with and (iii) all the leaves have disjoint input sets by Lemma F.4. $\qquad \square$

## F.1 Proof of Lemma F.6

We now prove that filtering does not remove too many inliers using a standard martingale argument.

**Lemma F.6.** *Recall the notations and assumptions from Assumption 3.1. Consider an execution of* COVLISTDECODING($T_0$) *with $T_0 = T$ (with $T$ satisfying the assumptions from Assumption 3.1), and suppose that $|T_0| \ge C'/(\alpha^6\epsilon_0^2)\log(1/(\alpha\delta))$. Further assume that $|S \setminus T_0| \le 1.5\epsilon_0|S|$ (which is a strengthening over Assumption 3.1). Moreover, denote by $T_t$ the dataset through the loop iterations of Algorithm 1. Then, with probability at least $1 - 0.01\alpha\delta$, it holds that*

$$\alpha^3(\epsilon_0/40)|T_t| + |S \setminus T_t| \le \alpha^3(\epsilon_0/40)|T_0| + |S \setminus T_0| + \alpha^3(\epsilon_0/40)|T_0| ,$$

*simultaneously for all iterations $t$ until the execution of* COVLISTDECODING *enters either Line 13 or Line 19.*

*Proof.* We will use the notation of Assumption 3.1 for a single call of Algorithm 1. Denote by $t$ the iteration count. Also define the stopping time $t_{\mathrm{end}}$ to be the first iteration when $\alpha^3(\epsilon_0/40)|T_{t_{\mathrm{end}}}| + |S \setminus T_{t_{\mathrm{end}}}| > 2\epsilon_0|S|$ or when the iteration goes into Line 12 or Line 19 instead of Line 16 (that is, when $\mathbf{E}_{X \sim T_{t_{\mathrm{end}}}}[f(H_t^{\dagger/2}X)] \le C'R^2/\alpha^3$ or when $q_{\mathrm{right}} - q_{\mathrm{left}} > R$ for the iteration $t_{\mathrm{end}}$). Now, define $\Delta_t = \alpha^3(\epsilon_0/40)|T_{\min\{t,t_{\mathrm{end}}\}}| + |S \setminus T_{\min\{t,t_{\mathrm{end}}\}}|$.

In order to prove the lemma, we will show that at the first $t^*$ (if one exists) such that $\alpha^3(\epsilon_0/40)|T_{t^*}| + |S \setminus T_{t^*}| > \alpha^3(\epsilon_0/40)|T_0| + |S \setminus T_0| + \alpha^3(\epsilon_0/40)|T_0|$, then $\Delta_{t^*} > \alpha^3(\epsilon_0/40)|T_0| + |S \setminus T_0| + \alpha^3(\epsilon_0/40)|T_0|$ as well. Afterwards, we will show that $\Delta_t$ is a sub-martingale and use sub-martingale tail bounds to show that the sequence $\Delta_t$ remains small over an entire trajectory of $|T_0|$ steps with high probability.

The first step is easy to show, since the threshold of $\alpha^3(\epsilon_0/40)|T_0| + |S \setminus T_0| + \alpha^3(\epsilon_0/40)|T_0| < \alpha^3(\epsilon_0/20)|T_0| + 1.5\epsilon_0|S| \le \alpha^2(\epsilon_0/20)|S| + 1.5\epsilon_0|S| \le 2\epsilon_0|S|$, where the first inequality is by the lemma assumption. Therefore, $t^* \le t_{\mathrm{end}}$ if it exists, meaning that $\Delta_{t^*} = \alpha^3(\epsilon_0/40)|T_{t^*}| + |S \setminus T_{t^*}|$.

Now we need to show that $\Delta_t$ is a sub-martingale with respect to the sequence $T_{\min\{t,t_{\mathrm{end}}\}}$. If $t \ge t_{\mathrm{end}}$, then $\Delta_{t+1}$ is by definition equal to $\Delta_t$ and thus $\mathbf{E}[\Delta_{t+1} \mid T_{\min\{t,t_{\mathrm{end}}\}}] = \Delta_t$. Otherwise, we have that $\mathbf{E}[\Delta_{t+1} \mid T_t]$ is equal to $\Delta_t$ plus the expected number of inliers removed minus $\alpha^3(\epsilon_0/40)$ times the expected number of all points removed by our filter. Since the stopping condition is not satisfied, we have $|S \setminus T_t| \le 2\epsilon_0|S|$, meaning that Assumption 3.1 continues to hold for $T_t$, as well as that the other conditions for Lemma E.4 hold. We can therefore apply Lemma E.4 to obtain

$$\mathbf{E}[\Delta_{t+1} \mid T_t] = \Delta_t + \sum_{\widetilde{x} \in \widetilde{S} \cap \widetilde{T}_t} f(\widetilde{x}) - \alpha^3 \frac{\epsilon_0}{40} \sum_{\widetilde{x} \in \widetilde{T}_t} f(\widetilde{x}) \le \Delta_t ,$$

Summarizing, the above case analysis shows that $\{\Delta_t\}_{t \in \mathbb{N}}$ is a sub-martingale with respect to $T_{\min\{t,t_{\mathrm{end}}\}}$.

We note also that for every $t$, $|\Delta_t - \Delta_{t+1}| \leq 1$ with probability 1. Thus, using the standard Azuma-Hoeffding inequality for sub-martingales (Fact B.2), we have that for every $t \leq |T_0|$,

$$\mathbf{Pr}\left[\Delta_t - \Delta_0 > (\epsilon_0/40)\alpha^3|T_0|\right] \leq e^{-(2/40^2)\cdot\epsilon_0^2\alpha^6|T_0|} .$$

By a union bound over all $t \in [|T_0|]$, we have

$$\mathbf{Pr}\left[\exists t \in [m] : \Delta_t > \Delta_0 + \alpha^3(\epsilon_0/40)|T_0|\right] \leq |T_0|e^{-(2/40^2)\cdot\epsilon_0^2\alpha^6|T_0|} \leq 0.01\alpha\delta , \qquad (20)$$

where the last inequality uses that $|T_0| > \frac{C'}{\alpha^6\epsilon_0^2} \log\left(\frac{1}{\alpha\delta}\right)$ for a sufficiently large absolute constant $C'$.

$\square$

# G  Applications to Learning Gaussian Mixture Models

In this section, we prove the applications of our result to robustly learning Gaussian mixture models.

**Theorem 1.4** (Outlier-Robust Clustering and Estimation of Covariances for GMM). *Let $C' > 0$ be a sufficiently large constant and $\epsilon_0 > 0$ be a sufficiently small constant. Let the parameters $\alpha \in (0, 1/2)$, $\epsilon \in (0, \epsilon_0)$, and failure probability $\delta \in (0, 1/2)$ be known. There is an $\tilde{O}(m^2d^2)$-time algorithm such that, on input $\alpha, \delta$, and $m > C'\frac{d^2 \log^5(d/\alpha\delta)}{\alpha^6}$ many $\epsilon$-corrupted samples from an unknown $k$-component Gaussian mixture $\sum_{p=1}^k \alpha_p\mathcal{N}(\mu_p, \Sigma_p)$ over $\mathbb{R}^d$ as in Definition 1.3, where all $\Sigma_p$'s are full-rank and all $\alpha_p$ satisfies $\alpha_p \geq \alpha$ and $k \leq \frac{1}{\alpha}$ is unknown to the algorithm, with probability at least $1 - \delta$ over the corrupted samples and the randomness of the algorithm, the algorithm returns a list of at most $k$ many disjoint subsets of samples $\{T_i\}$ such that:*

- *For the $p^{th}$ Gaussian component, denote the set $S_p$ as the samples in the inlier set $S$ that were drawn from component $p$. Let $n_p$ be the size of $S_p$, and let $\ell_p$ be the number of points in $S_p$ that the adversary replaced—$n_p$ and $\ell_p$ are both unknown to the algorithm except that $\mathbf{E}[n_p] = \alpha_p m \geq \alpha m$ for each $p$ and $\sum_p \ell_p \leq \epsilon\alpha m$. Then, for every Gaussian component $p$ in the mixture, there exists a set $T_{i_p}$ in the returned list such that $|T_{i_p} \cap S_p| \geq (1 - 0.01\alpha)(n_p - \ell_p)$.*

- *For every component $p$, there is a set of samples $T_{i_p}$ in the returned list such that, defining $H_{i_p} = \mathbf{E}_{X \sim T_{i_p}}[XX^\top]$, we have $H_{i_p}$ satisfying $\|H_{i_p}^{-1/2}\Sigma_p H_{i_p}^{-1/2} - I\|_F \lesssim (1/\alpha^4)\log(1/\alpha)$.*

- *Let $\Sigma$ be the (population-level) covariance matrix of the Gaussian mixture. For any two components $p \neq p'$ with $\|\Sigma^{-1/2}(\Sigma_p - \Sigma_{p'})\Sigma^{-1/2}\|_F > C(1/\alpha)^5 \log(1/\alpha)$ for a sufficiently large constant $C$, the sets $T_{i_p}$ and $T_{i_{p'}}$ from the previous bullet are guaranteed to be different.*

We show the first two bullets as a direct corollary of Theorem 1.2 applied to each Gaussian component $D = \mathcal{N}(\mu_p, \Sigma_p)$ as the inlier distribution. The proof of the final bullet involves some more involved linear algebraic lemmas. Here we give only the high-level proof of the final bullet, and defer the proof of these individual lemmas to Appendix C.

*Proof.* First observe that, with a sufficiently large $\text{poly}(d, \frac{1}{\alpha}, \log\frac{1}{\delta})$ samples, with probability at least $1 - \delta/4$, the number of inliers $n_p$ from component $p$ is at least $0.99\alpha m$ for all components $p$. We will condition on this event.

Furthermore, noting that in the corruption model defined in Definition 1.3, at most $\epsilon\alpha m$ points can be corrupted total, this implies that $\ell_p \leq \epsilon\alpha m \leq (\epsilon/0.99)n_p$ (since $n_p \geq 0.99\alpha m$).

Therefore, we apply Theorem 1.2 to all the $k$ Gaussian components, using parameter $0.99\alpha$ in place of $\alpha$, $\epsilon/0.99$ in place of $\epsilon$, and failure probability $\delta/(4k)$. This implies that, with a union bound, using $m = \text{poly}(d, 1/\alpha, \log k/\delta) = \text{poly}(d, 1/\alpha, \log 1/\delta)$ samples (since $k \lesssim 1/\alpha$, for otherwise the Gaussian mixture will have weight greater than 1), with probability at least $1 - \delta/4$, the two bullet points in the theorem holds for all the Gaussian components, directly guaranteed by Theorem 1.2.

It remains to show that the algorithm will return at most $k$ subsets in its output list. Noting that, across the subsets output by the algorithm for each of the components, there is already a total of $(1-0.01\alpha)\sum_{p=1}^k(n_p - \ell_p)$ points being output. Recall that $\sum_p n_p = m$ and $\sum_p \ell_p \leq \epsilon\alpha m$. Thus, in

the output subsets corresponding to the components, there are a total of at least $m - (\epsilon + 0.01)\alpha m \geq m - 0.4\alpha m$ points (where we used that $\epsilon$ is bounded by sufficiently small constant. Since (i) there were only $m$ points in the input and (ii) all the output subsets are disjoint subsets of the input, this means that there are at most $0.4\alpha m$ points that can be output in any additional subsets not corresponding to any Gaussian component. However, by Theorem 1.2, any output subset must have size at least $0.5(0.99\alpha)m$, and so there cannot be any extra subset output in the list that does not correspond to any Gaussian component. This shows that there can be at most $k$ subsets in the output list.

We turn to the final bullet of the theorem. We will show that any two components $p$ and $p'$ that belong to the same set $T_i$ in the output of the algorithm satisfy that $\|\Sigma^{-1/2}\Sigma_p\Sigma^{-1/2} - \Sigma^{-1/2}\Sigma_{p'}\Sigma^{-1/2}\|_F \lesssim \frac{1}{\alpha^5}\log(1/\alpha)$. Taking the contrapositive yields the final theorem bullet.

Let $H$ be the second moment matrix of this set, that is, $H = \mathbf{E}_{X \sim T_i}[XX^\top]$. Then $H$ satisfies that $\|I - H^{\dagger/2}\Sigma_p H^{\dagger/2}\|_F \leq r$ and $\|I - H^{\dagger/2}\Sigma_{p'}H^{\dagger/2}\|_F \leq r$ for $r \lesssim (1/\alpha^4)\log(1/\alpha)$, by the second bullet point of the theorem. Furthermore, $H$ is full rank because $H$ is the second moment matrix of a large subset of empirical points of component $p$ and thus Condition (L.5) implies that $\ker(H)$ is contained in $\ker(\Sigma_p)$, which is empty since $\Sigma_p$ is full rank by the theorem assumption.

We will now apply the following result (Lemma C.1) to infer that $\Sigma_p$ and $\Sigma_{p'}$ must be close, in the sense of the final theorem bullet point. We defer the proof of this lemma to Appendix C.

We will now apply Lemma C.1 to $\Sigma_1 = \Sigma_p$ and $\Sigma_2 = \Sigma_{p'}$, and upper bound the right hand side of the lemma to give the final theorem bullet point.

We claim that the operator norm of $\Sigma^{-1/2}\Sigma_p\Sigma^{-1/2}$ is bounded by $1/\alpha$, due to the fact that component $p$ has mass at least $\alpha$. Indeed, we have that

$$\Sigma_p \preceq \mathop{\mathbf{E}}_{X \sim \mathcal{N}(\mu_p,\Sigma_p)}[(X-\mu)(X-\mu)^\top] \preceq \frac{1}{\alpha_p} \cdot \mathop{\mathbf{E}}_{X \sim \sum_i \alpha_i \mathcal{N}(\mu_i,\Sigma_i)}[(X-\mu)(X-\mu)^\top] = \frac{1}{\alpha_p} \cdot \Sigma \ .$$

This in particular implies that $\Sigma^{-1/2}\Sigma_p\Sigma^{-1/2} \preceq \frac{1}{\alpha_p}I \preceq \frac{1}{\alpha}I$, and hence $\|\Sigma^{-1/2}\Sigma_p\Sigma^{-1/2}\|_{op} \leq \frac{1}{\alpha}$. The same conclusion holds for the component $p'$. Recalling that $r \lesssim (1/\alpha^4)\log(1/\alpha)$, the right hand side of Lemma C.1 can be upper bounded by $O((1/\alpha^5)\log(1/\alpha))$, thus yielding the final bullet in the theorem, as discussed above.

$\square$

We end this section with a brief discussion on the proof of Lemma C.1. The lemma states that, if there exists a single matrix $H$ that is an approximate square root to both $\Sigma_1$ and $\Sigma_2$, in the sense that $\|I - H^{\dagger/2}\Sigma_1 H^{\dagger/2}\|_F$ and $\|I - H^{\dagger/2}\Sigma_1 H^{\dagger/2}\|_F$ are both upper bounded, then $\Sigma_1$ and $\Sigma_2$ cannot be too far from each other, in that $\Sigma_1 - \Sigma_2$ must also have bounded Frobenius norm. In fact, this can be generalized to the normalized version $\Sigma^{-1/2}(\Sigma_1 - \Sigma_2)\Sigma^{-1/2}$ for any positive definite matrix $\Sigma$.

For simplicity of the discussion, let us assume for now that $\Sigma = I$. To show Lemma C.1, a natural first idea is to show that, if $\|I - H^{\dagger/2}\Sigma_1 H^{\dagger/2}\|_F$ is bounded, say by some quantity $\rho$, then so must be $\|HH^\top - \Sigma_1\|_F$ by some quantity related to $O(\rho)$, times the operator norm of $\Sigma_1$ for appropriate scaling. Unfortunately, that is not true: Even for $\Sigma_1 = I$, we can have $H$ being infinity in the first diagonal element and 1s in the other diagonal elements, and $\|I - H^{\dagger/2}\Sigma_1 H^{\dagger/2}\|_F$ would only be 1 yet $HH^\top - \Sigma_1$ is infinite in the first dimension. The crucial observation, then, is that a slightly weaker statement is true: On an orthonormal basis that depends only on $H$ and not on $\Sigma_1$, $HH^\top$ approximates $\Sigma_1$ in most dimensions, namely $d - O(\rho^2)$ many dimensions. This statement is formally captured by Lemma C.2. Thus, in these dimensions, we can bound the difference between $\Sigma_1$ and $\Sigma_2$ (in Frobenius norm). In the rest of the dimensions where $HH^\top$ does not approximate either $\Sigma_1$ or $\Sigma_2$ well, we can bound these dimensions' contribution to the Frobenius norm of $\Sigma_1 - \Sigma_2$ simply by $\max(\|\Sigma_1\|_{op}, \|\Sigma_2\|_{op})$ times the number of such dimensions, which is small. Combining these bounds yields Lemma C.1 as desired.

