# OpenReview forum: "A Spectral Algorithm for List-Decodable Covariance Estimation in Relative Frobenius Norm"
_NeurIPS.cc/2023/Conference — NeurIPS 2023 spotlight_

### Official Review · Reviewer_WZLc · 2023-07-03

**Soundness:** 3 good
**Presentation:** 3 good
**Contribution:** 3 good
**Rating:** 7
**Confidence:** 3

**Summary:**

This work studies the problem of list-decodable covariance estimation. In this problem, we are given a number of samples, an $\alpha$-fraction of which were i.i.d. drawn from a Gaussian distribution $N(\mu,\Sigma)$. The authors design an algorithm that runs in time $\mathrm{poly}(d/\alpha)$ and produces a list of size $O(1/\alpha)$ such that one element in the list is close to $\Sigma$ in relative Frobenius norm (up to error $\mathrm{poly(1/\alpha)}$).

Previous work has given an $d^{\mathrm{poly(1/\alpha)}}$-time algorithm that achieves the stronger guarantee that one element in the list is close in total variation distance to $N(0,\Sigma)$ (in this case the list contains tuples of estimates, one for the mean and one for the covariance). While the TVD is related to the error metric the authors consider, the recovery guarantees of the authors do not imply anything about TVD-closeness. This is unsurprising, there are known SQ lower bounds that suggest that for TVD guarantees $d^{\mathrm{poly(1/\alpha)}}$ samples are required. The authors show that if one settles for the weaker guarantee of "only" relative Frobenius norm error, efficient algorithms, i.e., that run in time polynomial in all problem parameters, are possible.

Their techniques are similar to the techniques for list-decodable mean estimation with additional technical tweaks to handle the more challenging covariance estimation setting. They also provide an application to robustly learning mixtures of non-spherical Gaussians.

**Strengths:**

List-decodable covariance estimation is a very natural and important problem. The relative Frobenius norm metric the authors consider is the natural choice here since it captures the "intrinsic" scale of the problem induced by the true covariance matrix -- even in the regime where it is not related to the total variation distance.

The authors show how to extend the techniques for list-decodable mean estimation to this setting. This requires non-trivial modifications since covariance estimation in general is a more difficult and much less well-understood task than mean estimation.

The paper is fairly well-written and the authors give a nice proof overview in Section 1.2.

**Weaknesses:**

The relation to IK22 was not clear to me in the beginning, and I believe it would be important to clarify already in the beginning. My main confusion comes from the fact that in Line 49 you say that small relative Frobenius norm is equivalent to small TVD. Then you go on to say that IK22 achieve error in TVD and you in relative Frobenius norm. At this point it seems that the two error guarantees are the same, and it also seems like your algorithm is breaking the SQ lower bound.

Only later you clarify that for the equivalence to hold the relative frobenius norm needs to be smaller than 1. (You do say in Line 49 that it only holds if the relative Frobenius norm is smaller than a small enough constant but do not specify what this is and how it relates to $\mathrm{poly(1/\alpha)}$, your error guarantee.

**Questions:**

You say that your list-decoding algorithm can be used to obtain a "sum-of-squares-free" algorithm for robustly clustering non-spherical Gaussians. Does this come with any (theoretical) speed-ups? If yes, I think it would be good to mention what these are.

**Limitations:**

Maybe I missed this, but I didn't see any statements about limitations in their work.

---

> ### Author Rebuttal · Authors · 2023-08-09
>
>
> We thank the reviewer for their effort and feedback, and address their points below.
>
>
>
> **(Relation to [IK22])**
>
> Thank you for pointing out the issue with phrasing. We will emphasize earlier on in the paper that only relative Frobenius error <= 1 would imply TV distance bounds.
>
>
>
> **(Speed-ups)**
>
> As mentioned in the statement of Theorem 1.2 and Remark F.2 in the Appendix, the runtime of our algorithm is $\tilde{O}(m^2d^2)$, where $m$ is the sample size and $d$ the ambient dimensionality. SDP optimization and SoS algorithms are generally much slower (note that in [BDJKKV22] the runtime is polynomial in these parameters with the power being some large constant). We thus expect a sizable speed-up in the SoS-free version that we provide. If the paper is accepted, we will move the runtime discussion into the main body given the extra page.

---

> > ### Comment · Reviewer_WZLc · 2023-08-14
> >
> > Thank you for your response and answering my questions.

---

### Official Review · Reviewer_Wync · 2023-07-05

**Soundness:** 4 excellent
**Presentation:** 4 excellent
**Contribution:** 4 excellent
**Rating:** 8
**Confidence:** 4

**Summary:**

This paper studies the problem of robustly estimating the covariance of a multivariate Gaussian distribution in the presence of outliers. Specifically, the authors focus on a list-decodable setting, where the algorithm is allowed to output a small list of hypotheses with the guarantee that at least one of them is close to the target. The authors propose a novel polynomial-time algorithm that efficiently list-decodes the covariance of a Gaussian distribution in relative Frobenius norm under adversarial corruption.

**Strengths:**

This is a very nice paper that tackles an important problem in robust statistics, and the result has direct applications in robustly learning GMMs. The contributions of the paper are novel algorithms that significantly improve the state-of-art algorithm in terms of sample complexity and answer size. The paper is very well written and it is a pleasure to read. In our opinion, this is an important contribution to the literature.

**Weaknesses:**

The organization of this paper can be improved. In particular, I think the stability section deserves some space in the main paper and
the paper could benefit from a discussion section at the end of the paper.

**Questions:**

 *

---

> ### Author Rebuttal · Authors · 2023-08-09
>
> We thank the reviewer for their effort and comments. If the paper is accepted, we will move content into the main body as suggested, using the extra page allowed in the camera ready version.

---

### Official Review · Reviewer_64Ly · 2023-07-06

**Soundness:** 4 excellent
**Presentation:** 3 good
**Contribution:** 4 excellent
**Rating:** 7
**Confidence:** 4

**Summary:**

Summary: This paper studies the following problem: given a collection of datapoints in $R^d$, of which a small $\alpha$ fraction (inliers) are guaranteed to be drawn from an unknown Gaussian distribution, and the rest are drawn adversarially (outliers), the goal is approximately identify the distribution. Note that for $\alpha < 1/2,$ it is not possible to uniquely identify the distribution, but one could hope to output a list of $\alpha^{-1}$ hypothesis distributions, one of which is correct (this is called list decoding).

This problem has received significant attention recently from a computational perspective. By now, several algorithms are known for list-decoding in these settings. However, most of them are based on solving large, expensive convex programs (sums-of-squares SDPs) and require $d^{poly(1/\alpha)}$ running time and sample complexity. While the strongest of these algorithms can recover a list of hypotheses such that at least one of them has a bounded total variation distance from the unknown Gaussian, there are lower bounds showing that such a recovery requires  $d^{poly(1/\alpha)}$ samples for a wide class of algorithms.

This paper presents a significantly simpler and faster algorithm ($poly(d/\alpha)$ time and samples) that is able to generate a list of $O(1/\alpha)$ hypotheses such that at least one of the covariance matrices is close to the original covariance matrix in relative Frobenius norm. This is necessarily a weaker guarantee than total variation distance, since it bypasses the known lower bounds.

**Strengths:**


Strengths:
+ A fairly simple to describe algorithm based on multi-filtering
+ avoids solve large convex programs
+ achieves a reasonable $poly(d/\alpha)$ sample complexity and running time guarantee, much better than most previous algorithms

**Weaknesses:**

The main theorems in the paper only talk of recovering the covariance matrix, but not a pair $(\tilde{\mu}, \tilde{\Sigma})$ where both the mean and covariance are close. Are these guarantees also achievable? If yes, I strongly suggest the authors include such a complete result


**Questions:**

Your algorithm removes points one by one in line 16, what is the obstacle in removing a large fraction of the points at each step?

Minor comments:
1. In Def 1.1, since the adversary choses all the outliers after looking at the distribution, is it not equivalent to the setting of having exactly $\alpha m$ inliers?
2. Theorem 1.2, first bullet point -- it's awkward to recall within a theorem statement, I suggest removing the sentence "Recall ... $\ell \le \epsilon n$"

**Limitations:**

Theoretical work.
See weakness pointed above.

---

> ### Author Rebuttal · Authors · 2023-08-09
>
>
> We thank the reviewer for their time and effort. We address their questions below:
>
>
>
> **(Guarantees on pairs of covariance-mean candidates)**
>
> Yes, it is possible to output $(\hat{\mu},\hat{\Sigma})$ pairs such that, $|| \hat{\Sigma}^{-1/2}\Sigma \hat{\Sigma}^{-1/2} - I ||_{F} \leq \mathrm{poly}(1/\alpha)$ (as in the paper) and furthermore that $\hat{\mu}$ is a bounded estimate of the true $\mu$ in Mahalanobis norm with respect to $\hat{\Sigma}$: $(\hat{\mu} - \mu) \hat{\Sigma} (\hat{\mu} - \mu) \leq \mathrm{poly}(1/\alpha)$.
>
>
>
> The idea is simple: after running our list-decoding algorithm for $\hat{\Sigma}$, for each candidate $\hat{\Sigma}$, whiten the data by $\hat{\Sigma}$ and run a list-decoding algorithm for candidate means $\hat{\mu}$ (see for example [CSV17, DKK20]).
>
>
>
> To show this works, we need to check the main assumption for the list-decoding algorithm for the mean, that the data distribution has bounded covariance. After whitening, the data is distributed with covariance $\hat{\Sigma}^{-1/2}\Sigma \hat{\Sigma}^{-1/2}$, which has operator norm $\le \mathrm{poly}(1/\alpha)$ by our list-decoding guarantees.
>
>
>
>
> **(Removing single point vs multiple points per filtering iteration)**
>
> Some other filtering-based algorithms in robust statistics filters by flipping an independent coin for each sample (with bias proportional to the score), thus deleting multiple samples for each filter. In these works, this results in slightly better runtime analysis, replacing a factor of $m$ by $d$. However, most of these other works assume a known covariance (or at least, sphericality), which does not apply immediately in our setting. Given that our runtime is $\tilde{O}(m^2d^2)$, and the sample complexity is $d^2$ resulting in a $d^6$ runtime, the improvement would only bring the runtime down to $d^5$ anyway, so we did not pursue this potential improvement. We emphasize that the focus of this work is getting the first SoS-free spectral algorithm, and not getting the fastest runtime. We hope that the tools and techniques in our paper can inspire future work on optimizing the runtime of the algorithm.
>
>
> **(Contamination model)**
>
> The contamination model that we define is slightly stronger in the sense that the adversary can also remove a small fraction of inliers too. Subtractive corruption can introduce biases into the data, and is non-trivial to handle.

---

> > ### Comment · Reviewer_64Ly · 2023-08-18
> >
> > Thank you for your detailed responses.
> > Consider including these comments to your paper, if you think they might be helpful to the reader.

---

### Official Review · Reviewer_ymEk · 2023-07-07

**Soundness:** 4 excellent
**Presentation:** 4 excellent
**Contribution:** 4 excellent
**Rating:** 8
**Confidence:** 3

**Summary:**

The paper focuses on a fundamental problem in the field of robust statistics, the problem of estimating the covariance matrix of a high-dimensional Gaussian distribution, given access to corrupted samples. The amount of corruption allowed in the setting considered is such that there is not enough information for retrieving an estimate of the unknown covariance matrix. Moreover, prior work has provided strong evidence that even if the algorithm is allowed to output a list of candidate matrices, it is computationally intractable to form a list that contains a multiplicative spectral approximation of the ground truth covariance matrix. This work overcomes the existing lower bounds (by relaxing the measure of closeness between matrices) and proposes a simple and efficient algorithm that outputs a short list of matrices that contains (with high probability) a matrix that is close to the ground truth in relative Frobenius norm. En route, it is shown that the same algorithm, when the samples are drawn from a Gaussian Mixture Model (GMM) corrupted with a small fraction of outliers, clusters the samples according to the covariance matrix of the corresponding generating Gaussian, up to error in relative Frobenius norm. The latter result, combined with existing results from [BDJKKV22], provides a new algorithm for robustly learning GMMs that does not use semi-definite programming.

The main algorithm recursively computes the empirical covariances corresponding to each sample (in the current set of samples) and measures the variance of these quantities across the samples (variance of the point-wise empirical covariances). If the variance is small, then the point-wise empirical covariances agree and therefore the empirical covariance of the current set is provided to the output. If the variance is large, the current sample set is either filtered (removing outliers) or split in two sets (each of which will be recursively processed in the next steps) according to whether the empirical covariances seem to have multiple modes when projected on the direction of maximum variance.

**Strengths:**

The paper proposes a simple and efficient algorithm for a fundamental problem in robust statistics. The proposed algorithm also has an application in learning Mixtures of Gaussian distributions, resolving an open problem in the literature. Although the main idea of the algorithm is simple, the analysis is quite delicate and requires a considerable amount of technical work, as well as careful choice of the overall objective (decoding only up to a list and guarantees in terms of the relative Frobenius norm). Moreover, the presentation of the setting, the results, the techniques and the related work are complete and organized and the proofs are split in several lemmas that might be of independent interest.

**Weaknesses:**

One minor presentation issue is that the paper becomes overly technical in some places. For example, in section 1.2 (and in particular in the beginning of the paragraph), it might be preferable to provide some abstract description of the main approach, instead of (or in addition to) defining the particular quantities that the algorithm uses (like the covariance matrix whose eigenvalues determine whether the algorithm stops). In other parts of the paper, however, providing exact (and therefore technical) claims is indeed important (for example in the theorem statements and the proofs).

**Questions:**

My main question concerns the choice of the relative Frobenius norm. I understand the existence of lower bounds for different metrics, the intuition provided in the paper regarding the relative Frobenius norm, as well as the fact that minimizing the relative Frobenius norm has applications in learning GMMs, but what are other metrics that one might hope to get results for (if any)?

**Limitations:**

Yes, limitations are adequately addressed.

---

> ### Author Rebuttal · Authors · 2023-08-07
>
> Thank you for your appreciation of our work and for your question. There are some variants for the relative Frobenius norm one could consider: 1) currently the relative Frobenius norm is measured with respect to the basis of the output $H$, one could ask for measuring with respect to the true $\Sigma$ instead, and 2) instead of Frobenius norm (which is already stronger than the spectral norm), one could perhaps ask for even stronger guarantees such as the nuclear norm. While we believe 1) should not be possible in general, it is an interesting direction to investigate other error norms for future work.
>
> For 1), this should not be possible based on evidence in known SQ lower bounds. Suppose the true $\Sigma$ is very thin in an unknown (and computationally difficult to detect) direction $v$ and unit variance in all other directions. To ensure $\Sigma^{-1/2} H \Sigma^{-1/2} - I$ has a small Frobenius norm, the output $H$ would also have to be (i) small in the direction $v$ and (ii) have roughly unit variance in all other directions. Such an $H$ would contradict known SQ lower bounds. See for example [DKS17,DKPPS21].
>
> For 2), the naïve bound on the relative nuclear norm blows up our guarantees by a factor of $\sqrt{d}$. We don’t know if it is possible to remove this $d$-dependence. The algorithm we propose is intrinsically $\ell_2$ in nature: we consider flattening second moment matrices of samples, and consider the Euclidean geometry of these flattened matrices. The $\ell_2$ geometry of flattened matrices naturally correspond to the Frobenius norm, and so it is unclear if the current algorithm would also offer any guarantees with respect to the relative nuclear norm (which is $\ell_1$ in terms of the eigenvalues of the matrices, instead of $\ell_2$).

---

> > ### Comment · Reviewer_ymEk · 2023-08-11
> >
> > Thank you for your detailed response to my question.

---

### Author Rebuttal · Authors · 2023-08-07

We thank the reviewers for the in-depth reviews, and for the positive assessment of our paper. We are particularly encouraged by reviewers recognizing our paper for 1) tackling a fundamental/important problem in robust statistics (Reviewers ymEk, Wync, WZLc), and 2) proposing a simple and efficient algorithm based on multi-filtering (Reviewers ymEk, 64Ly, Wync, WZLc). We are also pleased that the reviewers found our paper well-written (Reviewers ymEk, Wync, WZLc), and we will take into account the further suggestions to improve certain parts of the presentation, particularly using the extra page available in the camera-ready version, if the paper is accepted.

---

### Decision · Program_Chairs · 2023-09-21

**Decision:**

Accept (spotlight)

**Comment:**

The paper addresses the problem of estimating the covariance matrix of a high-dimensional Gaussian distribution from corrupted samples in robust statistics. It introduces a simple and efficient algorithm that outputs a concise list of matrices, containing a matrix close to the true covariance in relative Frobenius norm. The algorithm computes empirical covariances for each sample, filters or splits the sample sets based on variance, and provides accurate estimates when projections on the direction of maximum variance show multiple modes in empirical covariances. The approach extends to Gaussian Mixture Models and provides a new algorithm for robustly learning GMMs without using semi-definite programming. While some sections are highly technical, the paper's presentation is organized and comprehensive, and it tackles an open problem in the field with significant contributions. One referee wondered about the choice of relative Frobenius norm over other metrics and potential metrics for obtaining results.